# Coupled Training with Privileged Information and Unlabeled Data

**Jiahao Shi** [1]  **Omar Hagrass** [2]  **Jason M. Klusowski** [2]

## Abstract

In many prediction problems, we have extra information during training (for example, measurements that are expensive or slow to collect) that will not be available when the model is deployed. A common strategy is to first train a model that uses all training information, then use its predictions on unlabeled examples to train a second model that only uses the inputs available at test time. However, when the extra training-only information is weak or noisy, this Two-Stage approach can mislead the deployment model and even hurt accuracy. We propose a joint training method that learns the two models together, so the deployment model can benefit from the extra information only when it actually helps, instead of inheriting its mistakes. We provide guarantees that describe when joint training improves prediction accuracy and analyze a simple alternating training algorithm for large, high-dimensional models. Experiments on synthetic data and real-world prediction tasks show that our approach avoids these failures and robustly outperforms standard Two-Stage baselines.

## 1. Introduction

Many modern learning settings provide extra information during training that cannot be used at deployment. Alongside the primary features $X$ (e.g., an image or routine clinical measurements) and responses $Y$ (e.g., a continuous outcome such as disease severity or a lab value), training examples may also include privileged features $W$ (e.g., expensive biomarkers, specialist assessments). Because $W$ may be unavailable, costly, or impractical to obtain at test time, the deployed predictor must depend only on $X$. At the same time, a large fraction of the training data may contain $W$ but not the labels. The goal is therefore to use $W$ to

[1]Department of Electrical and Computer Engineering, Princeton University [2]Department of Operations Research and Financial Engineering, Princeton University. Correspondence to: Jiahao Shi <js6028@princeton.edu>.

*Proceedings of the 43$^{rd}$ International Conference on Machine Learning*, Seoul, South Korea. PMLR 306, 2026. Copyright 2026 by the author(s).

improve training while still producing a final predictor that uses only $X$. This setting is motivated by many applications, including medical domains where reliable labels are expensive (Rajkomar et al., 2019; Fries et al., 2019), forecasting and longitudinal studies where intermediate or future measurements are observed only during training (Hogan et al., 2004), as well as transfer learning (Zhuang et al., 2020) and distribution shift settings (Koh et al., 2021).

**Problem Formulation.** We formalize this setting as follows. Let $\mathcal{Z} = \mathcal{X} \times \mathcal{W}$ and define $Z = (X, W) \in \mathcal{Z}$. We observe $n$ labeled samples

$$\mathscr{D}_L = \{(Z_i, Y_i)\}_{i=1}^n \overset{\text{i.i.d.}}{\sim} P_{X,W,Y},$$

and $m := N - n \gg n$ unlabeled samples

$$\mathscr{D}_U = \{Z_j\}_{j=n+1}^N \overset{\text{i.i.d.}}{\sim} P_{X,W},$$

where $P_{X,W,Y}$ is a distribution on $\mathcal{X} \times \mathcal{W} \times \mathbb{R}$ and $P_{X,W}$ is its marginal over $\mathcal{Z}$.

Our goal is to estimate a prediction rule that minimizes the expected risk

$$\mathbb{E}[\ell\,(Y, f(X))],$$

where $\ell$ is a general loss function. When $\ell$ is the square loss, i.e., $\ell(y, y') = (y - y')^2$, the regression function

$$\mu(x) := \mathbb{E}[Y \mid X = x],$$

uniquely minimizes the squared risk $\mathbb{E}\big[(Y - f(X))^2\big]$ over measurable functions $f : \mathcal{X} \to \mathbb{R}$, assuming $\mathbb{E}\big[Y^2\big] < \infty$. Given this setup, the central question is whether we can use the privileged information $W$ to achieve better performance than training on $X$ alone, while still ensuring that the deployed predictor depends only on $X$.

**Two-Stage Procedure.** One common way to leverage privileged data is through a Two-Stage procedure. In the first stage, we estimate a pseudo-response function $\hat{g}$, which we refer to as the *rich-view model*, using all available labeled samples $\{(Z_i, Y_i)\}_{i=1}^n$ by minimizing the empirical risk

$$\hat{g} \in \underset{g \in \mathcal{G}}{\arg\min} \; \frac{1}{n} \sum_{i=1}^n \ell\,(Y_i, g(Z_i)),$$

where $\mathcal{G}$ is a collection of functions defined on $\mathcal{Z}$. The fitted function $\hat{g}$ is then used to transfer knowledge by imputing

pseudo-responses $\hat{Y}_j = \hat{g}(Z_j)$ (or $\hat{Y}_j = \mathbf{1}(\hat{g}(Z_j) \geq 1/2)$ for 0/1 binary classification) for the unlabeled samples $\{Z_j\}_{j=n+1}^N$. In the second stage, we estimate the target function $\hat{f}$, which we refer to as the *deployment model*, from the combined dataset $\{(X_i, Y_i)\}_{i=1}^n \cup \{(X_j, \hat{Y}_j)\}_{j=n+1}^N$ by minimizing

$$\hat{f} \in \operatorname*{argmin}_{f \in \mathcal{F}} \frac{1}{N} \left( \sum_{i=1}^n \ell\left(Y_i, f(X_i)\right) \right.$$
$$\left. + \sum_{j=n+1}^N \ell\left(\hat{Y}_j, f(X_j)\right) \right),$$

where $\mathcal{F}$ is a collection of functions defined on $\mathcal{X}$. This general strategy was studied in Hou et al. (2023) for specific parametric models and extended to more general nonparametric settings in Xia & Wainwright (2024).

Although it is appealingly simple, this Two-Stage pseudo-labeling strategy can be fragile when the privileged features $W$ provide only weak or noisy information about the response. In such regimes, the first-stage estimator $\hat{g}$ may produce pseudo-responses that are poorly aligned with the true regression function $\mu$, and these errors are then propagated to the second stage through the imputed labels. As a consequence, the deployment model $\hat{f}$ may be biased toward artifacts introduced by $\hat{g}$, leading to degraded performance relative to training on the labeled data alone. This phenomenon, often referred to as *negative transfer*, is especially pronounced when the predictive advantage of $(X, W)$ over $X$ is marginal, or when the privileged variables contain high-dimensional nuisance components that overwhelm their useful signal. In such settings, naive Two-Stage procedures can perform strictly worse than supervised learning using only the labeled samples, despite access to abundant unlabeled data (Xia & Wainwright, 2024). These limitations motivate approaches that more carefully control the influence of privileged information and adaptively balance the contribution of $W$ during training.

**Our Contributions.** We summarize our main contributions as follows.

- **Coupled Training Framework.** We propose a unified estimation framework that jointly learns the deployment and rich-view models. By flexibly calibrating the influence of privileged information, our approach effectively mitigates the negative transfer often observed in Two-Stage pseudo-labeling methods.

- **Efficient Algorithms.** We develop efficient implementations of the coupled training framework and extend it to high-dimensional settings using a greedy forward selection procedure with provable optimization guarantees.

- **Theoretical and Empirical Analysis.** We provide theoretical insights into the conditions under which unlabeled data improves estimation and demonstrate, through synthetic experiments and real-world regression and classification benchmarks, that our method consistently outperforms standard baselines when privileged signals are noisy or weak.

### 1.1. Pseudo-Response Calibration with Coupled Training

The main limitation of the Two-Stage approach is that the pseudo-responses produced by the rich-view model are treated as fixed targets in the second stage, with no mechanism to correct or attenuate their influence when the privileged signal is weak or noisy. To address this issue, we propose a coupled training procedure that jointly estimates the deployment model $f$ and the rich-view model $g$ through the updates below. The core idea is to allow information to flow in both directions, with $g$ providing pseudo-responses that enrich the effective sample size for learning $f$, while the current estimate of $f$ is in turn used to recalibrate $g$ on the unlabeled data, subject to an explicit constraint that limits deviation from the labeled responses. This coupling yields a flexible regularization of the privileged information and provides a principled mechanism for avoiding negative transfer.

We now describe the resulting algorithm and its associated optimization formulation. For simplicity, we focus on regression, but the same ideas can be adapted for classification, e.g., by treating $\hat{Y}$ as soft labels and using logistic loss.

**Coupled Training Algorithm.** Initialize with any feasible $g_0 \in \mathcal{G}$ and fix a constraint level $\nu \geq 0$. For $k = 1, 2, \ldots$, alternate until convergence.

1. **Update** $f$**.** Given the current pseudo-responses $\hat{Y}_j = g_{k-1}(Z_j)$, update $f$ by solving

$$f_k \in \operatorname*{argmin}_{f \in \mathcal{F}} \frac{1}{N} \left( \sum_{i=1}^n \ell\left(Y_i, f(X_i)\right) \right.$$
$$\left. + \sum_{j=n+1}^N \ell\left(g_{k-1}(Z_j), f(X_j)\right) \right).$$

2. **Recalibrate** $g$**.** Given the current deployment model $f_k$, update $g$ by solving

$$g_k \in \operatorname*{argmin}_{g \in \mathcal{G}} \frac{1}{m} \sum_{j=n+1}^N \ell\left(g(Z_j), f_k(X_j)\right)$$

$$\text{s.t.} \quad \frac{1}{n} \sum_{i=1}^n \ell\left(Y_i, g(Z_i)\right) \leq \nu.$$

If $\mathcal{F} \subset L^2(\mathcal{X})$ and $\mathcal{G} \subset L^2(\mathcal{Z})$ are convex function classes, the loss $\ell(y, y')$ is jointly convex in its arguments $(y, y')$ (e.g., square loss) and the constraint set is feasible, then the

coupled training algorithm gives exact updates for the joint convex optimization problem.

$$\min_{(f,g)\in\mathcal{F}\times\mathcal{G}} \frac{1}{N}\left(\sum_{i=1}^{n}\ell\left(Y_i, f(X_i)\right)\right.$$
$$\left.+\sum_{j=n+1}^{N}\ell\left(g(Z_j), f(X_j)\right)\right)$$
$$\text{s.t.}\quad \frac{1}{n}\sum_{i=1}^{n}\ell\left(Y_i, g(Z_i)\right)\leq\nu. \tag{1}$$

In particular, the objective value is nonincreasing along the iterates. Under standard regularity conditions for these exact updates, every cluster point is a stationary point, and hence globally optimal by convexity for (1); see Grippo & Sciandrone (2000).

The implementation of the coupled training algorithm depends on the model class. For linear models with square loss, the penalized joint objective is quadratic in the coefficients of $(f,g)$, so its first-order conditions reduce to a single block linear system. Thus, we solve this system in closed form. For differentiable nonlinear models, such closed-form equations are unavailable, so we compute the same coupled training updates using gradient-based solvers. For tree ensembles, we use the penalized Lagrangian form in (2), so each alternating update reduces to a standard weighted tree regression. For high-dimensional dictionary models, the updates are computed using the greedy selection routine in Section 3. See Appendix B for implementation details. Throughout, "joint" training means that $f$ and $g$ are coupled through one objective, not that every experiment uses a single simultaneous optimizer over all parameters.

Henceforth, we restrict attention to the square loss $\ell(y,y') = (y - y')^2$, which allows for a tractable analysis. Of course, the proposed coupled training algorithm is not restricted to this choice of loss.

We identify any $f : \mathcal{X} \to \mathbb{R}$ with its lift $\tilde{f} : \mathcal{Z} \to \mathbb{R}$, defined by $\tilde{f}(x,w) = f(x)$; below, we simply write $f$ for this lift when no confusion is possible.

**Negative Transfer.** The constraint level $\nu$ governs how much the rich-view model $g$ is allowed to deviate from the labeled responses, and therefore how strongly privileged information can influence the deployment model $f$. In particular, feasibility requires

$$\nu \geq \min_{g\in\mathcal{G}} \frac{1}{n}\sum_{i=1}^{n}(Y_i - g(Z_i))^2.$$

If $\mathcal{F} \subseteq \mathcal{G}$ (identifying $f \in \mathcal{F}$ with its lift $\tilde{f}(x,w) = f(x)$), then choosing $\nu$ large enough that the labeled least-squares fit over $\mathcal{F}$ is feasible with $g = f$ makes the unlabeled agreement term vanish, and the solution reduces to the labeled least-squares fit over $\mathcal{F}$. In contrast, choosing $\nu$ close to the minimum labeled risk over $\mathcal{G}$ forces $g$ to be (approximately) the labeled least-squares fit over $\mathcal{G}$, yielding behavior akin to the aforementioned Two-Stage pseudo-labeling procedure. Thus, choosing $\nu$ too small can overemphasize noisy pseudo-responses and cause negative transfer, while choosing larger $\nu$ attenuates the influence of $g$ and interpolates back toward supervised learning that ignores $W$.

To illustrate this behavior, we consider a linear model

$$Y = \beta^\top X + \theta^\top W + \varepsilon,$$

where $(X, W)$ are jointly Gaussian and $\varepsilon$ is independent mean-zero Gaussian noise. The strength of the privileged feature is governed by $\|\theta\|_2$, while all other aspects of the data-generating process are held fixed.

Figure 1 illustrates the performance of the proposed method under varying levels of privileged signal strength. In panel (a), where $\|\theta\|_2$ is large, the privileged variables are highly informative, and the optimal performance is achieved for small values of $\nu$. In this regime, the method behaves like a Two-Stage procedure, leveraging the strong privileged signal to improve estimation accuracy.

In contrast, panel (b) corresponds to a setting in which the privileged signal is weak. Here, aggressively incorporating pseudo-responses can degrade performance, yielding worse error than the least-squares estimator (OLS) based only on the labeled pairs $\{(X_i, Y_i)\}_{i=1}^n$. This phenomenon of negative transfer was also noted in Xia & Wainwright (2024, Section 2.4.2) as a limitation of the Two-Stage approach.

Together, these results show that the proposed coupled training framework adaptively moderates the influence of the rich-view model, interpolating between regimes in which privileged information helps or hurts prediction, and thereby yields a robust, principled approach to leveraging privileged information.

### 1.2. Related Work

We place our coupled training framework in context by reviewing connections to learning using privileged information, semi-supervised learning (especially pseudo-labeling), multi-view agreement methods, and related optimization approaches.

**Learning Using Privileged Information.** Our setup is closely related to Learning Using Privileged Information (LUPI) (Vapnik & Vashist, 2009; Vapnik et al., 2015), where privileged variables $W$ are available during training but the goal is to learn a predictor that depends only on $X$. However, in contrast to our framework, classical LUPI formulations assume fully labeled data and do not leverage unlabeled samples.

**Semi-Supervised Learning.** Learning from both labeled and unlabeled data has been widely studied across statistics and machine learning under the name semi-supervised learning (SSL) (Chapelle et al., 2006).

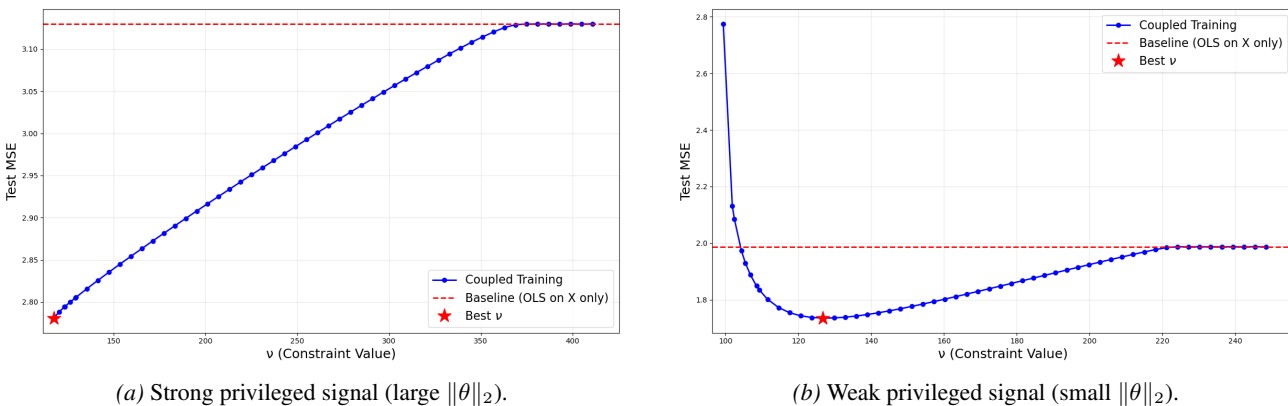

*(a)* Strong privileged signal (large $\|\theta\|_2$).

*(b)* Weak privileged signal (small $\|\theta\|_2$).

*Figure 1.* **Linear Gaussian signal strength.** Performance of the proposed method under varying levels of privileged signal strength.

Among SSL methods, pseudo-labeling is a particularly prominent approach. In classification, pseudo-labeling was explored in the context of support vector machines by Joachims (1999) and later extended to deep learning frameworks by Lee (2013). In regression, pseudo-responses were constructed via kernel smoothing and imputation in linear models (Chakrabortty & Cai, 2018), extended to kernel ridge regression (Wang, 2023), and studied in general settings by Xia & Wainwright (2024). From an inferential perspective, mean estimation with SSL data was investigated by Zhang et al. (2019); Zhang & Bradic (2021), while Laine & Aila (2017) proposed an ensemble-based method that iteratively assigns pseudo-responses using historical predictions. A parallel line of work is weakly supervised learning, which relies on weak or noisy labels rather than ground truth. The data programming paradigm of Ratner et al. (2016; 2017) assumes access to labeling functions and was applied to MRI analysis by Fries et al. (2019). Weak observations were also used for data augmentation by Robinson et al. (2020), who established theoretical guarantees under universal central conditions. Closely related is the literature on learning with noisy labels (Natarajan et al., 2013; Song et al., 2022).

In contrast to these approaches, which primarily rely on pseudo-responses or weak supervision, our framework explicitly controls the influence of privileged information within an adaptive self-training scheme.

**Multi-View Learning.** The objective in (1) is related to the *co-regularized least-squares* objective of Sindhwani et al. (2005), which encourages predictors built from different views to agree on unlabeled data. In their multi-view SSL setting, one observes labeled examples $\{(X_i, Y_i)\}_{i=1}^n$ and unlabeled examples $\{X_i\}_{i=n+1}^N$, where each input $X_i = (X_i^{(1)}, X_i^{(2)})$ comes with two complementary representations $X_i^{(1)} \in \mathcal{X}^{(1)}$ and $X_i^{(2)} \in \mathcal{X}^{(2)}$. The goal is to learn $h = (h^{(1)}, h^{(2)})$ so that each component fits the labeled data in its own view, while their predictions are pushed to match on the unlabeled inputs (see Sindhwani et al. (2005, Section 3)). Related agreement-style objectives also appear in Ding et al. (2022), though their setting is fully supervised and does not use unlabeled data.

In our case, (1) uses a similar kind of agreement term, but for a different reason. We work in a *privileged information* setting where $W$ is available during training but not at deployment, so only $f(X)$ can be used at test time. That makes the coupling inherently asymmetric, with $f$ as the deployment model and $g$ as an auxiliary rich-view model used to pass information to $f$ during training. We also take $g$ to act on the joint input $Z = (X, W)$, rather than on $W$ alone. This lets $g$ capture effects of the privileged variables that depend on $X$ (including $X$–$W$ interactions) and helps translate whatever signal is present in $W$ into improvements in $f$.

**Greedy and Dictionary-Based Optimization.** In Section 3, we restrict the deployment and rich-view models to dictionary spans and fit them using an *alternating forward selection* procedure. This connects our method to the literature on greedy approximation in Hilbert spaces, where algorithms construct an additive predictor by sequentially selecting atoms that best reduce a squared-loss objective and enjoy deterministic approximation guarantees with sublinear objective or approximation error decay (Barron et al., 2008; DeVore & Temlyakov, 1996). It can also be viewed as a block-coordinate variant of classical forward (stepwise) selection in regression (Hastie et al., 2009, Sec. 3.3.2).

Our setting differs in that we perform greedy atom selection within each block while alternating between updating $f$ with $g$ fixed and updating $g$ with $f$ fixed, so the greedy search is carried out in a product space with interdependent residuals. Although support selection is combinatorial, we prove that the alternating greedy iterates achieve a global sublinear decay of the empirical coupled objective (Theorem 3.1), extending classical greedy approximation analyses to the coupled privileged information setting. This

is complementary to blockwise optimization analyses for nonconvex problems such as dictionary learning, which typically focus on reconstruction or coefficient recovery under structural assumptions (Chatterji & Bartlett, 2017; Agarwal et al., 2014; Ruetz & Schnass, 2023); here the dictionaries are fixed and our guarantees control the empirical comparison term through an optimization error bound, and translate it to prediction error (risk) for the coupled estimator.

### 1.3. Notation

Let $\mathcal{F} \subset L^2(\mathcal{X})$ and $\mathcal{G} \subset L^2(\mathcal{Z})$ be closed, convex function classes. For a nonempty closed convex set $\mathcal{C}$ in a Hilbert space $(\mathcal{H}, \langle \cdot, \cdot \rangle_{\mathcal{H}})$, let $\Pi_{\mathcal{C}} h \in \arg\min_{c \in \mathcal{C}} \|h - c\|_{\mathcal{H}}^2$ denote the (metric) projection. When $\mathcal{C}$ is a closed linear subspace, $\Pi_{\mathcal{C}}$ coincides with the orthogonal projector.

For fixed points $x^N = (x_1, \ldots, x_N) \in \mathcal{X}^N$ and $z^N = (z_1, \ldots, z_N) \in \mathcal{Z}^N$, where $z_i = (x_i, w_i)$, let $\hat{P}_N^X$ and $\hat{P}_N^Z$ denote the corresponding normalized empirical measures. For functions $f : \mathcal{X} \to \mathbb{R}$ and $g : \mathcal{Z} \to \mathbb{R}$, we write $\|f\|_{L^p(\hat{P}_N^X)} := \left( \frac{1}{N} \sum_{i=1}^N |f(x_i)|^p \right)^{1/p}$, and $\|g\|_{L^p(\hat{P}_N^Z)} := \left( \frac{1}{N} \sum_{i=1}^N |g(z_i)|^p \right)^{1/p}$. For a class $\mathcal{F} \times \mathcal{G}$, define the max norm $\|(f, g)\|_{1,\infty} = \max\{\|f\|_{L^1(\hat{P}_N^X)}, \|g\|_{L^1(\hat{P}_N^Z)}\}$. An $\epsilon$-cover of $\mathcal{F} \times \mathcal{G}$ with respect to $\|\cdot\|_{1,\infty}$ is a finite collection $\{(f_1, g_1), \ldots, (f_M, g_M)\} \subset \mathcal{F} \times \mathcal{G}$ such that every $(f, g) \in \mathcal{F} \times \mathcal{G}$ is within $\epsilon$ of some $(f_j, g_j)$ under this norm, and the minimal such $M$ is denoted by $\mathcal{N}_{1,\infty}(\epsilon, \mathcal{F} \times \mathcal{G}, z^N)$.

## 2. Risk Bounds

**Lagrangian Relaxation.** The constrained problem (1) admits a Lagrangian relaxation, leading to the penalized objective

$$\hat{\mathcal{L}}(f, g; \lambda) := \frac{1}{N} \left( \sum_{i \leq n} (Y_i - f(X_i))^2 \right.$$
$$\left. + \sum_{j > n} (g(Z_j) - f(X_j))^2 + \lambda \sum_{i \leq n} (Y_i - g(Z_i))^2 \right). \tag{2}$$

Under the usual convex duality conditions for (1), for example feasibility together with a suitable constraint qualification, the constrained problem admits KKT multipliers. In that case, for a primal solution $(\hat{f}, \hat{g})$ there exists a multiplier $\tilde{\lambda} \geq 0$ such that $(\hat{f}, \hat{g})$ minimizes the penalized objective (2) with $\lambda = \tilde{\lambda}$ and satisfies the complementary-slackness condition

$$\tilde{\lambda} \left( \frac{1}{n} \sum_{i=1}^n (Y_i - \hat{g}(Z_i))^2 - \nu \right) = 0.$$

In the rest of the paper we treat $\lambda$ as a tuning parameter and work directly with the penalized formulation.

Note that $\lambda$ plays a role analogous to $\nu$, but in the opposite direction. Smaller $\lambda$ relaxes the effective constraint on $g$ and

attenuates the influence of pseudo-responses, while larger $\lambda$ forces $g$ to fit the labeled data more closely and increases the impact of pseudo-labeling. In particular, if $\mathcal{F} \subseteq \mathcal{G}$, then as $\lambda \to 0$, the unlabeled data play no role and the solution reduces to the ordinary least-squares fit based only on the labeled data. As $\lambda \to \infty$, the solution approaches the Two-Stage procedure. In this case, since $\hat{\mathcal{L}}(\hat{f}, \hat{g}; \lambda) \leq \hat{\mathcal{L}}(\hat{f}, \hat{f}; \lambda)$, we also have the bound $\frac{1}{m} \sum_{j=n+1}^N (\hat{Y}_j - \hat{f}(X_j))^2 \leq \frac{\lambda n}{m} \cdot \frac{1}{n} \sum_{i=1}^n (Y_i - \hat{f}(X_i))^2$, meaning that the (pseudo-) training error of $\hat{f}$ on the unlabeled features is at most $\lambda n/m$ times the training error of $\hat{f}$ on the labeled features.

**Population Minimizers.** We now turn to a theoretical analysis of the proposed method and derive bounds on the prediction error of the learned deployment model $\hat{f}$. To this end, we consider the population-level counterpart of the empirical objective in (2), given by

$$\mathcal{L}(f, g; \lambda) := \frac{n}{N} \mathbb{E}[(Y - f(X))^2]$$
$$+ \frac{m}{N} \mathbb{E}[(g(Z) - f(X))^2]$$
$$+ \lambda \frac{n}{N} \mathbb{E}[(Y - g(Z))^2]. \tag{3}$$

Recall we identify any $f : \mathcal{X} \to \mathbb{R}$ with its lift to $\mathcal{Z}$ via $\tilde{f}(x, w) = f(x)$. We define the rich-view regression function

$$\eta(z) := \mathbb{E}[Y \mid Z = z],$$

which is related to $\mu$ via $\mu(x) = \mathbb{E}[\eta(Z) \mid X = x]$.

**Theorem 2.1** (Population Minimizers). *Any $(f^\star, g^\star) \in \mathcal{F} \times \mathcal{G}$ minimizing (3) satisfies the fixed point equations.*

$$f^\star = \Pi_{\mathcal{F}} \left( \frac{n}{N} \mu + \frac{m}{N} \mathbb{E}[g^\star(Z) \mid X] \right),$$
$$g^\star = \Pi_{\mathcal{G}} \left( \frac{m}{m + n\lambda} f^\star + \frac{n\lambda}{m + n\lambda} \eta \right).$$

*Furthermore, if $\mu \in \mathcal{F}$, $\mu \in \mathcal{G}$, and $\eta \in \mathcal{G}$, then*

$$f^\star = \mu, \quad and \quad g^\star = \frac{m}{m + n\lambda} \mu + \frac{n\lambda}{m + n\lambda} \eta.$$

*Remark* 2.2 (Interpolation Effect). The characterization in Theorem 2.1 highlights how the rich-view model $g^\star$ interpolates between the deployment target $\mu$ and the rich-view regression function $\eta$. In particular, under the realizability assumptions in Theorem 2.1, as $\lambda \to 0$, the penalty on the rich-view model becomes inactive and the solution satisfies $g^\star \to \mu$, corresponding to a regime in which the privileged information is ignored. On the other hand, as $\lambda$ increases, $g^\star$ places increasing weight on $\eta$, recovering behavior akin to a Two-Stage pseudo-labeling procedure. Intermediate values of $\lambda$ yield an interpolation between these extremes, allowing the method to adapt to the informativeness of the privileged variables.

**Correlation Controlled Risk Bound.** We now study the statistical behavior of the proposed estimator by analyzing its prediction error. Let $(\hat{f}, \hat{g}) \in \mathcal{F} \times \mathcal{G}$ denote the solution (or an approximate solution) of the empirical problem (2) and let $(f^\star, g^\star) \in \mathcal{F} \times \mathcal{G}$ denote the minimizer of (3). Our goal is to control the excess risk

$$\mathbb{E}_{\mathscr{D}, X}\left[(\hat{f}(X) - \mu(X))^2\right],$$

where the expectation is over the training data $\mathscr{D}$ used to construct $\hat{f}$ and an independent test point $X$.

A key quantity in our analysis is a correlation coefficient that captures the interaction between the estimation errors of the deployment and rich-view models. Specifically, define the estimation errors

$$\hat{e}_f(X) := f^\star(X) - \hat{f}(X), \qquad \hat{e}_g(Z) := g^\star(Z) - \hat{g}(Z).$$

We then define

$$\rho_\star = \frac{\left|\mathbb{E}_{\mathscr{D}, Z}\left[\hat{e}_f(X)\,\hat{e}_g(Z)\right]\right|}{\left(\mathbb{E}_{\mathscr{D}, X}\left[\hat{e}_f^2(X)\right]\right)^{1/2}\left(\mathbb{E}_{\mathscr{D}, Z}\left[\hat{e}_g^2(Z)\right]\right)^{1/2}} \in [0, 1],$$

where $Z = (X, W)$, which measures the alignment between the residuals of the two estimators, with the convention that $\rho_\star = 0$ if either denominator term is zero.

The following corollary, which is a direct consequence of the excess loss decomposition in Theorem C.1, shows that $\rho_\star$ plays a central role in controlling the prediction error.

**Corollary 2.3** (Correlation Controlled Risk Bound). *If $\mu \in \mathcal{F}$, $\mu \in \mathcal{G}$ (note that functions on $\mathcal{X}$ are implicitly lifted to $\mathcal{Z}$), and $\eta \in \mathcal{G}$, then any estimator $(\hat{f}, \hat{g}) \in \mathcal{F} \times \mathcal{G}$ satisfies*

$$\mathbb{E}_{\mathscr{D}, X}\left[(\hat{f}(X) - \mu(X))^2\right]$$
$$\leq \frac{\mathbb{E}_{\mathscr{D}}\left[\mathcal{L}(\hat{f}, \hat{g}; \lambda) - \mathcal{L}(f^\star, g^\star; \lambda)\right]}{\gamma_{n,m,\lambda}(\rho_\star)},$$

*where*

$$\gamma_{n,m,\lambda}(\rho_\star) := 1 - \frac{m^2}{N(m + n\lambda)}\rho_\star^2 \geq \frac{n}{N}.$$

The above bound shows that excess loss in the joint objective translates into prediction error for $\hat{f}$, with a constant that degrades as the correlation $\rho_\star$ between the estimation errors increases.

*Remark* 2.4 (Correlation Score Interpretation). (i) The score $\rho_\star$ measures the alignment between the deployment model and rich-view estimation errors; small $\rho_\star$ yields a sharper conversion from coupled excess loss to deployment risk and corresponds to privileged information contributing variation not already captured by $X$, whereas large $\rho_\star$ limits the gain from incorporating $W$. In this regime, leveraging the privileged information can substantially improve the accuracy

of $\hat{f}$. In contrast, when $\rho_\star$ is large, the residuals of $\hat{f}$ and $\hat{g}$ are highly correlated, suggesting that $W$ provides little additional information beyond what is already contained in $X$, and the benefit of incorporating privileged data becomes limited.

(ii) By Cauchy–Schwarz and the law of total variance, it can be shown (see Section C.4 for details) that

$$\rho_\star^2 \leq \frac{\mathbb{E}_{\mathscr{D}, X}\left[\left(\mathbb{E}[\hat{e}_g(Z) \mid \mathscr{D}, X]\right)^2\right]}{\mathbb{E}_{\mathscr{D}, Z}\left[\hat{e}_g^2(Z)\right]} \leq 1.$$

Furthermore, if $\mu \in \mathcal{F}$, $\mu \in \mathcal{G}$, and $\eta \in \mathcal{G}$, then $\mathbb{E}[g^\star(Z) \mid X] = \mu(X)$, and hence

$$\mathbb{E}[\hat{e}_g(Z) \mid \mathscr{D}, X] = \mu(X) - \mathbb{E}[\hat{g}(Z) \mid \mathscr{D}, X].$$

Consequently, $\rho_\star$ is small when the conditional rich-view model error $\mathbb{E}_{\mathscr{D}, X}\left[\left(\mathbb{E}[\hat{e}_g(Z) \mid \mathscr{D}, X]\right)^2\right]$ is small relative to the total rich-view model error $\mathbb{E}_{\mathscr{D}, Z}\left[\hat{e}_g^2(Z)\right]$. This corresponds to a regime in which the rich-view model $\hat{g}$ leverages privileged information $W$ to capture predictive signal about $Y$ that is not well explained by $X$ alone, thereby reducing the component of the rich-view error that is aligned with $X$. Hence, $\rho_\star$, and consequently the risk bound, are controlled by a *multiplicative* relative error, in contrast to Xia & Wainwright (2024), where the risk is bounded in terms of an *additive* absolute error $\mathbb{E}_{\mathscr{D}, X}\left[\left(\mathbb{E}[\hat{e}_g(Z) \mid \mathscr{D}, X]\right)^2\right]$. This means that the risk bound degrades more gracefully as $\hat{g}$ worsens, because it depends on the fraction of rich-view error explainable from $X$ rather than on its absolute size.

### 2.1. Main Risk Bound

**Assumption 2.5.** The function classes $\mathcal{F}$ and $\mathcal{G}$ are uniformly bounded. Specifically, there exists a constant $B > 0$ such that

$$\sup_{(f,g)\in\mathcal{F}\times\mathcal{G}} \|(f, g)\|_\infty \leq B \quad \text{and} \quad |Y| \leq B, \quad \text{a.s.},$$

where $\|(f, g)\|_\infty := \max\left\{\|f\|_\infty, \|g\|_\infty\right\}$.

*Remark* 2.6 (On the boundedness assumption). The condition $\sup_{(f,g)\in\mathcal{F}\times\mathcal{G}} \|(f, g)\|_\infty \leq B$ is made for convenience. When $\mathcal{F}$ and $\mathcal{G}$ are unbounded but $|Y| \leq B$ a.s., the same conclusions apply to the truncated estimators $T_B\hat{f}$ and $T_B\hat{g}$, with $T_B(t) := \text{sign}(t)\,(|t| \wedge B)$ following the truncation argument of Barron et al. (2008). Since $|Y| \leq B$ a.s., truncation can only decrease the (squared) loss, so no rate is lost.

We introduce a complexity term that captures the joint richness of the deployment and rich-view function classes, at scales determined by the sample sizes and the regularization parameter $\lambda$. Define

$$\mathfrak{E}_{\mathcal{F}\times\mathcal{G}}^{N,\lambda} := \sup_{z^N} \log\left(\mathcal{N}_{1,\infty}\left(\epsilon_N, \mathcal{F} \times \mathcal{G}, z^N\right)\right),$$

with $\epsilon_N = \frac{1}{160B(\lambda+1)^{3/2}N}$.

**Theorem 2.7** (Risk Bound). *If $\mu \in \mathcal{F}$, $\mu \in \mathcal{G}$, $\eta \in \mathcal{G}$ and Assumption 2.5 holds, then any estimator $(\hat{f}, \hat{g}) \in \mathcal{F} \times \mathcal{G}$ satisfies*

$$\mathbb{E}_{\mathscr{D}, X}\left[(\hat{f}(X) - \mu(X))^2\right]$$

$$\leq \frac{1}{\gamma_{n,m,\lambda}(\rho_\star)}\left(\frac{C_1 B^2 (\lambda + 1)\mathfrak{E}_{\mathcal{F} \times \mathcal{G}}^{N,\lambda}}{N}\right. \tag{4}$$

$$\left. + 2\mathbb{E}_{\mathscr{D}}\left[\hat{\Delta}(\hat{f}, \hat{g})\right]\right),$$

*where $C_1 > 0$ is a universal constant, and*

$$\hat{\Delta}(\hat{f}, \hat{g}) := \hat{\mathcal{L}}(\hat{f}, \hat{g}; \lambda) - \hat{\mathcal{L}}(f^\star, g^\star; \lambda).$$

*Remark* 2.8 (Empirical Comparison Term). The term $\hat{\Delta}(\hat{f}, \hat{g})$ is an empirical comparison term, comparing the empirical coupled objective attained by the estimator $(\hat{f}, \hat{g})$ with that of the population reference pair $(f^\star, g^\star)$. It is not necessarily nonnegative, since $(f^\star, g^\star)$ need not minimize the empirical objective. In particular, if $(\hat{f}, \hat{g})$ is an exact empirical minimizer of (2), then

$$\hat{\Delta}(\hat{f}, \hat{g}) \leq 0,$$

so this term can be dropped from (4). For algorithmically constructed estimators, $\hat{\Delta}(\hat{f}, \hat{g})$ records how well the computed pair competes with $(f^\star, g^\star)$ on the empirical objective; for the alternating forward selection iterates, this comparison is controlled by Theorem 3.1.

*Remark* 2.9 (Effect of Correlation Score). (i) Note that, for any values of $\rho_\star$ and $\lambda$, the quantity $\gamma_{n,m,\lambda}(\rho_\star)$ satisfies the uniform lower bound $\gamma_{n,m,\lambda}(\rho_\star) \geq n/N$. As a consequence,

$$\frac{1}{\gamma_{n,m,\lambda}(\rho_\star)} \cdot \frac{1}{N} \leq \frac{1}{n},$$

so the $O(1/N)$ term in (4) is never worse than the $O(1/n)$ dependence obtained from using only the labeled data.
(ii) When the estimation errors of the deployment and rich-view estimators are highly correlated (i.e., $\rho_\star$ is close to 1), one has $\gamma_{n,m,\lambda}(\rho_\star) \approx 1 - m^2/(N(m + n\lambda))$. In particular, when $\lambda = 0$, this becomes $\gamma_{n,m,0}(\rho_\star) \approx n/N$, and hence

$$\frac{1}{\gamma_{n,m,0}(\rho_\star)} \cdot \frac{1}{N} \approx \frac{1}{n},$$

so the $O(1/N)$ term in (4) recovers the $O(1/n)$ dependence, obtained when learning solely from $n$ labeled observations.
(iii) More generally, $\gamma_{n,m,\lambda}(\rho_\star)$ controls the multiplicative constant in (4) through its dependence on $\rho_\star$. When the deployment and rich-view estimation errors are weakly aligned (small $\rho_\star$), $\gamma_{n,m,\lambda}(\rho_\star)$ remains close to one, so the $O(1/N)$ term is not significantly inflated.
(iv) The quantity $\rho_\star$ is introduced only for the analysis, not observed from data, and not used to tune the method. Rather,

it summarizes how much of the rich-view error remains explainable from $X$ alone. Constructing a practical proxy for $\rho_\star$ that could guide the choice of $\lambda$ is an interesting direction for future work.

## 3. High-Dimensional Extension

In Section 1.1, we proposed a coupled training algorithm for jointly estimating the deployment model $\hat{f}$ and the rich-view model $\hat{g}$. In practice, however, directly optimizing over arbitrary functions in infinite-dimensional Hilbert spaces is computationally inefficient, especially when the combined inputs $(X, W)$ are high-dimensional.

To bridge this gap between theory and practice, we adopt a dictionary-based approximation strategy in which functions are represented as sparse linear combinations of elementary functions, or *atoms*, drawn from predefined dictionaries. The resulting procedure is efficient and remains theoretically well-justified. In this section, we introduce a greedy alternating forward selection procedure that incrementally constructs approximations of $(\hat{f}, \hat{g})$ using finite dictionaries. Let $\mathcal{D}_f$ and $\mathcal{D}_g$ be finite, symmetric dictionaries such that $\|\psi\|_{L^2(\hat{P}_N^X)} \leq 1$ for all $\psi \in \mathcal{D}_f$ and $\|\phi\|_{L^2(\hat{P}_N^Z)} \leq 1$ for all $\phi \in \mathcal{D}_g$. For functions $f \in \mathcal{F} = \text{span}(\mathcal{D}_f)$ and $g \in \mathcal{G} = \text{span}(\mathcal{D}_g)$ of the form

$$f = \sum_{\psi \in \mathcal{D}_f} c_\psi \psi, \qquad g = \sum_{\phi \in \mathcal{D}_g} c_\phi \phi,$$

we define the associated atomic norms by

$$\|f\|_{L^1(\mathcal{D}_f)} := \inf\left\{\sum_{\psi \in \mathcal{D}_f} |c_\psi| : f = \sum_{\psi \in \mathcal{D}_f} c_\psi \psi\right\},$$

$$\|g\|_{L^1(\mathcal{D}_g)} := \inf\left\{\sum_{\phi \in \mathcal{D}_g} |c_\phi| : g = \sum_{\phi \in \mathcal{D}_g} c_\phi \phi\right\}.$$

### 3.1. Alternating Forward Selection Algorithm

We now describe a greedy algorithm that alternates between updating $f$ and recalibrating $g$.

The procedure approximately minimizes $\hat{\mathcal{L}}(f, g; \lambda)$ in (2) by alternately updating $f$ and $g$ using forward (stepwise) selection (Hastie et al., 2009, Sec. 3.3.2).

**Alternating Forward Selection Algorithm.** Initialize $g_0 = 0$. For $k = 1, 2, \ldots$, alternate between the following updates.

1. **Update $f$.** Given the current pseudo-responses $\hat{Y}_j = g_{k-1}(Z_j)$, update $f$ by solving

$$(\psi_k, f_k) \in \underset{\substack{\psi \in \mathcal{D}_f \\ f \in \text{span}(\psi_1, \ldots, \psi_{k-1}, \psi)}}{\text{argmin}} \frac{1}{N}\left(\sum_{i=1}^{n}(Y_i - f(X_i))^2\right.$$

$$\left. + \sum_{j=n+1}^{N}(g_{k-1}(Z_j) - f(X_j))^2\right).$$

2. **Recalibrate** $g$. Given the current deployment model $f_k$, update $g$ by solving

$$(\phi_k, g_k) \in \underset{\substack{\phi \in \mathcal{D}_g \\ g \in \mathrm{span}\{\phi_1, \ldots, \\ \phi_{k-1}, \phi\}}}{\mathrm{argmin}} \frac{1}{N} \left( \sum_{j=n+1}^{N} (g(Z_j) - f_k(X_j))^2 \right.$$
$$\left. + \lambda \sum_{i=1}^{n} (Y_i - g(Z_i))^2 \right).$$

An efficient implementation of this procedure is provided in Appendix A (see Algorithm 1). We first establish a convergence rate, then summarize the implementation.

**Theorem 3.1** (Alternating Forward Selection Convergence Rate). *For all $k \geq 1$, and all $f \in \mathrm{span}(\mathcal{D}_f)$ and $g \in \mathrm{span}(\mathcal{D}_g)$, the iterates $(\hat{f}, \hat{g}) = (f_k, g_k)$ produced by the Alternating Forward Selection Algorithm satisfy*

$$\hat{\mathcal{L}}(\hat{f}, \hat{g}; \lambda) \leq \hat{\mathcal{L}}(f, g; \lambda) + C_2 \frac{\max\{1, \lambda\} \log(k+1)}{k}$$
$$\times \left( \|f\|_{L^1(\mathcal{D}_f)} + \|g\|_{L^1(\mathcal{D}_g)} \right)^2,$$

*for a universal constant $C_2 > 0$.*

This result shows that the greedy alternating procedure gives sublinear control of the empirical comparison term through its optimization error contribution. Moreover, when the function classes $\mathcal{F}$ and $\mathcal{G}$ are restricted to the spans of dictionaries $\mathcal{D}_f$ and $\mathcal{D}_g$, respectively, the statistical complexity term in Theorem 2.7 admits an explicit bound in terms of the number of greedy iterations. Specifically, conditional on a fixed selected set of dictionary atoms, equivalently within a fixed selected span, if $(\hat{f}, \hat{g}) = (f_k, g_k)$ lie in $\mathrm{span}(\mathcal{D}_f) \times \mathrm{span}(\mathcal{D}_g)$ with $k$ selected atoms in each component, then applying Barron et al. (2008, Lemma 3.3) to the associated covering numbers yields

$$\mathfrak{E}_{\mathcal{F} \times \mathcal{G}}^{N, \lambda} \leq (k+1) \log \left( C_3 B (\lambda+1)^{3/2} N \right),$$

for some universal constant $C_3 > 0$.
Consequently, under the assumptions of Theorem 2.7, the iterates $(\hat{f}, \hat{g}) = (f_k, g_k)$ produced by Algorithm 1 satisfy

$$\mathbb{E}_{\mathscr{D}, X} \left[ (\hat{f}(X) - \mu(X))^2 \right]$$
$$\leq \frac{1}{\gamma_{n,m,\lambda}(\rho_\star)} \left( \frac{C_1 B^2 (\lambda+1)(k+1)}{N} \right.$$
$$\times \log \left( C_3 B (\lambda+1)^{3/2} N \right)$$
$$+ \frac{2 C_2 \max\{1, \lambda\} \log(k+1)}{k}$$
$$\left. \times \left( \|f^\star\|_{L^1(\mathcal{D}_f)} + \|g^\star\|_{L^1(\mathcal{D}_g)} \right)^2 \right).$$

### 3.2. Efficient Implementation

We briefly describe an efficient implementation of the Alternating Forward Selection Algorithm; full derivations and pseudocode are deferred to Appendix A. At a high level, each block update (the $f$-step over $\mathrm{span}(\mathcal{D}_f)$ and the $g$-step over $\mathrm{span}(\mathcal{D}_g)$) is a least-squares refit over the currently selected atoms. Rather than resolving these refits from scratch, we maintain incremental QR decompositions for the (weighted) design matrices associated with the selected atoms, enabling $O(Nk)$ updates per iteration, where $k$ is the number of selected atoms per block.

**Time complexity.** Assuming each candidate atom can be evaluated on all $N$ samples in $O(N)$ time, selecting the next atom at iteration $k$ amounts to scanning the remaining candidates in the relevant dictionary and computing their correlations with the current residual after orthogonalization against the selected span (via the maintained QR factors). This yields a total cost up to step $k$ of $O(N(|\mathcal{D}_f| + |\mathcal{D}_g|)k)$. For comparison, a naive version of forward selection over paired atoms $(\psi, \phi)$ (i.e., scanning a product dictionary) scales as $O(N|\mathcal{D}_f||\mathcal{D}_g|k)$, which is substantially larger. As shown in Theorem 3.1, this computational improvement comes with only a $\log(k)$ factor difference in the optimization guarantee relative to classical greedy schemes Barron et al. (2008); DeVore & Temlyakov (1996).

## 4. Experiments

We evaluate the coupled objective on controlled synthetic data and two real-world tasks. The tuning parameter $\lambda$ interpolates between labeled-only learning on $X$ and the Two-Stage method. Unless otherwise stated, $\lambda$ is selected by cross-validation using only labeled training data. We compare against (i) supervised learning on $\{(X_i, Y_i)\}_{i=1}^{n}$ using only $X$ (Baseline), and (ii) the Two-Stage procedure from Section 1. The square-loss synthetic controls match our theory, while the logistic synthetic diagnostic, random-forest regression, and Bank Marketing classification experiments are empirical demonstrations beyond the current proof regime.

**Synthetic controls.** We first use a linear model with known target $\mu(X)$ and privileged coordinates $W$ containing both signal and nuisance variation. Figure 2 shows that the coupled method improves as the privileged signal strengthens, remains stable as nuisance privileged dimensions increase, and improves as the number of unlabeled samples increases. In contrast, Two-Stage pseudo-labeling can degrade when the rich-view teacher overfits nuisance variation.

**Synthetic classification diagnostic.** We also evaluate a binary logistic analogue of the coupled objective, replacing squared-loss term in (2) by cross-entropy and using soft rich-view pseudo-labels. As shown in Figure 6 in Appendix B, the test error is minimized at an intermediate value of $\lambda$, while the Two-Stage limit over-transfers rich-view noise.

*Table 1.* **Real-data summary.** Lower is better. For Parkinson's, we report the single-seed subject-level split with $\lambda$ selected by labeled-only CV. For Bank Marketing, entries are mean holdout Brier scores across 26 outer seeds.

| Dataset | Metric | Baseline | Two-Stage | Gen. Distill. | SVM+ | Ours |
|---|---|---|---|---|---|---|
| Parkinson's | MSE | 77.34 | 83.32 | – | – | **72.60** |
| Bank Marketing | Brier | 0.0893 | 0.0928 | 0.0907 | 0.1470 | **0.0881** |

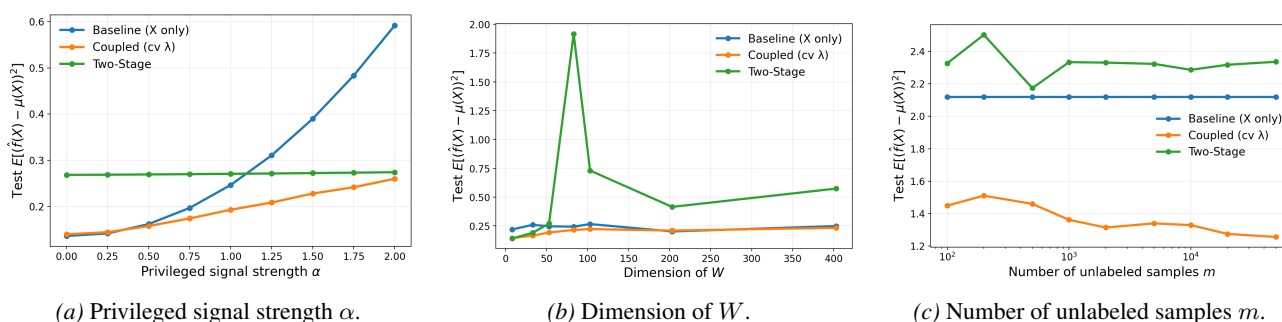

*(a)* Privileged signal strength $\alpha$.     *(b)* Dimension of $W$.     *(c)* Number of unlabeled samples $m$.

*Figure 2.* **Synthetic controls.** Test error is $\mathbb{E}[(\hat{f}(X) - \mu(X))^2]$. Coupled training adapts to useful privileged signal, is more stable than Two-Stage under nuisance privileged dimensions, and improves with additional unlabeled data.

**Real-data benchmarks.** For regression, we use the UCI Parkinson's Telemonitoring dataset and predict disease severity from voice measurements. Basic perturbation measures form the deployment view $X$, while more complex spectral features form the privileged view $W$. Figure 3 in Appendix B shows the expected U-shaped dependence on $\lambda$; Table 1 reports the same subject-level split with the labeled-only CV choice $\hat{\lambda} = 3$. Due to high variance from subject-level splitting, the main table reports single-seed results for Parkinson's. The larger Bank task averages 26 seeds. Cross-validation (CV) robustness checks for both are provided in Appendix B.

For classification, we use the Bank Marketing dataset and evaluate by holdout Brier score. Static demographic, contact, and macroeconomic variables form the deployment view $X$, while current-call and campaign outcome variables form the privileged view $W$. We evaluate over 26 stratified outer splits, using $n = 200$ labeled examples and $m = 10{,}000$ unlabeled paired $(X, W)$ examples per split. The tuning parameter $\lambda$ and all baselines are selected by the same fixed 4-fold cross-validation protocol on the labeled set only. We compare against the $X$-only Baseline, Two-Stage pseudo-labeling, a squared-loss generalized distillation baseline, and SVM+ (Vapnik & Vashist, 2009). Coupled Training beats the $X$-only Baseline and generalized distillation in 22/26 seeds, and beats Two-Stage and SVM+ in 26/26 seeds. Figure 4 in Appendix B shows a representative split, where the cross-validated $\hat{\lambda}$ lands near the interior low-Brier region of the holdout diagnostic curve. Table 1 summarizes the real-data results.

Overall, the experiments support the intended role of Coupled Training, exploiting privileged information when useful while attenuating it when direct pseudo-labeling would transfer nuisance signal into the final $X$-only predictor. Further implementation details and diagnostics are deferred to Appendix B.

## 5. Conclusion

We studied learning with privileged features and unlabeled data when the deployment predictor may only use $X$ at test time. The main difficulty in this setting is negative transfer, where a rich-view predictor trained on $(X, W)$ can be useful when $W$ carries a genuine extra signal, but can also mislead the deployment model when the privileged view is weak or noisy. Our coupled objective addresses this by jointly fitting the deployment model and the rich-view model, rather than treating the rich-view predictor as a fixed teacher.

Our estimator interpolates between labeled-only learning (weak coupling) and Two-Stage pseudo-labeling (strong coupling). This interpolation is reflected both in the population characterization and in the empirical U-shaped dependence on $\lambda$ observed across our synthetic and real data experiments. In addition, for the high-dimensional setting, we developed a dictionary-based AFS procedure and established a sublinear optimization guarantee, which significantly reduces the time complexity.

**Limitations.** Our theory covers square-loss regression with convex function classes, so the random-forest and classification experiments are empirical evidence beyond the present proof setting. The method also assumes paired unlabeled samples with privileged features; extensions to misspecification, distribution shift, semi-paired data, and broader classification guarantees remain future work.

## Acknowledgements

We extend our special thanks to the anonymous reviewers for their insightful comments and suggestions that greatly enhanced this work. Hagrass gratefully acknowledges financial support from the Schmidt DataX Fund at Princeton University made possible through a major gift from the Schmidt Futures Foundation. Klusowski is grateful for support from the National Science Foundation through NSF CAREER DMS-2239448 and the Alfred P. Sloan Foundation through a Sloan Research Fellowship.

## Impact Statement

This paper presents work whose goal is to advance the field of machine learning. There are many potential societal consequences of our work, none of which we feel must be specifically highlighted here.

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

## A. Efficient Implementation of Alternating Forward Selection

Recall from Section 1 that we observe $n$ labeled and $m$ unlabeled samples with $N = n + m$, and that $\mathcal{Z} = \mathcal{X} \times \mathcal{W}$. Let $\pi_X : \mathcal{Z} \to \mathcal{X}$ be the canonical projection $\pi_X(x, w) = x$. Define the finite measures on $\mathcal{Z}$ by

$$\tilde{P}^{(1)} := \sum_{i=1}^{n} \delta_{Z_i}, \qquad \tilde{P}^{(2)} := \sum_{j=n+1}^{N} \delta_{Z_j}, \qquad \tilde{P}^{(3)} := \lambda \tilde{P}^{(1)},$$

and normalize by $N$.

$$\hat{P}_N^{(\ell)} := \frac{1}{N} \tilde{P}^{(\ell)}, \qquad \ell \in \{1, 2, 3\}.$$

Note that $\hat{P}_N^{(1)} + \hat{P}_N^{(2)} = \hat{P}_N^Z$. We will work exclusively with $Z$-indexed norms by viewing the deployment class as a subspace of functions on $\mathcal{Z}$. Define the lifted deployment space

$$\mathcal{H}_f := \{ f \in L^2(\hat{P}_N^Z) : f \text{ is } \sigma(\pi_X)\text{-measurable} \},$$

i.e., $f \in \mathcal{H}_f$ implies $f(x, w)$ depends only on $x$ (equivalently, there exists $\bar{f} : \mathcal{X} \to \mathbb{R}$ such that $f(x, w) = \bar{f}(x)$ $\hat{P}_N^Z$-a.e.). Let $Q_g := \hat{P}_N^{(2)} + \hat{P}_N^{(3)}$ and set

$$\mathcal{H}_g := L^2(Q_g).$$

We assume the function classes live in these Hilbert spaces, $\mathcal{F} \subset \mathcal{H}_f, \mathcal{G} \subset \mathcal{H}_g$. To keep notation light, we identify the model class $\mathcal{F}$ with its lift to $\mathcal{Z}$, namely, we view each $f \in \mathcal{F}$ as a function $f(x, w)$ that is $\sigma(\pi_X)$-measurable (hence depends only on $x$, i.e., is constant in $w$ given $x$). We use $\mathcal{D}_f$ in the same spirit, and throughout the appendix it refers to a dictionary in the lifted space $\mathcal{H}_f$. With this convention, for $f \in \mathcal{H}_f, g \in \mathcal{H}_g$, the empirical objective can be written as

$$\hat{\mathcal{L}}(f, g; \lambda) := \|Y - f\|_{L^2(\hat{P}_N^{(1)})}^2 + \|f - g\|_{L^2(\hat{P}_N^{(2)})}^2 + \|Y - g\|_{L^2(\hat{P}_N^{(3)})}^2.$$

Let

$$\mathcal{K}_1 := L^2(\hat{P}_N^{(1)} + \hat{P}_N^{(2)}), \qquad \mathcal{K}_2 := L^2(\hat{P}_N^{(2)} + \hat{P}_N^{(3)}), \qquad \mathcal{K} := \mathcal{K}_1 \times \mathcal{K}_2.$$

For $u = (u_1, u_2)$ and $v = (v_1, v_2)$ in $\mathcal{K}$, define

$$\langle u, v \rangle_\star = \langle u_1, v_1 \rangle_{L^2(\hat{P}_N^{(1)})} + \langle u_1 - u_2, v_1 - v_2 \rangle_{L^2(\hat{P}_N^{(2)})} + \langle u_2, v_2 \rangle_{L^2(\hat{P}_N^{(3)})}.$$

This bilinear form is positive semidefinite. Let

$$\mathcal{N}_\star := \{u \in \mathcal{K} : \|u\|_\star = 0\}$$

and define the quotient Hilbert space

$$\bar{\mathcal{K}} := \mathcal{K}/\mathcal{N}_\star.$$

We write $[u]$ for the equivalence class of $u$, although, when no confusion is possible, we suppress brackets and use the same notation for a representative and its quotient class. The deployment and rich-view model spaces are embedded as

$$\mathcal{H}^\star := \{[(f, g)] : f \in \mathcal{H}_f, g \in \mathcal{H}_g\} \subset \bar{\mathcal{K}}.$$

Below, all projections, residuals, spans, and orthogonality statements are taken in $\bar{\mathcal{K}}$ with respect to $\langle \cdot, \cdot \rangle_\star$.
Let $Y$ be any extension of the labeled responses to all $N$ sample points, with $Y(Z_i) = Y_i$ for $i \leq n$. Define

$$T := (Y, Y) \in \bar{\mathcal{K}}.$$

The choice of $Y$ on unlabeled samples is irrelevant, since the unlabeled term depends on $(Y - f) - (Y - g) = g - f$. Hence, for every $f, g$,

$$\|T - (f, g)\|_\star^2 = \|Y - f\|_{L^2(\hat{P}_N^{(1)})}^2 + \|f - g\|_{L^2(\hat{P}_N^{(2)})}^2 + \|Y - g\|_{L^2(\hat{P}_N^{(3)})}^2 = \hat{\mathcal{L}}(f, g; \lambda).$$

Without loss of generality, we suppose that the dictionaries $\mathcal{D}_f$ and $\mathcal{D}_g$ are bounded by $1$ in the *pooled* empirical norm, i.e.,

$$\sup_{\psi \in \mathcal{D}_f} \|\psi\|_{L^2(\hat{P}_N^Z)} \leq 1, \qquad \sup_{\phi \in \mathcal{D}_g} \|\phi\|_{L^2(\hat{P}_N^Z)} \leq 1.$$

---

**Algorithm 1** Alternating Forward Selection

---

**Input.** Dictionaries $\mathcal{D}_f^\star, \mathcal{D}_g^\star$; initial values $(f_0, g_0) = (0,0)$, $S_0^f = S_0^g = \{0\}$; target $(Y,Y)$.

**for** $k = 1, 2, \dots$ **do**

    Compute the residual.
$$r_k \leftarrow (Y,Y) - (f_{k-1}, g_{k-1}).$$

*Step 1. $f$-selection and projection*

Select
$$\psi_k^\star = (\psi_k, 0) \in \underset{\substack{a \in \mathcal{D}_f^\star \\ \|\Pi_{S_{k-1}^f}^\perp a\|_\star > 0}}{\operatorname{argmax}} \frac{\langle r_k, \Pi_{S_{k-1}^f}^\perp a \rangle_\star}{\|\Pi_{S_{k-1}^f}^\perp a\|_\star}.$$

Update
$$S_k^f \leftarrow \operatorname{span}(\psi_1^\star, \dots, \psi_k^\star),$$
$$(f_k, g_{k-1}) \leftarrow (f_{k-1}, g_{k-1}) + \Pi_{S_k^f} r_k,$$

and set
$$r_k^{(g)} \leftarrow r_k - \Pi_{S_k^f} r_k.$$

*Step 2. $g$-selection and projection*

Select
$$\phi_k^\star = (0, \phi_k) \in \underset{\substack{b \in \mathcal{D}_g^\star \\ \|\Pi_{S_{k-1}^g}^\perp b\|_\star > 0}}{\operatorname{argmax}} \frac{\langle r_k^{(g)}, \Pi_{S_{k-1}^g}^\perp b \rangle_\star}{\|\Pi_{S_{k-1}^g}^\perp b\|_\star}.$$

Update
$$S_k^g \leftarrow \operatorname{span}(\phi_1^\star, \dots, \phi_k^\star),$$
$$(f_k, g_k) \leftarrow (f_k, g_{k-1}) + \Pi_{S_k^g} r_k^{(g)},$$

**end for**

---

In the following, we use canonical embeddings to construct the dictionary for the Hilbert space $\mathcal{H}^\star$ based on $\mathcal{D}_f$ and $\mathcal{D}_g$. Define embedded versions.
$$\mathcal{D}_f^\star := \{(\psi, 0) : \psi \in \mathcal{D}_f\}, \qquad \mathcal{D}_g^\star := \{(0, \phi) : \phi \in \mathcal{D}_g\}.$$

Assume the union $\mathcal{D}^\star := \mathcal{D}_f^\star \cup \mathcal{D}_g^\star$ is balanced, i.e., whenever $\zeta \in \mathcal{D}^\star$, we have $-\zeta \in \mathcal{D}^\star$. Since $\hat{P}_N^{(1)} + \hat{P}_N^{(2)} = \hat{P}_N^Z$, we have
$$\|(\psi, 0)\|_\star^2 = \|\psi\|_{L^2(\hat{P}_N^{(1)})}^2 + \|\psi\|_{L^2(\hat{P}_N^{(2)})}^2 = \|\psi\|_{L^2(\hat{P}_N^Z)}^2 \leq 1.$$

Moreover,
$$\|(0, \phi)\|_\star^2 = \|\phi\|_{L^2(\hat{P}_N^{(2)})}^2 + \|\phi\|_{L^2(\hat{P}_N^{(3)})}^2 = \frac{1}{N} \sum_{j=n+1}^N \phi(Z_j)^2 + \frac{\lambda}{N} \sum_{i=1}^n \phi(Z_i)^2 \leq \max\{1, \lambda\} \|\phi\|_{L^2(\hat{P}_N^Z)}^2 \leq \max\{1, \lambda\}.$$

Denote $R_\lambda := \max\{1, \sqrt{\lambda}\}$. Thus every embedded atom satisfies $\|a\|_\star \leq R_\lambda$. Throughout this subsection, $\Pi_S$ and $\Pi_S^\perp$ denote the orthogonal projections onto a subspace $S \subset \bar{\mathcal{K}}$ and its orthogonal complement, respectively, with respect to the inner product $\langle \cdot, \cdot \rangle_\star$.

## B. Implementation Details and Additional Experiments

### B.1. Parkinson's (Telemonitoring)

**Dataset.** We use the UCI Parkinson's Telemonitoring dataset to evaluate Coupled Training. The target is `total_UPDRS`, which typically requires in-person clinical assessment, so labeled data are limited. Each row is a single voice recording, and each `subject` may contribute multiple recordings. Since recording conditions at deployment are less controlled, we treat some acoustic descriptors as privileged features available only during training.

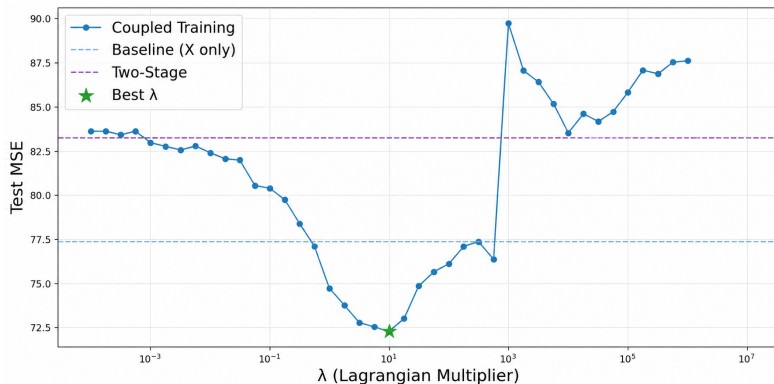

*Figure 3.* **Parkinson's dataset.** Test MSE versus $\lambda$.

**Feature split.** We partition covariates into deployment features $X$ and privileged features $W$ to model the fact that some high-fidelity acoustic descriptors are only reliable in controlled clinical conditions.

- **Deployment features** $X$ (10) are $\{$age, sex, test_time, Jitter(%), Jitter(Abs), Jitter:RAP, Jitter:PPQ5, Jitter:DDP, Shimmer, Shimmer(dB)$\}$.

- **Privileged features** $W$ (9) are $\{$Shimmer:APQ3, Shimmer:APQ5, Shimmer:APQ11, Shimmer:DDA, NHR, HNR, RPDE, DFA, PPE$\}$.

Here $W$ contains higher-fidelity acoustic descriptors that are reliable primarily in controlled clinical recordings, whereas $X$ contains features assumed available at deployment. We use $W$ only during training to improve the deployment predictor $\hat{f}(X)$.

**Data split.** To avoid leakage across repeated recordings from the same subject, we split the data at the subject level. We hold out 10% of subjects as a test set (seed=202). From the remaining subjects, we label 15% and treat the rest as unlabeled.

**Preprocessing.** All preprocessing is fit on the training set only. We standardize non-demographic voice features and apply PCA with 12 components (or fewer if limited by dimension), standardize demographic features (age and test time), and leave sex unchanged. We also standardize $Y$ using statistics computed from labeled training targets.

**Models and hyperparameters.** We use a random forest regressor with 1400 trees and maximum depth 12 (minimum leaf size 4); remaining settings are kept fixed across methods.

**Training procedure.** Baseline trains $f$ on labeled data only using $X \to Y$. Two-Stage trains a teacher $g$ on labeled data using $(X, W) \to Y$, pseudo-labels unlabeled data with $\tilde{Y}_U = g(X_U, W_U)$, then trains a student on $(X_L \cup X_U) \to (Y_L \cup \tilde{Y}_U)$. **Coupled Training** runs the coupled training updates for up to 15 iterations. For each $\lambda$ in a log-spaced grid $\{10^{-4}, \ldots, 10^6\}$ with 41 points, we initialize $f$ from the labeled baseline and initialize $g$ by fitting to a constant target $g_0 \equiv 0$. In this random-forest experiment we use the penalized Lagrangian form of the square-loss objective,

$$\sum_{i \in L}(Y_i - f(X_i))^2 + \sum_{j \in U}(g(X_j, W_j) - f(X_j))^2 + \lambda \sum_{i \in L}(Y_i - g(X_i, W_i))^2,$$

with all targets standardized using the labeled training responses. Given the current $g$, the $f$-update is a random-forest regression of the targets $(Y_L, g(X_U, W_U))$ on the deployment features $(X_L, X_U)$. Given the current $f$, the $g$-update is a weighted random-forest regression of the targets $(Y_L, f(X_U))$ on the rich-view features $((X_L, W_L), (X_U, W_U))$, using weight $\lambda$ for labeled samples and weight 1 for unlabeled samples. We employ early stopping based on stabilization of the disagreement on unlabeled data (patience 2). We then evaluate the resulting $\hat{f}(X)$ on the held-out test set.

**Evaluation.** All models are trained to predict scaled $Y$; at evaluation, we inversely transform predictions back to the original UPDRS scale, clip to $[0, 100]$, and report test mean squared error (MSE).
Figure 3 reports test MSE as a function of $\lambda$; performance is best at an intermediate value (around $10^1$).

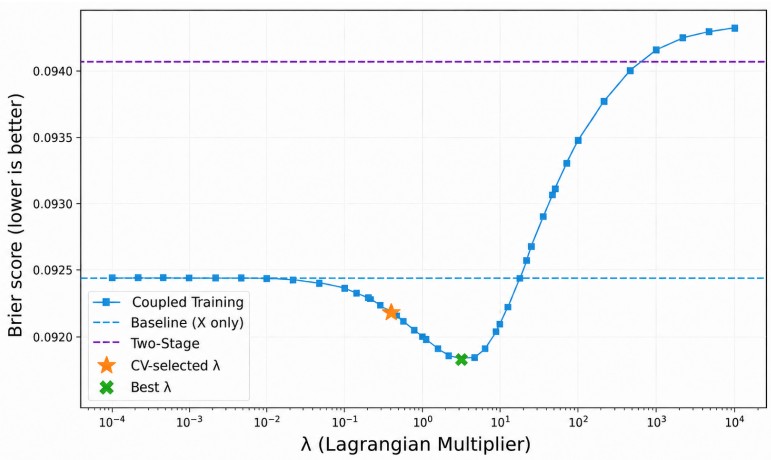

*Figure 4.* **Bank Marketing dataset.** Holdout Brier score versus $\lambda$.

**Cross-validation for $\lambda$.** We select $\hat{\lambda}$ using 5-fold cross-validation on the labeled set only, splitting by subject to prevent leakage (GROUPKFOLD). In each fold, we train Coupled Training using the fixed unlabeled pool $\mathcal{D}_U$ and evaluate the validation MSE of the deployment model $\hat{f}_\lambda$ on the held-out labeled fold. We take $\hat{\lambda} \in \arg\min_{\lambda \in \Lambda} \mathrm{MSE}_{\mathrm{val}}(\hat{f}_\lambda)$, where $\Lambda = \{10^{-4}, \dots, 10^6\}$ is a log-spaced grid with 41 points and $(\hat{f}_\lambda, \hat{g}_\lambda)$ denotes the models returned by Coupled Training when run with tuning parameter $\lambda$. The main-table result uses this CV-selected $\hat{\lambda}$; because the dataset has few independent subjects after subject-level splitting, we treat this as the primary diagnostic split rather than averaging a high-variance mean. As a robustness check over seeds $\{202, 42, 123, 999, 777, 2023, 2024, 888\}$, CV-selected Coupled Training beats both Baseline and Two-Stage in $6/8$ splits, with wins at seeds $\{202, 999, 777, 2023, 2024, 888\}$.

### B.2. Bank Marketing

**Dataset.** We evaluate Coupled Training on the UCI Bank Marketing dataset, a binary classification task where the goal is to predict whether a client subscribes to a term deposit. The dataset contains $41{,}188$ examples and has a positive class rate of approximately $0.113$. We report Brier score, so lower values are better.

**Feature split.** We use a train-time privileged information split motivated by the distinction between pre-deployment information and information that is tied to the current campaign call or campaign outcomes.

- **Deployment features** $X$ are $\{$age, job, marital, education, default, housing, loan, contact, month, day_of_week, emp.var.rate, cons.price.idx, cons.conf.idx, euribor3m, nr.employed$\}$.

- **Privileged features** $W$ are $\{$duration, pdays, previous, poutcome$\}$.

The final deployed predictor uses only $X$. The privileged view $W$ is used only during training.

**Data split.** For each outer seed in

$$\{2034, 2035, \dots, 2059\},$$

we use a stratified random $80/20$ train–test split. From the training pool, we sample $n = 200$ labeled examples stratified by class and $m = 10{,}000$ unlabeled examples from the remaining training examples. In each split, the labeled set contains 23 positives and 177 negatives.

**Preprocessing.** All preprocessing is fitted on the training pool only. Numeric features are standardized, categorical features are one-hot encoded with unknown categories ignored at test time, and the transformed design matrices are dense. The test set is never used for preprocessing, hyperparameter selection, or model fitting.

**Training-time corruption of privileged features.** To model noisy or imperfect train-time privileged information, we corrupt $W$ only on training rows. The fixed corruption preset is mix_plus_noise, where a fraction $0.35$ of the training rows has its privileged feature block row-mixed, then featurewise dropout with probability $0.10$ is applied, followed by Gaussian noise with standard deviation $0.15$. The same corruption protocol is used for Coupled Training, Two-Stage, and squared-loss generalized distillation whenever those methods use $W$.

**Coupled model and primary baselines.** The $X$-only Baseline, Two-Stage, squared-loss generalized distillation, and Coupled Training models all use linear squared-loss predictors with an unpenalized intercept. The fixed ridge parameters for Coupled Training and Two-Stage are $\alpha_f = 300$ for the deployment model and $\alpha_g = 0.01$ for the rich-view model. The agreement weight on unlabeled examples is 1, and the Two-Stage pseudo-label weight is also 1.

Let $\bar{X}$ denote $X$ augmented with an intercept and let $\bar{Z}$ denote $(X, W)$ augmented with an intercept. For each $\lambda$, the coupled model solves

$$\min_{\beta,\gamma} \sum_{i \in L} (Y_i - \bar{X}_i^\top \beta)^2 + \sum_{j \in U} (\bar{X}_j^\top \beta - \bar{Z}_j^\top \gamma)^2 + \lambda \sum_{i \in L} (Y_i - \bar{Z}_i^\top \gamma)^2 + \alpha_f \|\beta_{-0}\|_2^2 + \alpha_g \|\gamma_{-0}\|_2^2,$$

where the subscript $-0$ excludes the intercept.

**Cross-validation for $\lambda$.** We select $\lambda$ by fixed 4-fold stratified cross-validation on the 200 labeled examples only. The unlabeled examples are used for training inside each fold, but their labels are never used for selection. The grid is

$$\Lambda = \text{unique}\Big( \text{logspace}(-4, 4, 25) \cup \text{logspace}(-1, 2, 21)\Big).$$

After selecting $\hat{\lambda}$, we refit on the full labeled set and the same unlabeled pool, then evaluate the $X$-only deployment predictor on the held-out test set.

**Squared-loss generalized distillation.** We include a same-loss generalized-distillation baseline. The teacher is a ridge-regularized linear predictor trained on a privileged train-time view $Z$, where $Z$ is selected from either $W$ alone or $[X, W]$. For a teacher ridge parameter $\alpha_T$, the teacher solves

$$\hat{\gamma} \in \arg \min_{\gamma} \sum_{i \in L} (Y_i - \bar{Z}_i^\top \gamma)^2 + \alpha_T \|\gamma_{-0}\|_2^2.$$

It produces soft teacher targets

$$q_i = \bar{Z}_i^\top \hat{\gamma}.$$

The student is an $X$-only ridge model trained by soft squared-loss distillation.

$$\min_{\beta} \sum_{i \in L} (Y_i - \bar{X}_i^\top \beta)^2 + a_L \sum_{i \in L} (q_i - \bar{X}_i^\top \beta)^2 + a_U \sum_{j \in U} (q_j - \bar{X}_j^\top \beta)^2 + \alpha_S \|\beta_{-0}\|_2^2.$$

There is no hard pseudo-label term. We select the teacher view, teacher ridge parameter, student ridge parameter, and soft-distillation weights by the same fixed 4-fold labeled-only CV protocol used for the coupled model. The narrowed search grid is

$$Z \in \{W, [X, W]\}, \qquad \alpha_T \in \{0.003, 0.01, 0.03\}, \qquad \alpha_S \in \{90, 300, 900\},$$

and

$$(a_L, a_U) \in \{(0, 0.5), (0, 1), (0.25, 0.5), (0.25, 1), (0.5, 0.5), (0.5, 1)\}.$$

This grid contains the ordinary squared-loss Two-Stage baseline as the special case

$$Z = [X, W], \qquad \alpha_T = \alpha_g = 0.01, \qquad \alpha_S = \alpha_f = 300, \qquad (a_L, a_U) = (0, 1).$$

**Vapnik SVM+.** We also compare against Vapnik SVM+, a labeled-only privileged information baseline. SVM+ uses $W$ only for the labeled training examples and predicts with $X$ only at test time. Since classical SVM+ is not a semi-supervised method, it does not use the unlabeled pool for learning. We tune

$$C \in \{0.1, 1, 10\}, \qquad \gamma \in \{0.1, 1, 10\}$$

by the same fixed 4-fold labeled-only CV protocol, then refit on all 200 labeled examples. We use the classical SVM+ limit $\Delta = \infty$ and Platt calibration with calibration parameter $C = 1$ to obtain probabilities for Brier-score evaluation.

**Evaluation.** For all methods, the final test-time predictor is an $X$-only predictor. We clip predicted probabilities to $[10^{-6}, 1 - 10^{-6}]$ and report holdout Brier score. All model choices are made without using holdout labels.

*Table 2.* **Bank Marketing detailed comparison.** Means and confidence intervals are computed across 26 outer seeds. "Coupled gain" is the mean Brier-score improvement of the CV-selected coupled model over the corresponding method; positive values favor Coupled Training.

| Method | Uses unlabeled? | Mean Brier | 95% CI | Coupled gain |
|---|---|---|---|---|
| Baseline ($X$ only) | No | 0.0893 | [0.0888, 0.0897] | 0.0011 |
| Two-Stage | Yes | 0.0928 | [0.0918, 0.0939] | 0.0047 |
| Gen. Distill. (squared) | Yes | 0.0907 | [0.0894, 0.0921] | 0.0026 |
| Vapnik SVM+ | No | 0.1470 | [0.1221, 0.1720] | 0.0589 |
| Coupled, CV-selected $\lambda$ | Yes | **0.0881** | [0.0874, 0.0889] | – |

**Results.** Table 2 reports the 26-seed comparison. The CV-selected coupled model obtains mean holdout Brier score 0.0881, improving over the $X$-only Baseline, Two-Stage, squared-loss generalized distillation, and Vapnik SVM+. Squared-loss generalized distillation improves over Two-Stage, with mean Brier score 0.0907 compared with 0.0928, but remains worse than Coupled Training. The mean Brier-score gain of Coupled Training is 0.0011 against the Baseline, 0.0047 against Two-Stage, 0.0026 against squared-loss generalized distillation, and 0.0589 against Vapnik SVM+. Coupled Training beats the Baseline in $22/26$ seeds, Two-Stage in $26/26$ seeds, squared-loss generalized distillation in $22/26$ seeds, and Vapnik SVM+ in $26/26$ seeds. The mean regret of the CV-selected $\hat{\lambda}$ relative to the per-seed holdout-best $\lambda$ is 0.00125 Brier score, with median regret 0.00050.

### B.3. PneumoniaMNIST Experiment for Algorithm 1

**Dataset.** We evaluate Algorithm 1 on PneumoniaMNIST, a binary classification task with $28 \times 28$ grayscale images. We use the flattened image as deployment features $X \in \mathbb{R}^{784}$ and construct privileged features $W \in \mathbb{R}^{18}$ available only during training, with the first coordinate of $W$ a noisy version of $Y$ and the remaining ones random linear projections of $X$. The label $Y \in \{0, 1\}$ indicates pneumonia.

**Feature split.** Let $X \in \mathbb{R}^{N \times 784}$ be pixel features scaled to $[0, 1]$. We generate $W \in \mathbb{R}^{N \times 18}$ by (i) $W_{:,1} = Y + \varepsilon$ with i.i.d. $\varepsilon \sim \mathcal{N}(0, 0.3^2)$, and (ii) $W_{:,2:18} = XP$ where $P \in \mathbb{R}^{784 \times 17}$ has i.i.d. entries $\mathcal{N}(0, 0.1^2)$ (fixed given the seed).

**Data split.** We use a single stratified split with $n = 100$ labeled points, $m = 2000$ unlabeled points, and $n_{\text{test}} = 1500$ test points (`seed=42`).

**Preprocessing.** For all linear/kernel methods, we standardize both $X$ and $W$ using `StandardScaler` fit on labeled∪unlabeled data only (the test set is never used for fitting). We also feed standardized $W$ to teacher models for stable optimization.

**Models and hyperparameters.** We run Algorithm 1 in $\mathcal{H}^\star = \mathcal{H}_f \times \mathcal{H}_g$ (Section 3.2) with $K = 70$ iterations, squared loss, and evaluation by test AUROC using the deployment score $f(X)$. We sweep $\lambda$ over 10 log-spaced values from $10^{-2}$ to $10^2$. We consider the following options.

- **Linear AFS (random-feature dictionary).** Embedded dictionaries use fixed Gaussian random projections with sizes `dict_size_f = dict_size_g = 2048` for $f$ and $g$ respectively.

- **Kernelized AFS (RBF kernel dictionary).** Centers are all labeled points plus up to 500 randomly chosen unlabeled points. Kernel bandwidths $\gamma_f$ (for $X$) and $\gamma_g$ (for $(X, W)$) are initialized by the median heuristic (computed on up to 600 samples), and tuned by stratified 5-fold CV on labeled data, maximizing AUROC over $\gamma \in \gamma_0 \cdot \{0.25, 0.5, 1, 2, 4\}$ and $\alpha \in \{10^{-3}, 10^{-2}, 10^{-1}, 1, 10\}$.

**Refit.** After greedy selection, we refit the deployment predictor restricted to the selected dictionary elements by ridge regression with $\alpha_{\text{refit}} = 10^{-3}$. This post-selection refit is a standard stabilization step that reduces shrinkage from the greedy stage. We obtain pseudo-targets $\tilde{Y}_U = g(X_U, W_U)$ from a teacher $g$ trained on labeled data using $(X, W) \to Y$ (ridge in the linear case; kernel ridge in the kernel case with tuned $\alpha_g$), and refit on $(X_L \cup X_U) \to (Y_L \cup \tilde{Y}_U)$.

**Training procedure.** Baseline trains $f$ on labeled data only using $X \to Y$ (ridge for linear; kernel ridge with tuned $(\gamma_f, \alpha_f)$ for kernel). Two-Stage trains a teacher on labeled data using $(X, W) \to Y$, pseudo-labels unlabeled data, then trains a student on $(X_L \cup X_U) \to (Y_L \cup \tilde{Y}_U)$ using the same model class. For reference, we also report a CNN baseline (image-only student) and a CNN Two-Stage teacher–student pipeline where the teacher fuses image features with standardized $W$, pseudo-labels unlabeled data (threshold 0.5), and retrains the image-only student; we use Adam with learning rate $10^{-3}$,

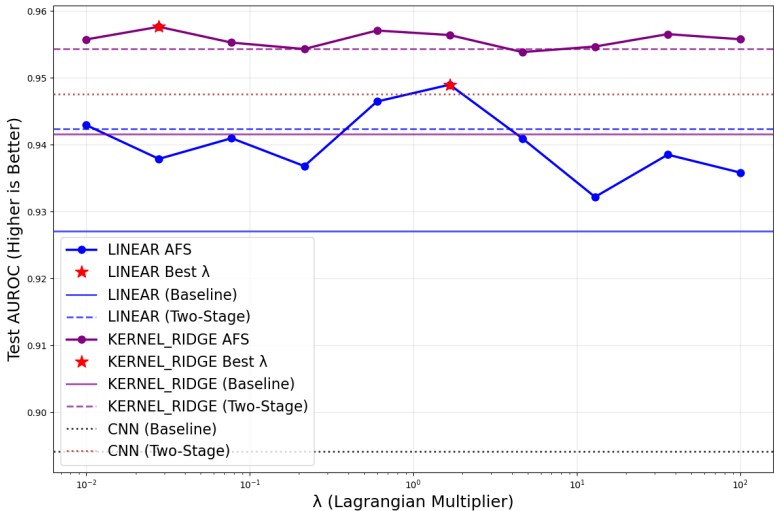

*Figure 5.* **PneumoniaMNIST.** Test AUROC versus $\lambda$ for Algorithm 1.

batch size 32, and 30 epochs.

**Evaluation.** Models output real-valued scores; we apply a sigmoid to obtain probabilities and report test AUROC (primary), along with accuracy at threshold $0.5$ and probability MSE against $\{0, 1\}$ targets.

**Results.** Figure 5 shows test AUROC as a function of $\lambda$. Coupled Training with AFS performs better than the Baseline and Two-Stage methods for many $\lambda$ values, with the strongest results at intermediate $\lambda$. The effect is especially clear for linear models. For a subset of $\lambda$ choices, linear AFS even outperforms both CNN Baseline and Two-Stage.

### B.4. Controlled Synthetic Linear Experiments

**Dataset.** We generate controlled synthetic data to isolate three effects that are central to the behavior of Coupled Training, namely the strength of the privileged signal, the amount of nuisance variation in the privileged view, and the number of unlabeled samples. The data-generating process separates the deployment signal from the privileged signal. Specifically, we sample

$$X \sim \mathcal{N}(0, I_{d_X}), \qquad H \sim \mathcal{N}(0, I_q), \qquad V \sim \mathcal{N}(0, I_{d_{\text{noise}}}),$$

where $H$ contains the informative latent privileged factors and $V$ contains nuisance privileged coordinates. We then form

$$W_{\text{sig}} = \rho_{XW} X A + \sqrt{1 - \rho_{XW}^2}\, H, \qquad W = (W_{\text{sig}}, V),$$

where the columns of $A$ are normalized. The response is generated as

$$Y = X^\top \beta + \alpha H^\top \theta + \varepsilon, \qquad \varepsilon \sim \mathcal{N}(0, \sigma^2).$$

Here $\alpha$ controls the strength of the privileged signal. Since $H$ is not available at deployment, the deployment target is

$$\mu(X) = X^\top \beta.$$

Thus, unlike ordinary test MSE against noisy labels, this construction lets us evaluate the theory-aligned estimation error of the learned deployment predictor.

**Feature split.** The deployment view is $X \in \mathbb{R}^{d_X}$ and the privileged view is $W \in \mathbb{R}^{q+d_{\text{noise}}}$. The first $q$ coordinates of $W$ are correlated with the latent privileged signal, while the remaining $d_{\text{noise}}$ coordinates are nuisance dimensions. This design allows us to test whether the rich-view model helps when the privileged signal is informative, and whether it becomes harmful when the privileged view contains many irrelevant coordinates.

**Data split.** For each random seed, we independently sample labeled, unlabeled, and test sets from the same ground-truth model. Unless otherwise stated, we use $d_X = 10$, $q = 3$, $\rho_{XW} = 0.7$, and $\sigma = 1$. The signal-strength and nuisance-dimension sweeps use $n = 100$ labeled samples, $m = 20,000$ unlabeled samples, and $n_{\text{test}} = 10,000$ test samples. The unlabeled-sample-size sweep uses a smaller labeled budget, $n = 40$, while varying $m$.

**Training procedure.** All methods use linear predictors with an intercept. Baseline fits the deployment model $f$ using only labeled pairs $(X, Y)$. Two-Stage first fits a rich-view teacher $g$ using labeled pairs $((X, W), Y)$, pseudo-labels the unlabeled examples, and then fits an $X$-only student on the union of labeled and pseudo-labeled examples. Coupled Training solves the linear penalized objective for each value of $\lambda$ in a logarithmic grid. For numerical stability, we use a small ridge penalty in all linear solves.

**Hyperparameter selection.** For the summary sweeps in the main text, $\lambda$ is selected by cross-validation on the labeled set only. The unlabeled samples are used during training but their labels are never used for selecting $\lambda$. We sweep $\lambda$ over a log-spaced grid from $10^{-4}$ to $10^4$.

**Evaluation.** For the controlled sweeps, we report the theory-aligned test estimation error

$$\mathbb{E}_{\text{test}}\big[(\hat{f}(X) - \mu(X))^2\big],$$

where $\mu(X) = X^\top \beta$ is known from the data-generating process. This removes irreducible label noise from the evaluation and directly measures how well the deployment predictor estimates the target regression function. We average results over multiple random seeds and report mean curves.

**Controlled sweeps.** The signal-strength sweep varies $\alpha$ while holding the nuisance dimension fixed. The nuisance-dimension sweep varies $d_{\text{noise}}$ while holding the privileged signal strength fixed. The unlabeled-sample-size sweep varies $m$ while keeping the labeled budget fixed. These three sweeps correspond respectively to the three panels in Figure 2, showing that Coupled Training improves as the privileged signal becomes useful, remains more stable than Two-Stage when nuisance privileged dimensions grow, and benefits from additional unlabeled paired $(X, W)$ samples.

## B.5. Synthetic Binary Classification Diagnostic

**Dataset.** We also run a controlled binary classification diagnostic to test the cross-entropy analogue of Coupled Training. The synthetic generator produces deployment features $X \in \mathbb{R}^5$ and privileged features $W \in \mathbb{R}^{40}$, with correlated deployment and privileged views. Labels are sampled from a Bernoulli model whose logit depends on both the deployment signal and the privileged signal, with additive logit noise. The default setting uses correlation strength 0.95, $X$ scale 1.0, $W$ scale 1.05, logit-noise standard deviation 0.70, and unlabeled mean parameter 1.0.

**Feature split.** The deployment view is $X \in \mathbb{R}^5$ and the privileged view is $W \in \mathbb{R}^{40}$. The final predictor must use only $X$ at test time. The privileged block $W$ is available only during training and is intentionally higher-dimensional than $X$, so the experiment tests whether Coupled Training can benefit from privileged information without fully inheriting rich-view noise.

**Data split.** For each random seed, we generate independent labeled, unlabeled, and test samples. We use

$$n = 50, \qquad m = 3000, \qquad n_{\text{test}} = 6000,$$

and average results over seeds $\{0, 1, 2, 3, 4\}$. The labels of the unlabeled samples are discarded during training and are used only by the data generator.

**Models and hyperparameters.** All methods use linear logistic predictors with intercepts. Let

$$p_f(x) = \sigma(\bar{x}^\top \beta), \qquad p_g(x, w) = \sigma(\bar{z}^\top \gamma),$$

where $\bar{x}$ and $\bar{z}$ include intercepts and $\sigma(t) = (1 + e^{-t})^{-1}$. The deployment model uses ridge parameter $\alpha_f = 10^{-4}$, and the rich-view model uses ridge parameter $\alpha_g = 10^{-1}$. The $X$-only baseline is trained for 500 gradient steps with learning rate 0.05. The rich-view teacher and the Two-Stage student are each trained for 700 gradient steps with learning rate 0.03.

**Training procedure.** Baseline trains the deployment model using only labeled pairs $(X_L, Y_L)$. Two-Stage first trains a rich-view teacher using labeled pairs $((X_L, W_L), Y_L)$, pseudo-labels the unlabeled examples with teacher probabilities $p_g(X_U, W_U)$, and then trains an $X$-only student on the union of labeled and pseudo-labeled examples.
For Coupled Training, we use the cross-entropy analogue of the alternating updates. For $a \in [0, 1]$ and $p \in (0, 1)$, define

$$\text{CE}(a, p) = -a \log p - (1 - a) \log(1 - p).$$

Given $g$, the deployment update minimizes

$$\sum_{i \in L} \text{CE}\big(Y_i, p_f(X_i)\big) + \sum_{j \in U} \text{CE}\big(p_g(X_j, W_j), p_f(X_j)\big) + \frac{\alpha_f}{2} \|\beta_{-0}\|_2^2,$$

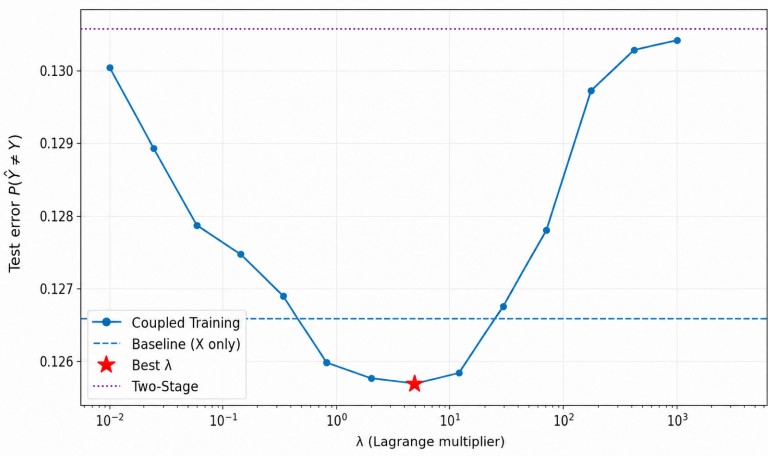

*Figure 6.* **Synthetic binary classification diagnostic.** Test 0–1 error for the cross-entropy analogue of Coupled Training as a function of $\lambda$, averaged over seeds $\{0, 1, 2, 3, 4\}$.

where the subscript $-0$ excludes the intercept. Given $f$, the rich-view update minimizes

$$\sum_{j \in U} \mathrm{CE}\big(p_f(X_j), p_g(X_j, W_j)\big) + \lambda \sum_{i \in L} \mathrm{CE}\big(Y_i, p_g(X_i, W_i)\big) + \frac{\alpha_g}{2} \|\gamma_{-0}\|_2^2 .$$

We run 5 outer coupled iterations. Each $f$-update and each $g$-update uses 150 gradient steps with learning rate 0.02.

**Hyperparameter sweep.** For this diagnostic, $\lambda$ is swept rather than selected by cross-validation. We use 14 log-spaced values from $10^{-2}$ to $10^3$. Thus, the best point in the figure should be interpreted as the best point in the displayed diagnostic sweep, not as a validation-selected tuning parameter.

**Evaluation.** The final test-time predictor is the deployment model $p_f(X)$, which uses only $X$. We classify by thresholding at $1/2$ and report test error

$$\mathbb{P}_{\text{test}}\big(\mathbf{1}\{p_f(X) \geq 1/2\} \neq Y\big).$$

All reported curves are averaged over the five random seeds.

**Results.** Figure 6 shows that the cross-entropy coupled model has an interior optimum in $\lambda$. The best coupled model occurs around $\lambda \approx 5$ and improves slightly over the labeled $X$-only baseline, while the Two-Stage limit performs worse. This mirrors the regression experiments, where moderate coupling can use privileged information, whereas overly strong transfer can propagate rich-view noise into the deployment model.

# C. Proofs

In what follows, let $a = \frac{n}{N}$, $b = \frac{m}{N}$, and $c = b + a\lambda = \frac{m+n\lambda}{N}$, and define

$$s_f(z) := \frac{b\, f(x) + a\lambda\, \eta(z)}{c}, \qquad t_g(x) := a\, \mu(x) + b\, \mathbb{E}[g(Z) \mid X = x].$$

For functions $f \in \mathcal{F}$ and $g \in \mathcal{G}$, we define the (squared) $L^2$ norms $\|f\|_X^2 = \mathbb{E}[f^2(X)]$ and $\|g\|_Z^2 = \mathbb{E}[g^2(Z)]$. Define the inner products $\langle \cdot, \cdot \rangle_X$ and $\langle \cdot, \cdot \rangle_Z$ analogously.

For fixed points $s^n = (s_1, \ldots, s_n)$ in a measurable space $\mathcal{S}$, let $\hat{P}_n$ denote the empirical measure with respect to these points. For a function $h : \mathcal{S} \to \mathbb{R}$, we write $\|h\|_{L^p(\hat{P}_n)} = \left( \frac{1}{n} \sum_{i=1}^n |h(s_i)|^p \right)^{1/p}$. An $L^p$-$\epsilon$-cover of a class $\mathcal{C}$ with respect to $\|\cdot\|_{L^p(\hat{P}_n)}$ is a finite collection $\{h_1, \ldots, h_M\} \subset \mathcal{C}$ such that every $h \in \mathcal{C}$ is within $\varepsilon$ of some $h_j$ under this norm, and the minimal such $M$ is the empirical covering number $\mathcal{N}_p(\epsilon, \mathcal{C}, s^n)$.

## C.1. Proof of Theorem 2.1

For a fixed $f \in \mathcal{F}$, let $d$ denote constants that do not depend on $g$ (and may change from line to line). Then, we can write the population loss as follows

$$
\begin{aligned}
\mathcal{L}(f, g; \lambda) &= b\mathbb{E}[(g(Z) - f(X))^2] + a\lambda\mathbb{E}[(Y - g(Z))^2] + d \\
&= b\left(\mathbb{E}g^2(Z) - 2\mathbb{E}g(Z)f(X) + \mathbb{E}f^2(X)\right) + a\lambda\left(\mathbb{E}Y^2 - 2\mathbb{E}[Yg(Z)] + \mathbb{E}g^2(Z)\right) + d \\
&= c\mathbb{E}g^2(Z) - 2\mathbb{E}[g(Z)(bf(X) + a\lambda Y)] + d \\
&= c\mathbb{E}g^2(Z) - 2\mathbb{E}\left[g(Z)\mathbb{E}[(bf(X) + a\lambda Y) \mid Z]\right] + d \\
&= c\left(\mathbb{E}g^2(Z) - 2\mathbb{E}[g(Z)s_f(Z)]\right) + d \\
&= c\left(\mathbb{E}[g^2(Z)] - 2\mathbb{E}[g(Z)s_f(Z)] + \mathbb{E}[s_f^2(Z)]\right) + d && (5) \\
&= c\|g - s_f\|_Z^2 + d. && (6)
\end{aligned}
$$

Thus, for fixed $f \in \mathcal{F}$, the best response over $\mathcal{G}$ is the metric projection

$$g_{f,\lambda}^{\mathcal{G}} = \operatorname*{argmin}_{g \in \mathcal{G}} \mathcal{L}(f, g; \lambda) = \operatorname*{argmin}_{g \in \mathcal{G}} \|g - s_f\|_Z^2 = \Pi_{\mathcal{G}} s_f \tag{7}$$

.

For a fixed $g \in \mathcal{G}$, let $d$ denote constants that do not depend on $f$ (and may change from line to line). Then using same approach to derive (6), we get

$$
\begin{aligned}
\mathcal{L}(f, g; \lambda) &= a\mathbb{E}[(Y - f(X))^2] + b\mathbb{E}[(g(Z) - f(X))^2] + d \\
&= (a + b)\mathbb{E}f^2(X) - 2\mathbb{E}\left[f(X)\mathbb{E}[(bg(Z) + aY) \mid X]\right] + d \\
&= \|f - t_g\|_X^2 + d. && (8)
\end{aligned}
$$

Thus, for fixed $g \in \mathcal{G}$, the best response over $\mathcal{F}$ is

$$f_g^{\mathcal{F}} = \operatorname*{argmin}_{f \in \mathcal{F}} \mathcal{L}(f, g; \lambda) = \operatorname*{argmin}_{f \in \mathcal{F}} \|f - t_g\|_X^2 = \Pi_{\mathcal{F}} t_g. \tag{9}$$

Hence, the desired result follows from (7) and (9).

Furthermore, if $\mu \in \mathcal{F}$ and $\mu \in \mathcal{G}$ (identifying $\mu$ with its lift $\tilde{\mu}(x, w) = \mu(x)$ on $\mathcal{Z}$), and $\eta \in \mathcal{G}$, then

$$
\begin{aligned}
f^\star(x) &= \Pi_{\mathcal{F}}\left( a\mu(x) + b\mathbb{E}\left[ \Pi_{\mathcal{G}} \frac{bf^\star(X) + a\lambda\eta(Z)}{c} \;\Big|\; X = x \right] \right) \\
&= \Pi_{\mathcal{F}}\left( a\mu(x) + b\mathbb{E}\left[ \frac{b\Pi_{\mathcal{G}} f^\star(X) + a\lambda\Pi_{\mathcal{G}}\eta(Z)}{c} \;\Big|\; X = x \right] \right) \\
&= \Pi_{\mathcal{F}}\left( a\mu(x) + b\left( \frac{b}{c}\mathbb{E}\left[ \Pi_{\mathcal{G}} f^\star(X) \mid X = x \right] + \frac{a\lambda}{c}\mu(x) \right) \right).
\end{aligned}
$$

It is clear that $f^\star = \mu$ satisfies this equation, which means that it is a global minimizer, and thus, $g^\star = \frac{m}{m+n\lambda}\mu + \frac{n\lambda}{m+n\lambda}\eta$.

Next, before proving Theorem 2.7, we present a sequence of intermediate results that will be useful for its proof.

**Theorem C.1.** *The following identity decomposes the excess population loss around $(f^\star, g^\star)$.*

$$\mathcal{L}(\hat{f}, \hat{g}; \lambda) - \mathcal{L}(f^\star, g^\star; \lambda) = \|u\|_X^2 + c\|v\|_Z^2 - 2b\langle u, v\rangle_Z$$
$$+ \langle \nabla_f \mathcal{L}(f^\star, g^\star; \lambda), u\rangle_X + \langle \nabla_g \mathcal{L}(f^\star, g^\star; \lambda), v\rangle_Z, \tag{10}$$

*where*

$$u := \hat{f} - f^\star \in \mathcal{F} - f^\star, \qquad v := \hat{g} - g^\star \in \mathcal{G} - g^\star.$$

*Proof.* We start by decomposing the loss $\mathcal{L}(\hat{f}, \hat{g}; \lambda)$ into three terms.

$$\mathcal{L}(\hat{f}, \hat{g}; \lambda) = \text{①} + \text{②} + \text{③},$$

where

$$\text{①} = a\mathbb{E}[(Y - \hat{f}(X))^2], \quad \text{②} = b\mathbb{E}[(\hat{g}(Z) - \hat{f}(X))^2], \quad \text{and} \quad \text{③} = a\lambda\mathbb{E}[(Y - \hat{g}(Z))^2].$$

Writing $\hat{f} = f^\star + u$ and $\hat{g} = g^\star + v$, we can expand each term as follows.

$$\text{①} = a\left(\mathbb{E}[(Y - f^\star(X))^2] + \mathbb{E}u^2(X) - 2\mathbb{E}[(Y - f^\star(X))u(X)]\right)$$
$$= a\mathbb{E}[(Y - f^\star(X))^2] + a\|u\|_X^2 - 2\mathbb{E}[u(X)(\mathbb{E}(Y \mid X) - f^\star(X))]$$
$$= a\mathbb{E}[(Y - f^\star(X))^2] + a\|u\|_X^2 - 2a\langle \mu - f^\star, u\rangle_X.$$

$$\text{②} = b\left(\mathbb{E}[(g^\star(Z) - f^\star(X))^2] + \mathbb{E}[(v(Z) - u(X))^2] + 2\mathbb{E}[(g^\star(Z) - f^\star(X))(v(Z) - u(X))]\right)$$
$$= b\mathbb{E}[(g^\star(Z) - f^\star(X))^2] + b\|v\|_Z^2 + b\|u\|_X^2 - 2b\langle u, v\rangle_Z$$
$$+ 2b\langle g^\star - f^\star, v\rangle_Z - 2b\langle \tilde{g}^\star - f^\star, u\rangle_X,$$

where $\tilde{g}^\star(x) = \mathbb{E}[g^\star(Z) \mid X = x]$.

$$\text{③} = a\lambda\mathbb{E}[(Y - g^\star(Z))^2] + a\lambda\|v\|_Z^2 - 2a\lambda\mathbb{E}[(Y - g^\star(Z))v(Z)]$$
$$= a\lambda\mathbb{E}[(Y - g^\star(Z))^2] + a\lambda\|v\|_Z^2 - 2a\lambda\langle \eta - g^\star, v\rangle_Z.$$

Using these expansions, observe that

$$\mathcal{L}(\hat{f}, \hat{g}; \lambda) - \mathcal{L}(f^\star, g^\star; \lambda) = (a + b)\|u\|_X^2 + c\|v\|_Z^2 - 2b\langle u, v\rangle_Z$$
$$+ \langle 2(f^\star - b\tilde{g}^\star - a\mu), u\rangle_X + \langle 2(cg^\star - bf^\star - a\lambda\eta), v\rangle_Z$$
$$= \|u\|_X^2 + c\|v\|_Z^2 - 2b\langle u, v\rangle_Z$$
$$+ \langle 2(f^\star - t_{g^\star}), u\rangle_X + \langle 2c(g^\star - s_{f^\star}), v\rangle_Z$$
$$= \|u\|_X^2 + c\|v\|_Z^2 - 2b\langle u, v\rangle_Z$$
$$+ \langle \nabla_f \mathcal{L}(f^\star, g^\star; \lambda), u\rangle_X + \langle \nabla_g \mathcal{L}(f^\star, g^\star; \lambda), v\rangle_Z,$$

where the last equality follows from (6) and (8), which give

$$\nabla_g \mathcal{L}(f, g; \lambda) = 2c(g - s_f), \qquad \nabla_f \mathcal{L}(f, g; \lambda) = 2(f - t_g).$$

$\square$

## C.2. Proof of Corollary 2.3

Since $(f^\star, g^\star)$ minimizes $\mathcal{L}$ over the closed convex set $\mathcal{F} \times \mathcal{G}$, the variational inequalities give

$$\langle \nabla_f \mathcal{L}(f^\star, g^\star; \lambda), u\rangle_X \geq 0, \qquad \langle \nabla_g \mathcal{L}(f^\star, g^\star; \lambda), v\rangle_Z \geq 0$$

for all $u \in \mathcal{F} - f^\star, v \in \mathcal{G} - g^\star.$

Thus, dropping the nonnegative linear terms in (10) yields

$$\mathcal{L}(\hat{f}, \hat{g}; \lambda) - \mathcal{L}(f^\star, g^\star; \lambda) \geq \|u\|_X^2 + c\|v\|_Z^2 - 2b\langle u, v\rangle_Z. \tag{11}$$

The result follows by taking expectation with respect to $\mathscr{D}$ of both sides (11), then we can further lower bound the right hand by completing the square of the last two terms as done below.

$$\mathbb{E}_{\mathscr{D}} \|u\|_X^2 + c\mathbb{E}_{\mathscr{D}} \|v\|_Z^2 - 2b\mathbb{E}_{\mathscr{D}} \langle u, v\rangle_Z$$
$$\geq \mathbb{E}_{\mathscr{D}} \|u\|_X^2 + c\mathbb{E}_{\mathscr{D}} \|v\|_Z^2 - 2b |\mathbb{E}_{\mathscr{D}} \langle u, v\rangle_Z|$$
$$= \mathbb{E}_{\mathscr{D}} \|u\|_X^2 + c\mathbb{E}_{\mathscr{D}} \|v\|_Z^2 - 2b\rho_\star \sqrt{\mathbb{E}_{\mathscr{D}} \|u\|_X^2 \, \mathbb{E}_{\mathscr{D}} \|v\|_Z^2}$$
$$= \mathbb{E}_{\mathscr{D}} \|u\|_X^2 + c\left(\mathbb{E}_{\mathscr{D}} \|v\|_Z^2 - \frac{2b\rho_\star}{c}\sqrt{\mathbb{E}_{\mathscr{D}} \|u\|_X^2 \, \mathbb{E}_{\mathscr{D}} \|v\|_Z^2} + \frac{b^2\rho_\star^2}{c^2}\mathbb{E}_{\mathscr{D}} \|u\|_X^2\right) - \frac{b^2\rho_\star^2}{c}\mathbb{E}_{\mathscr{D}} \|u\|_X^2$$
$$= \gamma_{n,m,\lambda}(\rho_\star)\mathbb{E}_{\mathscr{D}} \|u\|_X^2 + c\left(\sqrt{\mathbb{E}_{\mathscr{D}} \|v\|_Z^2} - \frac{b\rho_\star}{c}\sqrt{\mathbb{E}_{\mathscr{D}} \|u\|_X^2}\right)^2$$
$$\geq \gamma_{n,m,\lambda}(\rho_\star)\mathbb{E}_{\mathscr{D}} \|u\|_X^2 .$$

Throughout, we use uppercase letters (e.g., $X_i, Z_i, Y_i$) for random variables and lowercase letters (e.g., $x_i, z_i, y_i$) for their realizations. We write $x^n = (x_1, \ldots, x_n)$ and $z^n = (z_1, \ldots, z_n)$ for the labeled sample locations, and $x^m = (x_{n+1}, \ldots, x_N)$ and $z^m = (z_{n+1}, \ldots, z_N)$ for the unlabeled ones (where $m = N - n$). We also use $x^N = (x_1, \ldots, x_N)$ and $z^N = (z_1, \ldots, z_N)$ to denote the full collection of sample points. Define

$$k_i(f, g) := (f(x_i) - y_i)^2 + \lambda(g(z_i) - y_i)^2 - \left[(f^\star(x_i) - y_i)^2 + \lambda(g^\star(z_i) - y_i)^2\right],$$
$$p_j(f, g) := (f(x_j) - g(z_j))^2 - (f^\star(x_j) - g^\star(z_j))^2.$$

For notational convenience, throughout the remainder of the text we write $\sum_i$ for $\sum_{i=1}^n$ and $\sum_j$ for $\sum_{j=n+1}^N$. We now state a result adapted from Györfi et al. (2002, Theorem 11.4).

**Theorem C.2.** *Assume that 2.5 holds.*
*Then*

$$\mathbb{P}\left\{\exists(f, g) \in \mathcal{F} \times \mathcal{G} : \mathbb{E}[\hat{\mathcal{L}}(f, g; \lambda)] - \mathbb{E}[\hat{\mathcal{L}}(f^\star, g^\star; \lambda)] - \frac{1}{N}\left(\sum_i k_i + \sum_j p_j\right)\right.$$

$$\left. \geq \epsilon\left(\alpha + \beta + \mathbb{E}[\hat{\mathcal{L}}(f, g; \lambda)] - \mathbb{E}[\hat{\mathcal{L}}(f^\star, g^\star; \lambda)]\right)\right\}$$

$$\leq 10\mathfrak{M}_{\mathcal{F},\mathcal{G}}^{n,m,\lambda} \exp\left(-\frac{\epsilon^2(1-\epsilon)\alpha N}{512B^2(\lambda+1)}\right),$$

*where* $\alpha, \beta > 0, 0 < \epsilon < 1,$ *and*

$$\mathfrak{M}_{\mathcal{F},\mathcal{G}}^{n,m,\lambda} := \max_{z^n} \mathcal{N}_1\left(\frac{(\alpha+\beta)}{80B^2\sqrt{\lambda+1}a}, \mathcal{F}, x^n\right) \cdot \mathcal{N}_1\left(\frac{(\alpha+\beta)}{80B^2 a(\lambda+1)^{3/2}}, \mathcal{G}, z^n\right) +$$

$$\max_{z^m} \mathcal{N}_1\left(\frac{(\alpha+\beta)}{80B^2\sqrt{\lambda+1}b}, \mathcal{F}, x^m\right) \cdot \mathcal{N}_1\left(\frac{(\alpha+\beta)}{80B^2\sqrt{\lambda+1}b}, \mathcal{G}, z^m\right) +$$

$$\max_{z^N} \mathcal{N}_1\left(\frac{\epsilon\beta}{40B\sqrt{\lambda+1}}, \mathcal{F}, x^N\right) \cdot \mathcal{N}_1\left(\frac{\epsilon\beta}{40(\lambda+1)^{3/2}B}, \mathcal{G}, z^N\right).$$

*Proof.* Throughout this proof, for $(f, g) \in \mathcal{F} \times \mathcal{G}$, we interpret $f$ and $g$ pointwise, writing $f \equiv f(X), g \equiv g(Z)$, so that expressions such as $(f - Y)^2, (g - Y)^2,$ and $(f - g)^2$ are understood as $(f(X) - Y)^2, (g(Z) - Y)^2,$ and $(f(X) - g(Z))^2$, respectively.

For a sample $(X, Z, Y)$, define the $*$-norm

$$\|(f, g) - (Y, Y)\|_*^2 := a(f - Y)^2 + a\lambda(g - Y)^2 + b(f - g)^2.$$

Denote

$$A_{f,g} := \|(f, g) - (Y, Y)\|_*^2 - \|(f^\star, g^\star) - (Y, Y)\|_*^2 = a\, k(f, g) + b\, p(f, g),$$

where $k(f, g) = (f - Y)^2 + \lambda(g - Y)^2 - [(f^\star - Y)^2 + \lambda(g^\star - Y)^2]$, and $p(f, g) = (f - g)^2 - (f^\star - g^\star)^2$.
By the definition above, our goal is to bound

$$\mathbb{P}\left\{\exists (f, g) \in \mathcal{F} \times \mathcal{G} : \mathbb{E}[A_{f,g}] - \frac{1}{N}\sum_i k_i - \frac{1}{N}\sum_j p_j \geq \epsilon\left(\alpha + \beta + \mathbb{E}[A_{f,g}]\right)\right\} \tag{12}$$

Observe that when $N \leq \dfrac{128(\lambda + 1)B^2}{\epsilon^2\,(\alpha + \beta)}$, we have

$$10\mathfrak{M}_{\mathcal{F},\mathcal{G}}^{n,m,\lambda}\exp\left(-\frac{\epsilon^2(1 - \epsilon)\alpha N}{512 B^2(\lambda + 1)}\right) \geq 15\exp\left(-\frac{\epsilon^2(1 - \epsilon)(\alpha + \beta)N}{512 B^2(\lambda + 1)}\right) \geq 15\exp\left(-\frac{128}{512}\right) > 1,$$

thus it is sufficient to only consider the case when $N > \dfrac{128(\lambda + 1)B^2}{\epsilon^2\,(\alpha + \beta)}$.

To bound (12), we divide the proof into several steps.

### Step 1. Symmetrization

Let $\mathscr{D}_L' = \{(X_i', W_i', Y_i')\}_{i=1}^n \overset{\text{i.i.d.}}{\sim} P_{X,W,Y}$, and $\mathscr{D}_U' = \{Z_j' = (X_j', W_j')\}_{j=n+1}^N \overset{\text{i.i.d.}}{\sim} P_{X,W}$, be a *ghost* sample independent of $\mathscr{D}_L$ and $\mathscr{D}_U$, respectively. Denote

$$k_i' = (f(X_i') - Y_i')^2 + \lambda\left(g(Z_i') - Y_i'\right)^2 - (f^\star(X_i') - Y_i')^2 - \lambda\left(g^\star(Z_i') - Y_i'\right)^2,$$
$$p_j' = (f(X_j') - g(Z_j'))^2 - (f^\star(X_j') - g^\star(Z_j'))^2,$$

and define the original and ghost collections of labeled and unlabeled summands

$$\mathscr{D} = \{\, k_i, \, p_j : 1 \leq i \leq n, \, n + 1 \leq j \leq N \,\}, \qquad \mathscr{D}' = \{\, k_i', \, p_j' : 1 \leq i \leq n, \, n + 1 \leq j \leq N \,\}.$$

We will show that

$$\mathbb{P}\left\{\exists (f, g) \in \mathcal{F} \times \mathcal{G} : \mathbb{E}[A_{f,g}] - \frac{1}{N}\sum_i k_i - \frac{1}{N}\sum_j p_j \geq \epsilon\left(\alpha + \beta + \mathbb{E}[A_{f,g}]\right)\right\}$$
$$\leq \tfrac{8}{7}\,\mathbb{P}\left\{\exists (f, g) \in \mathcal{F} \times \mathcal{G} : \frac{1}{N}\left(\sum_i (k_i' - k_i) + \sum_j (p_j' - p_j)\right) \geq \tfrac{\epsilon}{2}\left(\alpha + \beta + \mathbb{E}[A_{f,g}]\right)\right\} \tag{13}$$

To prove this, consider

$$
\mathrm{Var}\left(\frac{1}{N}\left(\sum_i k_i' + \sum_j p_j'\right)\right) = \frac{1}{N^2}\mathrm{Var}\left(\sum_i k_i' + \sum_j p_j'\right) \le \frac{1}{N^2}\left(n\mathrm{Var}(k_1') + m\mathrm{Var}(p_1')\right)
$$

$$
\le \frac{a}{N}\mathbb{E}\left[\left((f-y)^2 + \lambda(g-y)^2 - (f^\star - y)^2 - \lambda(g^\star - y)^2\right)^2\right] + \frac{b}{N}\mathbb{E}\left[\left((f-g)^2 - (f^\star - g^\star)^2\right)^2\right]
$$

$$
\le \frac{1}{N}\mathbb{E}\Big[a\big((f + f^\star - 2y)(f - f^\star) + \lambda(g + g^\star - 2y)(g - g^\star)\big)^2
$$
$$
+ b\big((f - g + f^\star - g^\star)(f - g - f^\star + g^\star)\big)^2\Big]
$$

$$
\overset{(*)}{\le} \frac{1}{N}\mathbb{E}\Big[a\big((f + f^\star - 2y)^2 + \lambda(g + g^\star - 2y)^2\big)\big((f - f^\star)^2 + \lambda(g - g^\star)^2\big)
$$
$$
+ b\big((f - g + f^\star - g^\star)(f - g - f^\star + g^\star)\big)^2\Big]
$$

$$
\le \frac{16(\lambda + 1)B^2}{N}\mathbb{E}\Big[\big(a\big((f - f^\star)^2 + \lambda(g - g^\star)^2\big) + b\big(f - f^\star + g^\star - g\big)^2\big)\Big]
$$

$$
= \frac{16(\lambda + 1)B^2}{N}\mathbb{E}\left[\|(f,g) - (f^\star, g^\star)\|_*^2\right]
$$

$$
\overset{(\dagger)}{\le} \frac{16(\lambda + 1)B^2}{N}\mathbb{E}\left[\|(f,g) - (Y,Y)\|_*^2 - \|(f^\star, g^\star) - (Y,Y)\|_*^2\right]
$$

$$
\le \frac{16(\lambda + 1)B^2}{N}\mathbb{E}\left[A_{f,g}\right], \tag{14}
$$

where $(*)$ follows by applying Cauchy–Schwarz, and $(\dagger)$ holds since $(f^\star, g^\star)$ minimizes $\mathbb{E}\|(f,g) - (Y,Y)\|_*^2 = \mathcal{L}(f,g;\lambda)$. Thus,

$$
\mathbb{E}\left[A_{f,g}\right] \ge \frac{1}{16(\lambda + 1)B^2}\left(a\mathbb{E}[k^2(f,g)] + b\mathbb{E}[p^2(f,g)]\right) \tag{15}
$$

Next, find $f_{\mathscr{D}}, g_{\mathscr{D}}$ such that

$$
\mathbb{E}\left[A_{f_{\mathscr{D}},g_{\mathscr{D}}} \mid \mathscr{D}\right] - \frac{1}{N}\sum_i k_i(f_{\mathscr{D}}, g_{\mathscr{D}}) - \frac{1}{N}\sum_j p_j(f_{\mathscr{D}}, g_{\mathscr{D}}) \ge \epsilon\left(\alpha + \beta + \mathbb{E}\left[A_{f_{\mathscr{D}},g_{\mathscr{D}}} \mid \mathscr{D}\right]\right).
$$

And consider

$$
\mathbb{P}\left\{\mathbb{E}\left[A_{f_{\mathscr{D}},g_{\mathscr{D}}} \mid \mathscr{D}\right] - \frac{1}{N}\sum_i k_i'(f_{\mathscr{D}}, g_{\mathscr{D}}) - \frac{1}{N}\sum_j p_j'(f_{\mathscr{D}}, g_{\mathscr{D}}) \ge \frac{\epsilon}{2}\left(\alpha + \beta + \mathbb{E}\left[A_{f_{\mathscr{D}},g_{\mathscr{D}}} \mid \mathscr{D}\right]\right)\right\}
$$

$$
\le \frac{\mathrm{Var}\left(\frac{1}{N}\left(\sum_i k_i' + \sum_j p_j'\right) \mid \mathscr{D}\right)}{\left(\frac{\epsilon}{2}\left(\alpha + \beta + \mathbb{E}\left[A_{f_{\mathscr{D}},g_{\mathscr{D}}} \mid \mathscr{D}\right]\right)\right)^2}
$$

$$
\overset{(*)}{\le} \frac{64(\lambda + 1)B^2}{N\epsilon^2\left(\alpha + \beta + \mathbb{E}\left[A_{f_{\mathscr{D}},g_{\mathscr{D}}} \mid \mathscr{D}\right]\right)^2}\mathbb{E}\left[A_{f_{\mathscr{D}},g_{\mathscr{D}}} \mid \mathscr{D}\right]
$$

$$
\le \frac{16(\lambda + 1)B^2}{N\epsilon^2(\alpha + \beta)} < \frac{1}{8},
$$

where $(*)$ follows by (14) and in the last inequality we used $N > \dfrac{128(\lambda + 1)B^2}{\epsilon^2(\alpha + \beta)}$.

Define the two events

$$\mathcal{A} := \left\{ \mathbb{E}\left[A_{f_{\mathscr{D}},g_{\mathscr{D}}} \mid \mathscr{D}\right] - \frac{1}{N}\sum_i k_i(f_{\mathscr{D}}, g_{\mathscr{D}}) - \frac{1}{N}\sum_j p_j(f_{\mathscr{D}}, g_{\mathscr{D}}) \geq \epsilon\left(\alpha + \beta + \mathbb{E}\left[A_{f_{\mathscr{D}},g_{\mathscr{D}}} \mid \mathscr{D}\right]\right) \right\}$$

$$\mathcal{B} := \left\{ \mathbb{E}\left[A_{f_{\mathscr{D}},g_{\mathscr{D}}} \mid \mathscr{D}\right] - \frac{1}{N}\sum_i k_i'(f_{\mathscr{D}}, g_{\mathscr{D}}) - \frac{1}{N}\sum_j p_j'(f_{\mathscr{D}}, g_{\mathscr{D}}) \leq \frac{\epsilon}{2}\left(\alpha + \beta + \mathbb{E}\left[A_{f_{\mathscr{D}},g_{\mathscr{D}}} \mid \mathscr{D}\right]\right) \right\}$$

Hence,

$$\mathbb{P}\left\{ \begin{array}{c} \exists (f,g) \in \mathcal{F} \times \mathcal{G} : \mathbb{E}[A_{f,g}] - \frac{1}{N}\sum_i k_i - \frac{1}{N}\sum_j p_j \\ \geq \epsilon\left(\alpha + \beta + \mathbb{E}[A_{f,g}]\right) \end{array} \right\}$$

$$= \mathbb{P}\left\{ \begin{array}{c} \mathbb{E}\left[A_{f_{\mathscr{D}},g_{\mathscr{D}}} \mid \mathscr{D}\right] - \frac{1}{N}\sum_i k_i(f_{\mathscr{D}}, g_{\mathscr{D}}) - \frac{1}{N}\sum_j p_j(f_{\mathscr{D}}, g_{\mathscr{D}}) \\ \geq \epsilon\left(\alpha + \beta + \mathbb{E}\left[A_{f_{\mathscr{D}},g_{\mathscr{D}}} \mid \mathscr{D}\right]\right) \end{array} \right\}$$

$$= \mathbb{P}\{\mathcal{A} \cap \mathcal{B}\}/\mathbb{P}\{\mathcal{B} \mid \mathcal{A}\} \leq \frac{8}{7}\mathbb{P}\{\mathcal{A} \cap \mathcal{B}\}$$

$$\leq \frac{8}{7}\mathbb{P}\left\{ \begin{array}{c} \frac{1}{N}\left(\sum_i k_i'(f_{\mathscr{D}}, g_{\mathscr{D}}) + \sum_j p_j'(f_{\mathscr{D}}, g_{\mathscr{D}}) - \sum_i k_i(f_{\mathscr{D}}, g_{\mathscr{D}}) - \sum_j p_j(f_{\mathscr{D}}, g_{\mathscr{D}})\right) \\ \geq \frac{\epsilon}{2}\left(\alpha + \beta + \mathbb{E}\left[A_{f_{\mathscr{D}},g_{\mathscr{D}}} \mid \mathscr{D}\right]\right) \end{array} \right\}$$

$$\leq \frac{8}{7}\mathbb{P}\left\{ \begin{array}{c} \exists (f,g) \in \mathcal{F} \times \mathcal{G} : \frac{1}{N}\left(\sum_i (k_i'(f,g) - k_i(f,g)) + \sum_j (p_j'(f,g) - p_j(f,g))\right) \\ \geq \frac{\epsilon}{2}\left(\alpha + \beta + \mathbb{E}\left[A_{f,g}\right]\right) \end{array} \right\}.$$

Therefore (13) holds.

**Step 2. Population to empirical**
Next, we show that

$$\mathbb{P}\left\{ \exists (f,g) \in \mathcal{F} \times \mathcal{G} : \frac{1}{N}\left(\sum_i (k_i' - k_i) + \sum_j (p_j' - p_j)\right) \geq \frac{\epsilon}{2}\left(\alpha + \beta + \mathbb{E}\left[A_{f,g}\right]\right) \right\}$$

$$\leq \mathbb{P}\left\{ \exists (f,g) \in \mathcal{F} \times \mathcal{G} : \frac{1}{N}\left(\sum_i (k_i' - k_i) + \sum_j (p_j' - p_j)\right) \geq \frac{\epsilon}{2}(\alpha + \beta) \right.$$

$$\left. + \frac{\epsilon}{64(\lambda+1)B^2}\frac{1-\epsilon}{1+\epsilon}\left(\frac{1}{N}\left(\sum_i (k_i^2 + k_i'^2) + \sum_j (p_j^2 + p_j'^2)\right) - \frac{2\epsilon}{1-\epsilon}(\alpha + \beta)\right) \right\}$$

$$+ 8\mathbb{E}\left[\mathcal{N}_1\left(\frac{\alpha+\beta}{10a}, \mathcal{G}_k, z^n\right) + \mathcal{N}_1\left(\frac{\alpha+\beta}{10b}, \mathcal{G}_p, z^m\right)\right]\exp\left(-\frac{\epsilon^2(\alpha+\beta)N}{480B^4(\lambda+1)^2}\right). \tag{16}$$

To prove (16), we first present the following lemma.

**Lemma C.3.** *Let* $\mathcal{G}_k = \{k(f,g) : (f,g) \in \mathcal{F} \times \mathcal{G}\}$, $\mathcal{G}_p = \{p(f,g) : (f,g) \in \mathcal{F} \times \mathcal{G}\}$. $(z^n, y^n), z^m$ *are the samples used*

*to generate $k_i, p_j$ respectively. Then*

$$\mathbb{P}\left\{\exists(f,g)\in\mathcal{F}\times\mathcal{G}: \frac{\frac{1}{N}\left(\sum_i k_i^2 + \sum_j p_j^2\right) - \mathbb{E}[ak^2 + bp^2]}{\alpha + \beta + \frac{1}{N}\left(\sum_i k_i^2 + \sum_j p_j^2\right) + \mathbb{E}[ak^2 + bp^2]} \geq \epsilon\right\}$$

$$\leq 4\mathbb{E}\left[\mathcal{N}_1\left(\frac{\alpha+\beta}{10a},\mathcal{G}_k,z^n\right) + \mathcal{N}_1\left(\frac{\alpha+\beta}{10b},\mathcal{G}_p,z^m\right)\right]\exp\left(-\frac{\epsilon^2(\alpha+\beta)N}{480B^4(\lambda+1)^2}\right).$$

*Proof.* The desired bound follows.

$$\mathbb{P}\left\{\exists(f,g)\in\mathcal{F}\times\mathcal{G}: \frac{\frac{1}{N}\left(\sum_i k_i^2 + \sum_j p_j^2\right) - \mathbb{E}[ak^2 + bp^2]}{\alpha + \beta + \frac{1}{N}\left(\sum_i k_i^2 + \sum_j p_j^2\right) + \mathbb{E}[ak^2 + bp^2]} \geq \epsilon\right\}$$

$$= \mathbb{P}\left\{\sup_{(f,g)\in\mathcal{F}\times\mathcal{G}} \frac{\frac{1}{N}\left(\sum_i k_i^2 + \sum_j p_j^2\right) - \mathbb{E}[ak^2 + bp^2]}{\alpha + \beta + \frac{1}{N}\left(\sum_i k_i^2 + \sum_j p_j^2\right) + \mathbb{E}[ak^2 + bp^2]} \geq \epsilon\right\}$$

$$= \mathbb{P}\left\{\sup_{(f,g)\in\mathcal{F}\times\mathcal{G}} \frac{a\left(\frac{1}{n}\sum_i k_i^2 - \mathbb{E}[k^2]\right) + b\left(\frac{1}{m}\sum_j p_j^2 - \mathbb{E}[p^2]\right)}{\alpha + \beta + a\left(\frac{1}{n}\sum_i k_i^2 + \mathbb{E}[k^2]\right) + b\left(\frac{1}{m}\sum_j p_j^2 + \mathbb{E}[p^2]\right)} \geq \epsilon\right\}$$

$$\leq \mathbb{P}\left\{\sup_{(f,g)\in\mathcal{F}\times\mathcal{G}} \frac{a\left(\frac{1}{n}\sum_i k_i^2 - \mathbb{E}[k^2]\right)}{\frac{\alpha+\beta}{2} + a\left(\frac{1}{n}\sum_i k_i^2 + \mathbb{E}[k^2]\right)} \geq \epsilon\right\} + \mathbb{P}\left\{\sup_{(f,g)\in\mathcal{F}\times\mathcal{G}} \frac{b\left(\frac{1}{m}\sum_j p_j^2 - \mathbb{E}[p^2]\right)}{\frac{\alpha+\beta}{2} + b\left(\frac{1}{m}\sum_j p_j^2 + \mathbb{E}[p^2]\right)} \geq \epsilon\right\}$$

$$\leq 4\mathbb{E}\left[\mathcal{N}_1\left(\frac{\alpha+\beta}{10a},\mathcal{G}_k,z^n\right) + \mathcal{N}_1\left(\frac{\alpha+\beta}{10b},\mathcal{G}_p,z^m\right)\right]\exp\left(-\frac{\epsilon^2(\alpha+\beta)N}{480B^4(\lambda+1)^2}\right),$$

where the last inequality uses Györfi et al. (2002, Theorem 11.6). $\qquad\square$

Next, define the events

$$\mathcal{S}_1 := \left\{ \begin{array}{c} \exists (f,g) \in \mathcal{F} \times \mathcal{G} : \dfrac{1}{N} \left( \sum_i (k_i' - k_i) + \sum_j (p_j' - p_j) \right) \\[2mm] \geq \dfrac{\epsilon}{2} \left( \alpha + \beta + \mathbb{E}[A_{f,g}] \right) \end{array} \right\},$$

$$\mathcal{S}_2 := \left\{ \begin{array}{c} \forall (f,g) \in \mathcal{F} \times \mathcal{G} : \\[2mm] \dfrac{1}{N} \left( \sum_i k_i^2 + \sum_j p_j^2 \right) - \mathbb{E}[ak^2 + bp^2] \\[2mm] \leq \epsilon \left( \alpha + \beta + \dfrac{1}{N} \left( \sum_i k_i^2 + \sum_j p_j^2 \right) + \mathbb{E}[ak^2 + bp^2] \right) \end{array} \right\},$$

$$\mathcal{S}_3 := \left\{ \begin{array}{c} \forall (f,g) \in \mathcal{F} \times \mathcal{G} : \\[2mm] \dfrac{1}{N} \left( \sum_i (k_i')^2 + \sum_j (p_j')^2 \right) - \mathbb{E}[ak^2 + bp^2] \\[2mm] \leq \epsilon \left( \alpha + \beta + \dfrac{1}{N} \left( \sum_i (k_i')^2 + \sum_j (p_j')^2 \right) + \mathbb{E}[ak^2 + bp^2] \right) \end{array} \right\}.$$

We have the set inclusion $\mathcal{S}_1 \subset (\mathcal{S}_1 \cap \mathcal{S}_2 \cap \mathcal{S}_3) \cup \mathcal{S}_2^c \cup \mathcal{S}_3^c$, hence

$$\begin{aligned}
\mathbb{P}(\mathcal{S}_1) &\leq \mathbb{P}(\mathcal{S}_1 \cap \mathcal{S}_2 \cap \mathcal{S}_3) + \mathbb{P}(\mathcal{S}_2^c) + \mathbb{P}(\mathcal{S}_3^c) \\
&\leq \mathbb{P}(\mathcal{S}_1 \cap \mathcal{S}_2 \cap \mathcal{S}_3) + 2\,\mathbb{P}(\mathcal{S}_2^c) \\
&\leq \mathbb{P}\left\{ \exists (f,g) \in \mathcal{F} \times \mathcal{G} : \dfrac{1}{N} \left( \sum_i (k_i' - k_i) + \sum_j (p_j' - p_j) \right) \geq \dfrac{\epsilon}{2}(\alpha + \beta) \right. \\
&\qquad \left. + \dfrac{\epsilon}{64(\lambda+1)B^2} \dfrac{1-\epsilon}{1+\epsilon} \left( \dfrac{1}{N} \left( \sum_i (k_i^2 + k_i'^2) + \sum_j (p_j^2 + p_j'^2) \right) - \dfrac{2\epsilon}{1-\epsilon}(\alpha+\beta) \right) \right\} \\
&\quad + 8\mathbb{E}\left[ \mathcal{N}_1\left( \dfrac{\alpha+\beta}{10a}, \mathcal{G}_k, z^n \right) + \mathcal{N}_1\left( \dfrac{\alpha+\beta}{10b}, \mathcal{G}_p, z^m \right) \right] \exp\left( -\dfrac{\epsilon^2(\alpha+\beta)N}{480B^4(\lambda+1)^2} \right),
\end{aligned}$$

where the last inequality uses Lemma C.3 and the fact that $\mathcal{S}_2$ implies

$$\mathbb{E}\left[A_{f,g}\right] \overset{(*)}{\geq} \dfrac{1}{16(\lambda+1)B^2} \mathbb{E}[ak^2 + bp^2] \geq \dfrac{1}{16(\lambda+1)B^2} \dfrac{1-\epsilon}{1+\epsilon} \left( \dfrac{1}{N} \left( \sum_i k_i^2 + \sum_j p_j^2 \right) - \dfrac{\epsilon}{1-\epsilon}(\alpha+\beta) \right)$$

Similarly for $\mathscr{D}'$, $\mathcal{S}_3$ implies

$$\mathbb{E}\left[A_{f,g}\right] \overset{(*)}{\geq} \dfrac{1}{16(\lambda+1)B^2} \mathbb{E}[ak^2 + bp^2] \geq \dfrac{1}{16(\lambda+1)B^2} \dfrac{1-\epsilon}{1+\epsilon} \left( \dfrac{1}{N} \left( \sum_i k_i'^2 + \sum_j p_j'^2 \right) - \dfrac{\epsilon}{1-\epsilon}(\alpha+\beta) \right),$$

where $(*)$ follows by (15). Therefore (16) holds.

**Step 3. Rademacher variables.** We introduce independent Rademacher variables $\{U_i\}_{i=1}^n$ and $\{U_j'\}_{j=n+1}^N$, i.e., i.i.d.

random variables satisfying $\mathbb{P}(U_i = 1) = \mathbb{P}(U_i = -1) = \mathbb{P}(U'_j = 1) = \mathbb{P}(U'_j = -1) = \frac{1}{2}$. Then

$$
\mathbb{P}\Bigg\{ \exists (f,g) \in \mathcal{F} \times \mathcal{G} : \frac{1}{N} \left( \sum_i (k'_i - k_i) + \sum_j (p'_j - p_j) \right) \geq \frac{\epsilon}{2} (\alpha + \beta)
$$
$$
+ \frac{\epsilon}{64(\lambda+1)B^2} \frac{1-\epsilon}{1+\epsilon} \left( \frac{1}{N} \left( \sum_i (k_i^2 + k'^2_i) + \sum_j (p_j^2 + p'^2_j) \right) - \frac{2\epsilon}{1-\epsilon}(\alpha + \beta) \right) \Bigg\}
$$
$$
\leq 2\mathbb{P}\Bigg\{ \exists (f,g) \in \mathcal{F} \times \mathcal{G} : \frac{1}{N} \left| \sum_i U_i k_i + \sum_j U_j p_j \right| \geq \frac{\epsilon}{4}(\alpha + \beta)
$$
$$
+ \frac{\epsilon}{64(\lambda+1)B^2} \frac{1-\epsilon}{1+\epsilon} \left( \frac{1}{N} \left( \sum_i k_i^2 + \sum_j p_j^2 \right) - \frac{\epsilon}{1-\epsilon}(\alpha + \beta) \right) \Bigg\}, \tag{17}
$$

which follows by symmetrization and a union bound.

$$
\mathbb{P}\Bigg\{ \exists (f,g) \in \mathcal{F} \times \mathcal{G} : \frac{1}{N} \left( \sum_i (k'_i - k_i) + \sum_j (p'_j - p_j) \right) \geq \frac{\epsilon}{2}(\alpha + \beta)
$$
$$
+ \frac{\epsilon}{64(\lambda+1)B^2} \frac{1-\epsilon}{1+\epsilon} \left( \frac{1}{N} \left( \sum_i (k_i^2 + k'^2_i) + \sum_j (p_j^2 + p'^2_j) \right) - \frac{2\epsilon}{1-\epsilon}(\alpha + \beta) \right) \Bigg\}
$$
$$
= \mathbb{P}\Bigg\{ \exists (f,g) \in \mathcal{F} \times \mathcal{G} : \frac{1}{N} \left( \sum_i U'_i k'_i - U_i k_i + \sum_j U'_j p'_j - U_j p_j \right) \geq \frac{\epsilon}{2}(\alpha + \beta)
$$
$$
+ \frac{\epsilon}{64(\lambda+1)B^2} \frac{1-\epsilon}{1+\epsilon} \left( \frac{1}{N} \left( \sum_i (k_i^2 + k'^2_i) + \sum_j (p_j^2 + p'^2_j) \right) - \frac{2\epsilon}{1-\epsilon}(\alpha + \beta) \right) \Bigg\}
$$
$$
\leq \mathbb{P}\Bigg\{ \exists (f,g) \in \mathcal{F} \times \mathcal{G} : \frac{1}{N} \left| \sum_i U_i k_i + \sum_j U_j p_j \right| \geq \frac{\epsilon}{4}(\alpha + \beta)
$$
$$
+ \frac{\epsilon}{64(\lambda+1)B^2} \frac{1-\epsilon}{1+\epsilon} \left( \frac{1}{N} \left( \sum_i k_i^2 + \sum_j p_j^2 \right) - \frac{\epsilon}{1-\epsilon}(\alpha + \beta) \right) \Bigg\}
$$
$$
+ \mathbb{P}\Bigg\{ \exists (f,g) \in \mathcal{F} \times \mathcal{G} : \frac{1}{N} \left| \sum_i U'_i k'_i + \sum_j U'_j p'_j \right| \geq \frac{\epsilon}{4}(\alpha + \beta)
$$
$$
+ \frac{\epsilon}{64(\lambda+1)B^2} \frac{1-\epsilon}{1+\epsilon} \left( \frac{1}{N} \left( \sum_i k'^2_i + \sum_j p'^2_j \right) - \frac{\epsilon}{1-\epsilon}(\alpha + \beta) \right) \Bigg\}
$$
$$
= 2\mathbb{P}\Bigg\{ \exists (f,g) \in \mathcal{F} \times \mathcal{G} : \frac{1}{N} \left| \sum_i U_i k_i + \sum_j U_j p_j \right| \geq \frac{\epsilon}{4}(\alpha + \beta)
$$
$$
+ \frac{\epsilon}{64(\lambda+1)B^2} \frac{1-\epsilon}{1+\epsilon} \left( \frac{1}{N} \left( \sum_i k_i^2 + \sum_j p_j^2 \right) - \frac{\epsilon}{1-\epsilon}(\alpha + \beta) \right) \Bigg\}.
$$

Therefore (17) holds.

**Step 4. Covering numbers.** Define $\Gamma_\delta$ as a smallest subset of $\mathcal{F} \times \mathcal{G}$ such that, for every $(f,g) \in \mathcal{F} \times \mathcal{G}$, there exists

$(\tilde{f}, \tilde{g}) \in \Gamma_\delta$ satisfying

$$\frac{1}{N} \left( \sum_i \left| k_i(f, g) - k_i(\tilde{f}, \tilde{g}) \right| + \sum_j \left| p_j(f, g) - p_j(\tilde{f}, \tilde{g}) \right| \right) \leq \delta.$$

For notational simplicity, we write $\tilde{k}_i$ and $\tilde{p}_j$ in place of $k_i(\tilde{f}, \tilde{g})$ and $p_j(\tilde{f}, \tilde{g})$, respectively. We show that

$$\mathbb{P} \left\{ \begin{array}{l} \exists (f, g) \in \mathcal{F} \times \mathcal{G} : \dfrac{1}{N} \left| \sum_i U_i k_i + \sum_j U_j p_j \right| \\[2ex] \geq \dfrac{\epsilon}{4} (\alpha + \beta) + \dfrac{\epsilon}{64(\lambda + 1)B^2} \dfrac{1 - \epsilon}{1 + \epsilon} \left( \dfrac{1}{N} \left( \sum_i k_i^2 + \sum_j p_j^2 \right) - \dfrac{\epsilon}{1 - \epsilon} (\alpha + \beta) \right) \end{array} \right\}$$

$$\leq \left| \Gamma_{\frac{\epsilon\beta}{5}} \right| \max_{(\tilde{f}, \tilde{g}) \in \Gamma_{\frac{\epsilon\beta}{5}}} \mathbb{P} \left\{ \begin{array}{l} \dfrac{1}{N} \left| \sum_i U_i \tilde{k}_i + \sum_j U_j \tilde{p}_j \right| \\[2ex] \geq \dfrac{\epsilon}{4}\alpha + \dfrac{\epsilon}{64(\lambda + 1)B^2} \dfrac{1 - \epsilon}{1 + \epsilon} \left( \dfrac{1}{N} \left( \sum_i \tilde{k}_i^2 + \sum_j \tilde{p}_j^2 \right) - \dfrac{\epsilon}{1 - \epsilon}\alpha \right) \end{array} \right\}. \tag{18}$$

We show (18) as follows. Conditioned on $\mathscr{D}$, we have

$$\mathbb{P} \left\{ \begin{array}{l} \exists (f, g) \in \mathcal{F} \times \mathcal{G} : \dfrac{1}{N} \left| \sum_i U_i k_i + \sum_j U_j p_j \right| \\[2ex] \geq \dfrac{\epsilon}{4} (\alpha + \beta) + \dfrac{\epsilon}{64(\lambda + 1)B^2} \dfrac{1 - \epsilon}{1 + \epsilon} \left( \dfrac{1}{N} \left( \sum_i k_i^2 + \sum_j p_j^2 \right) - \dfrac{\epsilon}{1 - \epsilon} (\alpha + \beta) \right) \end{array} \right\}$$

$$\overset{(*)}{\leq} |\Gamma_\delta| \max_{(\tilde{f}, \tilde{g}) \in \Gamma_\delta} \mathbb{P} \left\{ \begin{array}{l} \dfrac{1}{N} \left| \sum_i U_i \tilde{k}_i + \sum_j U_j \tilde{p}_j \right| + \delta \\[2ex] \geq \dfrac{\epsilon}{4} (\alpha + \beta) + \dfrac{\epsilon}{64(\lambda + 1)B^2} \dfrac{1 - \epsilon}{1 + \epsilon} \\[2ex] \times \left( \dfrac{1}{N} \left( \sum_i \tilde{k}_i^2 + \sum_j \tilde{p}_j^2 \right) - 8B^2(\lambda + 1)\delta - \dfrac{\epsilon}{1 - \epsilon} (\alpha + \beta) \right) \end{array} \right\}$$

$$\overset{(\dagger)}{\leq} \left| \Gamma_{\frac{\epsilon\beta}{5}} \right| \max_{(\tilde{f}, \tilde{g}) \in \Gamma_{\frac{\epsilon\beta}{5}}} \mathbb{P} \left\{ \begin{array}{l} \dfrac{1}{N} \left| \sum_i U_i \tilde{k}_i + \sum_j U_j \tilde{p}_j \right| \\[2ex] \geq \dfrac{\epsilon}{4}\alpha + \dfrac{\epsilon}{64(\lambda + 1)B^2} \dfrac{1 - \epsilon}{1 + \epsilon} \left( \dfrac{1}{N} \left( \sum_i \tilde{k}_i^2 + \sum_j \tilde{p}_j^2 \right) - \dfrac{\epsilon}{1 - \epsilon}\alpha \right) \end{array} \right\},$$

where $(*)$ follows by observing that

$$\frac{1}{N} \left| \sum_i U_i k_i + \sum_j U_j p_j \right| - \frac{1}{N} \left| \sum_i U_i \tilde{k}_i + \sum_j U_j \tilde{p}_j \right| \leq \frac{1}{N} \left| \sum_i U_i (k_i - \tilde{k}_i) + \sum_j U_j (p_j - \tilde{p}_j) \right|$$

$$\leq \frac{1}{N} \left( \sum_i \left| k_i(f, g) - k_i(\tilde{f}, \tilde{g}) \right| + \sum_j \left| p_j(f, g) - p_j(\tilde{f}, \tilde{g}) \right| \right) \leq \delta,$$

and

$$\frac{1}{N}\left(\sum_i \tilde{k}_i^2 + \sum_j \tilde{p}_j^2\right) - \frac{1}{N}\left(\sum_i k_i^2 + \sum_j p_j^2\right)$$

$$\leq \frac{1}{N}\left(\sum_i (\tilde{k}_i - k_i)(\tilde{k}_i + k_i) + \sum_j (\tilde{p}_j - p_j)(\tilde{p}_j + p_j)\right)$$

$$\leq \frac{8B^2(\lambda+1)}{N}\left(\sum_i \left|k_i(f,g) - k_i(\tilde{f},\tilde{g})\right| + \sum_j \left|p_j(f,g) - p_j(\tilde{f},\tilde{g})\right|\right) \leq 8B^2(\lambda+1)\delta.$$

(†) follows by setting $\delta = \dfrac{\epsilon\beta}{5}$, and using

$$\frac{\epsilon\beta}{4} - \frac{\epsilon\beta}{5} - \frac{\epsilon}{64(\lambda+1)B^2}\frac{1-\epsilon}{1+\epsilon}\left(8B^2(\lambda+1)\frac{\epsilon\beta}{5} + \frac{\epsilon}{1-\epsilon}\beta\right) = \frac{\epsilon\beta}{20} - \frac{\epsilon^2}{40}\frac{1-\epsilon}{1+\epsilon}\beta - \frac{\epsilon^2}{64(\lambda+1)B^2(1+\epsilon)}\beta$$

$$\geq \frac{\epsilon\beta}{20} - \frac{\epsilon^2}{40}\frac{1-\epsilon}{1+\epsilon}\beta - \frac{\epsilon^2}{64B^2(1+\epsilon)}\beta \geq 0.$$

**Step 5. Bernstein's inequality.** Next, Bernstein's inequality gives

$$\mathbb{P}\left\{ \begin{aligned} &\frac{1}{N}\left|\sum_i U_i\tilde{k}_i + \sum_j U_j\tilde{p}_j\right| \\ &\geq \frac{\epsilon}{4}\alpha + \frac{\epsilon}{64(\lambda+1)B^2}\frac{1-\epsilon}{1+\epsilon}\left(\frac{1}{N}\left(\sum_i \tilde{k}_i^2 + \sum_j \tilde{p}_j^2\right) - \frac{\epsilon}{1-\epsilon}\alpha\right) \end{aligned} \right\}$$

$$\leq 2\exp\left(-\frac{N\epsilon^2(1-\epsilon)\alpha}{512(\lambda+1)B^2}\right). \tag{19}$$

For a fixed $(\tilde{f},\tilde{g}) \in \Gamma_{\frac{\epsilon\beta}{5}}$, we define

$$\sigma^2 := \frac{1}{N}\left(\sum_i \tilde{k}_i^2 + \sum_j \tilde{p}_j^2\right) = \frac{1}{N}\mathrm{Var}\left(\sum_i U_i\tilde{k}_i + \sum_j U_j\tilde{p}_j\right),$$

$$A_1 := \frac{\epsilon}{4}\alpha - \frac{\epsilon^2}{64(\lambda+1)B^2(1+\epsilon)}\alpha, \quad \text{and} \quad A_2 := \frac{\epsilon}{64(\lambda+1)B^2}\frac{1-\epsilon}{1+\epsilon}.$$

By Bernstein's inequality and the independence of the samples, we have

$$\mathbb{P}\left\{ \frac{1}{N}\left|\sum_i U_i \tilde{k}_i + \sum_j U_j \tilde{p}_j\right| \geq \frac{\epsilon}{4}\alpha + \frac{\epsilon}{64(\lambda+1)B^2}\frac{1-\epsilon}{1+\epsilon}\left(\frac{1}{N}\left(\sum_i \tilde{k}_i^2 + \sum_j \tilde{p}_j^2\right) - \frac{\epsilon}{1-\epsilon}\alpha\right)\right\}$$

$$= \mathbb{P}\left\{ \frac{1}{N}\left|\sum_i U_i \tilde{k}_i + \sum_j U_j \tilde{p}_j\right| \geq A_1 + A_2\sigma^2\right\}$$

$$\leq 2\exp\left(-\frac{N(A_1+A_2\sigma^2)^2}{2\sigma^2 + \frac{8}{3}B^2(\lambda+1)(A_1+A_2\sigma^2)}\right) = 2\exp\left(-\frac{3NA_2}{8B^2(\lambda+1)}\cdot\frac{(A_1/A_2+\sigma^2)^2}{\left(\frac{3}{4B^2A_2(\lambda+1)}+1\right)\sigma^2+\frac{A_1}{A_2}}\right)$$

$$\overset{(\dagger)}{\leq} 2\exp\left(-\frac{3NA_2}{8B^2(\lambda+1)}\cdot\frac{192\,\epsilon(1+\epsilon)\big(16(\lambda+1)B^2(1+\epsilon)-\epsilon\big)\alpha}{\big(\epsilon(1-\epsilon)+48(1+\epsilon)\big)^2}\right)$$

$$\leq 2\exp\left(-\frac{3NA_2}{8B^2(\lambda+1)}\cdot\frac{192\,\epsilon(1+\epsilon)\,16(\lambda+1)B^2\alpha}{\big(\epsilon(1-\epsilon)+48(1+\epsilon)\big)^2}\right)$$

$$= 2\exp\left(-\frac{3N}{8B^2(\lambda+1)}\cdot\frac{\epsilon(1-\epsilon)}{64(\lambda+1)B^2(1+\epsilon)}\cdot\frac{192\,\epsilon(1+\epsilon)\,16(\lambda+1)B^2\alpha}{\big(\epsilon(1-\epsilon)+48(1+\epsilon)\big)^2}\right)$$

$$= 2\exp\left(-\frac{18N\,\epsilon^2(1-\epsilon)\alpha}{(\lambda+1)B^2\big(\epsilon(1-\epsilon)+48(1+\epsilon)\big)^2}\right) \leq 2\exp\left(-\frac{N\epsilon^2(1-\epsilon)\alpha}{512(\lambda+1)B^2}\right),$$

where the last inequality follows using $\max_{0\leq\epsilon\leq 1}\big(\epsilon(1-\epsilon)+48(1+\epsilon)\big)^2 = 9216$. ($\dagger$) follows by noting that for any $u > 0$, $b > 2$, we have

$$\frac{(a+u)^2}{a+b\,u} \geq \frac{\left(a+\frac{b-2}{b}a\right)^2}{a+b\frac{b-2}{b}a} = 4a\,\frac{b-1}{b^2},$$

which yields that,

$$\frac{(A_1/A_2+\sigma^2)^2}{\left(\frac{3}{4B^2A_2(\lambda+1)}+1\right)\sigma^2+\frac{A_1}{A_2}} \geq \frac{\frac{192A_1(1+\epsilon)}{\epsilon(1-\epsilon)}}{A_2\left(1+\frac{48(1+\epsilon)}{\epsilon(1-\epsilon)}\right)^2} = \frac{192\,\alpha\,\epsilon(1+\epsilon)\big(16(\lambda+1)B^2(1+\epsilon)-\epsilon\big)}{\big(\epsilon(1-\epsilon)+48(1+\epsilon)\big)^2}$$

$$\geq \frac{192\,\epsilon(1+\epsilon)\,16(\lambda+1)B^2\alpha}{\big(\epsilon(1-\epsilon)+48(1+\epsilon)\big)^2}.$$

Therefore (19) holds.

**Step 6. Final bound.** Observe that if we have

$$\frac{1}{N}\left(\sum_i \left|f(X_i)-\tilde{f}(X_i)\right| + \sum_j \left|f(X_j)-\tilde{f}(X_j)\right|\right) \leq \frac{\epsilon\beta}{40B},$$

$$\frac{1}{N}\left(\sum_i |g(Z_i)-\tilde{g}(Z_i)| + \sum_j |g(Z_j)-\tilde{g}(Z_j)|\right) \leq \frac{\epsilon\beta}{40B(\lambda+1)},$$

then we have

$$\frac{1}{N}\left(\sum_i \left|k_i(f,g) - k_i(\tilde{f},\tilde{g})\right| + \sum_j \left|p_j(f,g) - p_j(\tilde{f},\tilde{g})\right|\right)$$

$$\leq \frac{1}{N}\left(\sum_i 4B\left|f(X_i) - \tilde{f}(X_i)\right| + 4\lambda B\left|g(Z_i) - \tilde{g}(Z_i)\right| + \sum_j 4B\left|f(X_j) - \tilde{f}(X_j)\right| + 4B\left|g(Z_j) - \tilde{g}(Z_j)\right|\right) \leq \frac{\epsilon\beta}{5},$$

which implies that $\left|\Gamma_{\frac{\epsilon\beta}{5}}\right| \leq \mathcal{N}_1\left(\frac{\epsilon\beta}{40B}, \mathcal{F}, x^N\right) \cdot \mathcal{N}_1\left(\frac{\epsilon\beta}{40(\lambda+1)B}, \mathcal{G}, z^N\right)$.

Combining the steps, we have the following.

$$\mathbb{P}\left\{\exists(f,g) \in \mathcal{F} \times \mathcal{G} : \mathbb{E}[A_{f,g}] - \frac{1}{N}\sum_i k_i - \frac{1}{N}\sum_j p_j \geq \epsilon\left(\alpha + \beta + \mathbb{E}[A_{f,g}]\right)\right\}$$

$$\leq \frac{16}{7}\mathcal{M}_1 \exp\left(-\frac{N\epsilon^2(1-\epsilon)\alpha}{512(\lambda+1)B^2}\right) + \frac{64}{7}\mathcal{M}_2 \exp\left(-\frac{\epsilon^2(1-\epsilon)(\alpha+\beta)N}{480B^4(\lambda+1)^2}\right)$$

where $\mathcal{M}_1 = \max_{z^N} \mathcal{N}_1\left(\frac{\epsilon\beta}{40B}, \mathcal{F}, x^N\right) \cdot \mathcal{N}_1\left(\frac{\epsilon\beta}{40(\lambda+1)B}, \mathcal{G}, z^N\right)$, and

$$\mathcal{M}_2 = \max_{z^N}\left(\mathcal{N}_1\left(\frac{\alpha+\beta}{10a}, \mathcal{G}_k, z^n\right) + \mathcal{N}_1\left(\frac{\alpha+\beta}{10b}, \mathcal{G}_p, z^m\right)\right)$$

$$\leq \max_{z^n}\mathcal{N}_1\left(\frac{\alpha+\beta}{80Ba}, \mathcal{F}, x^n\right) \cdot \mathcal{N}_1\left(\frac{\alpha+\beta}{80Ba(\lambda+1)}, \mathcal{G}, z^n\right) + \max_{z^m}\mathcal{N}_1\left(\frac{\alpha+\beta}{80Bb}, \mathcal{F}, x^m\right) \cdot \mathcal{N}_1\left(\frac{\alpha+\beta}{80Bb}, \mathcal{G}, z^m\right).$$

Thus,

$$\mathbb{P}\left\{\exists(f,g) \in \mathcal{F} \times \mathcal{G} : \mathbb{E}[A_{f,g}] - \frac{1}{N}\sum_i k_i - \frac{1}{N}\sum_j p_j \geq \epsilon\left(\alpha + \beta + \mathbb{E}[A_{f,g}]\right)\right\}$$

$$\leq \frac{16}{7}\max_{z^N}\mathcal{N}_1\left(\frac{\epsilon\beta}{40B}, \mathcal{F}, x^N\right) \cdot \mathcal{N}_1\left(\frac{\epsilon\beta}{40(\lambda+1)B}, \mathcal{G}, z^N\right) \cdot \exp\left(-\frac{N\epsilon^2(1-\epsilon)\alpha}{512(\lambda+1)B^2}\right)$$

$$+ \frac{64}{7}\max_{z^n}\mathcal{N}_1\left(\frac{\alpha+\beta}{80Ba}, \mathcal{F}, x^n\right) \cdot \mathcal{N}_1\left(\frac{\alpha+\beta}{80Ba(\lambda+1)}, \mathcal{G}, z^n\right) \cdot \exp\left(-\frac{\epsilon^2(1-\epsilon)(\alpha+\beta)N}{480B^4(\lambda+1)^2}\right)$$

$$+ \frac{64}{7}\max_{z^m}\mathcal{N}_1\left(\frac{\alpha+\beta}{80Bb}, \mathcal{F}, x^m\right) \cdot \mathcal{N}_1\left(\frac{\alpha+\beta}{80Bb}, \mathcal{G}, z^m\right) \cdot \exp\left(-\frac{\epsilon^2(1-\epsilon)(\alpha+\beta)N}{480B^4(\lambda+1)^2}\right).$$

Finally, we use scaling to obtain a cleaner bound for $s > 0$, replacing $B^2, \alpha, \beta$ by $sB^2, s\alpha, s\beta$ as follows.

$$\mathbb{P}\left\{\exists(f,g) \in \mathcal{F} \times \mathcal{G} : \mathbb{E}[A_{f,g}] - \frac{1}{N}\sum_i k_i - \frac{1}{N}\sum_j p_j \geq \epsilon\left(\alpha + \beta + \mathbb{E}[A_{f,g}]\right)\right\}$$

$$= \mathbb{P}\left\{\exists(f,g) \in \mathcal{F} \times \mathcal{G} : \mathbb{E}[a(sk) + b(sp)] - \frac{1}{N}\sum_i sk_i - \frac{1}{N}\sum_j sp_j \geq \epsilon\left(s\alpha + s\beta + \mathbb{E}[a(sk) + b(sp)]\right)\right\}$$

$$\leq \frac{16}{7}\left[\max_{z^N}\mathcal{N}_1\left(\frac{\epsilon\beta\sqrt{s}}{40B},\mathcal{F},x^N\right)\cdot\mathcal{N}_1\left(\frac{\epsilon\beta\sqrt{s}}{40(\lambda+1)B},\mathcal{G},z^N\right)\right]\exp\left(-\frac{N\epsilon^2(1-\epsilon)\alpha}{512(\lambda+1)B^2}\right)$$

$$+\frac{64}{7}\left[\max_{z^n}\mathcal{N}_1\left(\frac{(\alpha+\beta)\sqrt{s}}{80Ba},\mathcal{F},x^n\right)\cdot\mathcal{N}_1\left(\frac{(\alpha+\beta)\sqrt{s}}{80Ba(\lambda+1)},\mathcal{G},z^n\right)\right.$$

$$\left.+\max_{z^m}\mathcal{N}_1\left(\frac{(\alpha+\beta)\sqrt{s}}{80Bb},\mathcal{F},x^m\right)\cdot\mathcal{N}_1\left(\frac{(\alpha+\beta)\sqrt{s}}{80Bb},\mathcal{G},z^m\right)\right]\cdot\exp\left(-\frac{\epsilon^2(1-\epsilon)(\alpha+\beta)N}{480B^4s(\lambda+1)^2}\right).$$

Setting $s=\frac{1}{(\lambda+1)B^2}$ gives

$$\mathbb{P}\left\{\exists(f,g)\in\mathcal{F}\times\mathcal{G}:\mathbb{E}[A_{f,g}]-\frac{1}{N}\sum_i k_i-\frac{1}{N}\sum_j p_j\geq\epsilon\left(\alpha+\beta+\mathbb{E}[A_{f,g}]\right)\right\}$$

$$\leq\frac{16}{7}\left[\max_{z^N}\mathcal{N}_1\left(\frac{\epsilon\beta}{40B\sqrt{\lambda+1}},\mathcal{F},x^N\right)\cdot\mathcal{N}_1\left(\frac{\epsilon\beta}{40(\lambda+1)^{3/2}B},\mathcal{G},z^N\right)\right]\exp\left(-\frac{N\epsilon^2(1-\epsilon)\alpha}{512(\lambda+1)B^2}\right)$$

$$+\frac{64}{7}\left[\max_{z^n}\mathcal{N}_1\left(\frac{(\alpha+\beta)}{80B^2\sqrt{\lambda+1}a},\mathcal{F},x^n\right)\cdot\mathcal{N}_1\left(\frac{(\alpha+\beta)}{80B^2a(\lambda+1)^{3/2}},\mathcal{G},z^n\right)\right.$$

$$\left.+\max_{z^m}\mathcal{N}_1\left(\frac{(\alpha+\beta)}{80B^2\sqrt{\lambda+1}b},\mathcal{F},x^m\right)\cdot\mathcal{N}_1\left(\frac{(\alpha+\beta)}{80B^2\sqrt{\lambda+1}b},\mathcal{G},z^m\right)\right]\cdot\exp\left(-\frac{\epsilon^2(1-\epsilon)(\alpha+\beta)N}{480B^2(\lambda+1)}\right)$$

$$\leq 10\mathfrak{M}_{\mathcal{F},\mathcal{G}}^{n,m,\lambda}\exp\left(-\frac{\epsilon^2(1-\epsilon)\alpha N}{512B^2(\lambda+1)}\right).$$

This completes the proof of Theorem C.2. $\qquad\square$

### C.3. Proof of Theorem 2.7

In Theorem C.2, we set $\epsilon=\frac{1}{2}$, $\alpha=\beta=\frac{t}{2}$, $C=\frac{1}{7680B^2}$, and we abbreviate $k_i(\hat{f},\hat{g})$ as $\hat{k}_i$, $p_j(\hat{f},\hat{g})$ as $\hat{p}_j$. Then

$$\mathbb{P}\left\{\mathbb{E}[\|(Y,Y)-(\hat{f}(X),\hat{g}(Z))\|_*^2]-\mathbb{E}[\|(Y,Y)-(f^\star(X),g^\star(Z))\|_*^2]-\frac{1}{N}\left(\sum_i\hat{k}_i+\sum_j\hat{p}_j\right)\right.$$

$$\left.\geq\epsilon\left(\alpha+\beta+\mathbb{E}[\|(Y,Y)-(\hat{f}(X),\hat{g}(Z))\|_*^2]-\mathbb{E}[\|(Y,Y)-(f^\star(X),g^\star(Z))\|_*^2]\right)\right\}$$

$$\leq 5\left(\max_{z^N}\mathcal{N}_1\left(\frac{t}{160B},\mathcal{F},x^N\right)\mathcal{N}_1\left(\frac{t}{160(\lambda+1)B},\mathcal{G},z^N\right)+\max_{z^n}\mathcal{N}_1\left(\frac{t}{80Ba},\mathcal{F},x^n\right)\mathcal{N}_1\left(\frac{t}{80Ba(\lambda+1)},\mathcal{G},z^n\right)\right.$$

$$\left.+\max_{z^m}\mathcal{N}_1\left(\frac{t}{80Bb},\mathcal{F},x^m\right)\mathcal{N}_1\left(\frac{t}{80Bb},\mathcal{G},z^m\right)\right)\exp\left(-\frac{CNt}{\lambda+1}\right)$$

Hence,

$$\mathbb{E}_D\left[\mathbb{E}[\|(Y,Y)-(\hat{f}(X),\hat{g}(Z))\|_*^2]-\mathbb{E}[\|(Y,Y)-(f^\star(X),g^\star(Z))\|_*^2]-\frac{2}{N}\left(\sum_i\hat{k}_i+\sum_j\hat{p}_j\right)\right]$$

$$\leq\int_0^\infty\mathbb{P}\left\{\mathbb{E}[\|(Y,Y)-(\hat{f}(X),\hat{g}(Z))\|_*^2]-\mathbb{E}[\|(Y,Y)-(f^\star(X),g^\star(Z))\|_*^2]-\frac{2}{N}\left(\sum_i\hat{k}_i+\sum_j\hat{p}_j\right)\geq t\right\}dt$$

$$\leq\delta+\int_\delta^\infty\mathbb{P}\left\{\mathbb{E}[\|(Y,Y)-(\hat{f}(X),\hat{g}(Z))\|_*^2]-\mathbb{E}[\|(Y,Y)-(f^\star(X),g^\star(Z))\|_*^2]-\frac{2}{N}\left(\sum_i\hat{k}_i+\sum_j\hat{p}_j\right)\geq t\right\}dt$$

$$\leq \delta + \int_\delta^\infty 5\left(\max_{z^n}\mathcal{N}_1\left(\frac{t}{80B\sqrt{\lambda+1}},\mathcal{F},x^n\right)\mathcal{N}_1\left(\frac{t}{80B(\lambda+1)^{3/2}},\mathcal{G},z^n\right)\right.$$

$$+\max_{z^m}\mathcal{N}_1\left(\frac{t}{80B\sqrt{\lambda+1}},\mathcal{F},x^m\right)\mathcal{N}_1\left(\frac{t}{80B\sqrt{\lambda+1}},\mathcal{G},z^m\right)$$

$$\left.+\max_{z^N}\mathcal{N}_1\left(\frac{t}{160B\sqrt{\lambda+1}},\mathcal{F},x^N\right)\mathcal{N}_1\left(\frac{t}{160(\lambda+1)^{3/2}B},\mathcal{G},z^N\right)\right)\exp\left(-\frac{CNt}{\lambda+1}\right)dt$$

$$\leq \delta + \int_\delta^\infty 5\left(\max_{z^n}\mathcal{N}_1\left(\frac{1}{80B\sqrt{\lambda+1}N},\mathcal{F},x^n\right)\mathcal{N}_1\left(\frac{1}{80BN(\lambda+1)^{3/2}},\mathcal{G},z^n\right)\right.$$

$$+\max_{z^m}\mathcal{N}_1\left(\frac{1}{80B\sqrt{\lambda+1}N},\mathcal{F},x^m\right)\mathcal{N}_1\left(\frac{1}{80B\sqrt{\lambda+1}N},\mathcal{G},z^m\right)$$

$$\left.+\max_{z^N}\mathcal{N}_1\left(\frac{1}{160BN\sqrt{\lambda+1}},\mathcal{F},x^N\right)\mathcal{N}_1\left(\frac{1}{160(\lambda+1)^{3/2}BN},\mathcal{G},z^N\right)\right)\exp\left(-\frac{CNt}{\lambda+1}\right)dt$$

$$= \delta + \exp(\mathfrak{R}_{\mathcal{F},\mathcal{G}}^{n,m,\lambda})\cdot\frac{5(\lambda+1)}{CN}\exp\left(-\frac{CN\delta}{\lambda+1}\right) = \frac{(\lambda+1)(5+\mathfrak{R}_{\mathcal{F},\mathcal{G}}^{n,m,\lambda})}{CN}\quad\left(\text{Take }\delta=\frac{(\lambda+1)\mathfrak{R}_{\mathcal{F},\mathcal{G}}^{n,m,\lambda}}{CN}\geq\frac{1}{N}\right)$$

$$\leq \frac{6(\lambda+1)\mathfrak{R}_{\mathcal{F},\mathcal{G}}^{n,m,\lambda}}{CN},$$

with

$$\mathfrak{R}_{\mathcal{F},\mathcal{G}}^{n,m,\lambda} := \log\left(T_n + T_m + T_{n+m}\right),$$

where for $k \in \{n, m, n+m\}$,

$$T_k := \sup_{z^k}\mathcal{N}_1\left(\frac{1}{c_k},\mathcal{F},x^k\right)\mathcal{N}_1\left(\frac{1}{d_k},\mathcal{G},z^k\right),$$

with

$$c_k = \begin{cases} 80B\sqrt{\lambda+1}\,N, & k \in \{n,m\}, \\ 160B\sqrt{\lambda+1}\,N, & k = n+m, \end{cases}$$

and

$$d_k = \begin{cases} 80B(\lambda+1)^{3/2}N, & k = n, \\ 80B\sqrt{\lambda+1}\,N, & k = m, \\ 160B(\lambda+1)^{3/2}N, & k = n+m. \end{cases}$$

Finally, observe that

$$\mathfrak{R}_{\mathcal{F},\mathcal{G}}^{n,m,\lambda} \leq \sup_{z^N}\log\left(3\mathcal{N}_1\left(\frac{1}{d_N},\mathcal{F},x^N\right)\mathcal{N}_1\left(\frac{1}{d_N},\mathcal{G},z^N\right)\right) \leq \sup_{z^N}\log\left(3\left(\mathcal{N}_{1,\infty}\left(\frac{1}{d_N},\mathcal{F}\times\mathcal{G},z^N\right)\right)^2\right),$$

$$\lesssim \sup_{z^N}\log\left(\mathcal{N}_{1,\infty}\left(\frac{1}{d_N},\mathcal{F}\times\mathcal{G},z^N\right)\right),$$

with $d_N = 160B(\lambda+1)^{3/2}N$.
Thus,

$$\mathbb{E}[\mathcal{L}(\hat{f},\hat{g};\lambda) - \mathcal{L}(f^\star,g^\star;\lambda)] - 2\mathbb{E}[\hat{\mathcal{L}}(\hat{f},\hat{g};\lambda) - \hat{\mathcal{L}}(f^\star,g^\star;\lambda)] \leq \frac{C_1B^2(\lambda+1)\mathfrak{E}_{\mathcal{F}\times\mathcal{G}}^{N,\lambda}}{N},$$

for a universal constant $C_1 > 0$.
Hence, using that $\hat{\mathcal{L}}(\hat{f},\hat{g};\lambda) = \hat{\mathcal{L}}(f^\star,g^\star;\lambda) + \hat{\Delta}(\hat{f},\hat{g})$, where $\hat{\Delta}$ is the empirical comparison term, we obtain

$$\mathbb{E}_{\mathscr{D}}[\mathcal{L}(\hat{f},\hat{g};\lambda)] - \mathcal{L}(f^\star,g^\star;\lambda) \leq \frac{C_1B^2(\lambda+1)\mathfrak{E}_{\mathcal{F}\times\mathcal{G}}^{N,\lambda}}{m+n} + 2\mathbb{E}_{\mathscr{D}}[\hat{\Delta}(\hat{f},\hat{g})]. \tag{20}$$

Combining Corollary 2.3 and (20) completes the proof.

## C.4. Proof of Remark 2.4 (ii)

Note that for any square-integrable random functions $k_1$ and $k_2$, we have

$$1 - \frac{\mathbb{E}[\langle k_1, k_2 \rangle]^2}{\mathbb{E}[\|k_1\|^2]\mathbb{E}[\|k_2\|^2]} = \frac{\mathbb{E}[\|k_1 - ck_2\|^2]}{\mathbb{E}[\|k_1\|^2]},$$

where

$$c = \frac{\mathbb{E}[\langle k_1, k_2 \rangle]}{\mathbb{E}[\|k_2\|^2]}.$$

Applying this identity with $k_1 = g^\star(Z) - \hat{g}(Z)$ and $k_2 = f^\star(X) - \hat{f}(X)$, we obtain

$$1 - \rho_\star^2 = \frac{\mathbb{E}_{\mathscr{D},Z}\left[\left(g^\star(Z) - \hat{g}(Z) - c(f^\star(X) - \hat{f}(X))\right)^2\right]}{\mathbb{E}_{\mathscr{D},Z}\left[\left(g^\star(Z) - \hat{g}(Z)\right)^2\right]}$$

$$\geq \frac{\mathbb{E}_{\mathscr{D},X}\left[\mathrm{Var}\left(g^\star(Z) - \hat{g}(Z) - c(f^\star(X) - \hat{f}(X)) \mid \mathscr{D}, X\right)\right]}{\mathbb{E}_{\mathscr{D},Z}\left[\left(g^\star(Z) - \hat{g}(Z)\right)^2\right]}$$

$$= \frac{\mathbb{E}_{\mathscr{D},X}\left[\mathrm{Var}\left(g^\star(Z) - \hat{g}(Z) \mid \mathscr{D}, X\right)\right]}{\mathbb{E}_{\mathscr{D},Z}\left[\left(g^\star(Z) - \hat{g}(Z)\right)^2\right]},$$

where the inequality follows from the law of total variance.
Rearranging yields

$$\rho_\star^2 \leq \frac{\mathbb{E}_{\mathscr{D},X}\left[\left(\mathbb{E}\left[g^\star(Z) - \hat{g}(Z) \mid \mathscr{D}, X\right]\right)^2\right]}{\mathbb{E}_{\mathscr{D},Z}\left[\left(g^\star(Z) - \hat{g}(Z)\right)^2\right]}.$$

## C.5. Proof of Theorem 3.1

For comparison element $h = (f, g)$ with atomic norms.

$$C_f := \|f\|_{L^1(\mathcal{D}_f)}, \quad C_g := \|g\|_{L^1(\mathcal{D}_g)}, \quad C_h := C_f + C_g.$$

Define the excess energy.

$$a_k := \|r_k\|_\star^2 - \|(Y, Y) - h\|_\star^2$$

and the norms

$$\alpha_k = \|\Pi_{S_k^f} r_k\|_\star, \quad \beta_k = \|\Pi_{S_k^g} r_k^{(g)}\|_\star.$$

By orthogonal projection properties,

$$\|r_{k+1}\|_\star^2 = \|r_k^{(g)}\|_\star^2 - \beta_k^2 = \|r_k\|_\star^2 - \alpha_k^2 - \beta_k^2 = \|r_k\|_\star^2 - c_k^2,$$

where $c_k^2 = \alpha_k^2 + \beta_k^2 = a_k - a_{k+1}$. To prove Theorem 3.1, we first prove the following lemmas.

**Lemma C.4.** *For every $k \geq 1$,*

$$A_k = \sup_{a \in \mathcal{D}_f^\star} \langle r_k, a \rangle_\star \leq R_\lambda \alpha_k, \quad B_k = \sup_{b \in \mathcal{D}_g^\star} \langle r_k, b \rangle_\star \leq R_\lambda \alpha_k + R_\lambda \beta_k.$$

*Proof.* Write $r_k = r_k^{(g)} + \Pi_{S_k^f} r_k$ with $r_k^{(g)} \perp S_k^f$ by Step 1. For any $b \in \mathcal{D}_g^\star$,

$$\langle r_k, b \rangle_\star = \langle r_k^{(g)}, b \rangle_\star + \langle \Pi_{S_k^f} r_k, b \rangle_\star \leq \sup_{b \in \mathcal{D}_g^\star} \langle r_k^{(g)}, b \rangle_\star + R_\lambda \|\Pi_{S_k^f} r_k\|_\star,$$

where the last inequality uses Cauchy–Schwarz and $\|b\|_\star \leq R_\lambda$. For $a \in \mathcal{D}_f^\star$, by definition of $\psi_k^\star$ we have

$$\sup_{a \in \mathcal{D}_f^\star} \langle r_k, a \rangle_\star \leq R_\lambda \sup_{a \in \mathcal{D}_f^\star} \|\Pi_{S_{k-1}^f \cup \{a\}} r_k\|_\star = R_\lambda \|\Pi_{S_k^f} r_k\|_\star = R_\lambda \alpha_k.$$

Likewise,

$$\sup_{b \in \mathcal{D}_g^\star} \langle r_k^{(g)}, b \rangle_\star \le R_\lambda \|\Pi_{S_k^g} r_k^{(g)}\|_\star = R_\lambda \beta_k,$$

hence $\sup_b \langle r_k, b \rangle_\star \le R_\lambda \beta_k + R_\lambda \alpha_k$. This gives the second bound. $\square$

**Lemma C.5.** *We have*

$$\left\langle r_k, (Y,Y) \right\rangle_\star = \|r_k\|_\star^2 + \left\langle \Pi_{S_k^f} r_k, (f_{k-1}, 0) \right\rangle_\star, \quad (f_k, 0) = (f_{k-1}, 0) + \Pi_{S_k^f} r_k, \tag{21}$$

*and, by polarization of the update,*

$$\left\langle \Pi_{S_k^f} r_k, (f_{k-1}, 0) \right\rangle_\star = \frac{1}{2} \left( \left\| (f_k, 0) \right\|_\star^2 - \left\| (f_{k-1}, 0) \right\|_\star^2 - \left\| \Pi_{S_k^f} r_k \right\|_\star^2 \right). \tag{22}$$

*Equivalently,*

$$\left\langle r_k, (Y,Y) \right\rangle_\star = \|r_k\|_\star^2 + \frac{1}{2} \left( \left\| (f_k, 0) \right\|_\star^2 - \left\| (f_{k-1}, 0) \right\|_\star^2 - \left\| \Pi_{S_k^f} r_k \right\|_\star^2 \right). \tag{23}$$

*Proof.* By definition $r_k = (Y,Y) - (f_{k-1}, g_{k-1})$, and the Step 1 update is the orthogonal projection in the $\star$–inner product, hence

$$(f_k, g_{k-1}) = (f_{k-1}, g_{k-1}) + \Pi_{S_k^f} r_k, \quad \text{so} \quad (f_k, 0) = (f_{k-1}, 0) + \Pi_{S_k^f} r_k.$$

For (21), write

$$\left\langle r_k, (Y,Y) \right\rangle_\star = \left\langle r_k, (f_{k-1}, g_{k-1}) + r_k \right\rangle_\star = \|r_k\|_\star^2 + \left\langle r_k, (f_{k-1}, g_{k-1}) \right\rangle_\star.$$

Since $(f_{k-1}, 0) \in \mathrm{span}(S_k^f)$ and $\Pi_{S_k^f} r_k$ is the orthogonal projection onto $S_k^f$,

$$\left\langle r_k, (f_{k-1}, 0) \right\rangle_\star = \left\langle \Pi_{S_k^f} r_k, (f_{k-1}, 0) \right\rangle_\star.$$

Moreover the previous step ends with $r_k \perp S_{k-1}^g$ and $(0, g_{k-1}) \in S_{k-1}^g$, hence $\left\langle r_k, (0, g_{k-1}) \right\rangle_\star = 0$, proving (21). For (22), expand the square using $(f_k, 0) = (f_{k-1}, 0) + \Pi_{S_k^f} r_k$.

$$\left\| (f_k, 0) \right\|_\star^2 = \left\| (f_{k-1}, 0) \right\|_\star^2 + 2 \left\langle \Pi_{S_k^f} r_k, (f_{k-1}, 0) \right\rangle_\star + \left\| \Pi_{S_k^f} r_k \right\|_\star^2,$$

and rearrange to obtain (22). Substituting (22) into (21) yields (23). $\square$

**Lemma C.6.** *For every $k \ge 1$,*

$$\sum_{i=2}^{k+1} a_i \le 2\sqrt{2} C_h R_\lambda \sum_{i=1}^{k} c_i, \qquad C_h = C_f + C_g, \quad c_i = \sqrt{a_i - a_{i+1}}.$$

*Proof.* Let $e := (Y,Y) - h$, $r_k := (Y,Y) - (f_{k-1}, g_{k-1})$, and recall

$$a_k := \|r_k\|_\star^2 - \|e\|_\star^2, \qquad c_k^2 = a_k - a_{k+1} = \alpha_k^2 + \beta_k^2.$$

By Lemma C.5,

$$\|r_k\|_\star^2 = \left\langle r_k, (Y,Y) \right\rangle_\star - \left\langle \Pi_{S_k^f} r_k, (f_{k-1}, 0) \right\rangle_\star.$$

Then, by arithmetic–geometric mean inequality

$$a_k = \|r_k\|_\star^2 - \|e\|_\star^2 = \left\langle r_k, e \right\rangle_\star + \left\langle r_k, (f, 0) \right\rangle_\star + \left\langle r_k, (0, g) \right\rangle_\star - \left\langle \Pi_{S_k^f} r_k, (f_{k-1}, 0) \right\rangle_\star - \|e\|_\star^2$$

$$\le \frac{1}{2} a_k + \left\langle r_k, (f, 0) \right\rangle_\star - \left\langle \Pi_{S_k^f} r_k, (f_{k-1}, 0) \right\rangle_\star + \left\langle r_k, (0, g) \right\rangle_\star,$$

and we use the bounds $\left\langle r_k, (f, 0) \right\rangle_\star \le C_f A_k$ and $\left\langle r_k, (0, g) \right\rangle_\star \le C_g B_k$ to control the two correlation terms, hence

$$\frac{a_k}{2} \le C_f A_k + C_g B_k - \left\langle \Pi_{S_k^f} r_k, (f_{k-1}, 0) \right\rangle_\star.$$

Apply the polarization identity (Lemma C.5) to the $f$-history:

$$\langle \Pi_{S_k^f} r_k, (f_{k-1}, 0) \rangle_\star = \frac{1}{2} \left( \|(f_k, 0)\|_\star^2 - \|(f_{k-1}, 0)\|_\star^2 - \alpha_k^2 \right).$$

Therefore,

$$\frac{a_k}{2} \le C_f A_k + C_g B_k + \frac{1}{2} \left( \|(f_{k-1}, 0)\|_\star^2 - \|(f_k, 0)\|_\star^2 \right) + \frac{1}{2} \alpha_k^2.$$

Summing over $i = 1, \ldots, k$ (with $f_0 = 0$) yields

$$\sum_{i=1}^{k} \frac{a_i}{2} \le C_f \sum_{i=1}^{k} A_i + C_g \sum_{i=1}^{k} B_i - \frac{1}{2} \|(f_k, 0)\|_\star^2 + \frac{1}{2} \sum_{i=1}^{k} \alpha_i^2. \tag{24}$$

Since $\sum_{i=1}^{k} \alpha_i^2 \le \sum_{i=1}^{k} c_i^2 = \sum_{i=1}^{k} (a_i - a_{i+1}) = a_1 - a_{k+1}$, $A_i \le R_\lambda \alpha_i$, and $B_i \le R_\lambda \alpha_i + R_\lambda \beta_i \le \sqrt{2} R_\lambda c_i$, we get

$$C_f \sum_{i=1}^{k} A_i + C_g \sum_{i=1}^{k} B_i \le C_f R_\lambda \sum_{i=1}^{k} c_i + \sqrt{2} C_g R_\lambda \sum_{i=1}^{k} c_i \le \sqrt{2}(C_f + C_g) R_\lambda \sum_{i=1}^{k} c_i = \sqrt{2} C_h R_\lambda \sum_{i=1}^{k} c_i.$$

Dropping the nonpositive term $-\frac{1}{2} \|(f_k, 0)\|_\star^2$ in (24) gives

$$\sum_{i=1}^{k} \frac{a_i}{2} \le \sqrt{2} C_h R_\lambda \sum_{i=1}^{k} c_i + \frac{a_1 - a_{k+1}}{2},$$

which implies

$$\sum_{i=2}^{k+1} a_i \le 2\sqrt{2} C_h R_\lambda \sum_{i=1}^{k} c_i.$$

$\square$

We are ready to prove Theorem 3.1.

*Proof.* Set

$$S_k := \sum_{i=1}^{k} a_i, \qquad H_k := \sum_{i=1}^{k} \frac{1}{i}.$$

We have that for every $k \ge 1$,

$$\sum_{i=2}^{k+1} a_i \le 2\sqrt{2} C_h R_\lambda \sum_{i=1}^{k} c_i, \qquad c_i = \sqrt{a_i - a_{i+1}}. \tag{25}$$

*Summation by parts (discrete Abel transform).* For every $k \ge 1$,

$$\sum_{i=1}^{k} i(a_i - a_{i+1}) = \sum_{i=1}^{k} i c_i^2 = S_k - k a_{k+1}. \tag{26}$$

Indeed,

$$\sum_{i=1}^{k} i(a_i - a_{i+1}) = \sum_{i=1}^{k} \sum_{j=1}^{i} (a_i - a_{i+1}) = \sum_{j=1}^{k} \sum_{i=j}^{k} (a_i - a_{i+1}) = \sum_{j=1}^{k} (a_j - a_{k+1}) = S_k - k a_{k+1}.$$

*Cauchy–Schwarz with weights and* (26)*.* By Cauchy–Schwarz,

$$\left( \sum_{i=1}^{k} c_i \right)^2 = \left( \sum_{i=1}^{k} \frac{1}{\sqrt{i}} \sqrt{i} c_i \right)^2 \le \left( \sum_{i=1}^{k} \frac{1}{i} \right) \left( \sum_{i=1}^{k} i c_i^2 \right) = H_k \left( S_k - k a_{k+1} \right) \le H_k S_k.$$

Using (25) and $S_k \leq S_{k+1}$,

$$S_{k+1} - a_1 = \sum_{i=2}^{k+1} a_i \leq 2\sqrt{2}C_h R_\lambda \sum_{i=1}^{k} c_i \leq 2\sqrt{2}C_h R_\lambda \sqrt{H_k S_{k+1}}.$$

Then

$$S_{k+1} \leq a_1 + 2\sqrt{2}C_h R_\lambda \sqrt{H_k S_{k+1}} \leq a_1 + S_{k+1}/2 + 4C_h^2 R_\lambda^2 H_k,$$

so $S_{k+1} \leq 2a_1 + 8C_h^2 R_\lambda^2 H_k$. Hence for all $k \geq 1$,

$$S_k \leq 2a_1 + 8C_h^2 R_\lambda^2 H_k. \tag{27}$$

*Pointwise rate.* Since $(a_k)$ is decreasing, $ka_k \leq S_k$. Combining with (27),

$$a_k \leq \frac{2a_1}{k} + \frac{8C_h^2 R_\lambda^2 H_k}{k} \leq C\frac{C_h^2 R_\lambda^2 \log(k+1)}{k},$$

for a universal constant $C > 0$, completing the proof. $\square$

