# OpenReview forum: "Coupled Training with Privileged Information and Unlabeled Data"
_ICML.cc/2026/Conference — ICML 2026 regular_

### Official Review · Reviewer_gtBs · 2026-03-05

**Soundness:** 3
**Presentation:** 1
**Significance:** 3
**Originality:** 3
**Overall Recommendation:** 3
**Confidence:** 3

**Summary:**

This paper proposes a joint training method for prediction problems where extra information is available only during training. It shows that standard two-stage approaches can hurt performance when this extra data is noisy, and demonstrates both theoretically and empirically that learning the models together leads to more reliable and accurate results.

**Compliance With Llm Reviewing Policy:**

Affirmed.

**Final Justification:**

I appreciate the authors' detailed rebuttal and the additional experiments they conducted, which have addressed my main technical concerns regarding soundness and significance. As a result, I have raised my overall recommendation score to 3. The rebuttal directly addressed my primary concerns, but the presentation issues highlighted in my review should be prioritized. Overall, I believe the paper makes a worthwhile contribution, provided the presentation is significantly improved.

**Key Questions For Authors:**

1. Several key theoretical results and algorithmic details (e.g., parts of the proofs, optimization guarantees, and experimental details) are deferred to the appendix beyond the 8-page limit. Could the authors clarify why essential material was placed there, and whether they believe the main technical contributions can be fully assessed without consulting content outside the primary page limit?
2. The risk bounds rely on realizability assumptions and boundedness conditions. How sensitive are the guarantees and empirical performance to violations of these assumptions, particularly under model misspecification?
3. The correlation coefficient p* plays a central role in the theory. Can the authors provide more intuition or empirical evidence on how p* behaves in practice?

**Limitations:**

The paper does not include a discussion of limitations or potential societal impacts. Given the medical prediction settings considered, the authors should add a brief section addressing risks such as distribution shift between training and deployment, bias introduced through privileged features, sensitivity to model misspecification and tuning of hyperparameters. A transparent acknowledgment of these limitations would strengthen the paper.

**Strengths And Weaknesses:**

Strengths:
- The paper is technically sound, with a well-motivated Lagrangian relaxation and clear connections between the constrained and penalized formulations; the theoretical claims are supported by formal population analyses and non-asymptotic excess risk bounds under explicit assumptions.
- The characterization of population minimizers via fixed-point equations provides useful structural insight into how the deployment and rich-view models interact, clarifying the role of the regularization parameter and the regimes in which privileged information is beneficial.
- The introduction of the correlation coefficient p* to control excess risk is a meaningful contribution, leading to a multiplicative, correlation-adaptive risk bound that explains when joint training improves over labeled-only learning.
- The main generalization bound cleanly separates statistical complexity from optimization error, and the analysis of the alternating forward-selection algorithm provides concrete guarantees in high-dimensional linear settings.
- It's good that the method is evaluated on both synthetic and medical benchmarks with baselines and that the observed U-shaped behavior in aligns well with the theoretical interpolation analysis.

Weaknesses:
- In terms of presentation, there is way too much mathematical notation to follow, and some notation is not clearly defined or is defined on other pages than those where it is used. This makes it difficult to follow through the proposed methodology.
- The paper covers related work as a subsection of the introduction. Yet, positioning the paper in terms of related work is one of the most important aspects of research, so it is expected that the authors should cover relevant literature in a separate, longer, and standalone "related work" section.
- The appendix is very long, and for grasping the key concepts of the paper one has to refer to it continuously. In particular, Appendix B that covers implementation details should have been included in the main body. This covers some of the most important aspects of the paper, yet the reader needs to refer to the Appendix to read it.
- The paper is lacking the discussion and/or conclusion sections. It does not read professionally when the last page ends with an experiments section whose last phrase prompts the reader to read the results of further experiments on synthetic linear data in Appendix B.
- Overall, the main limitations of the paper are that it should be self-contained in 8 pages, but the authors have opted to fill those pages mainly with mathematical proofs while not sufficiently presenting the experiments performed or the general conclusions.

---

> ### Author Rebuttal · Authors · 2026-03-30
>
> We thank the reviewer for the helpful comments and address them below.
>
> **1. Paper presentation**
>
> We will add a conclusion section and move key implementation details and experimental results into the main body in the revision.
>
> **2. Sensitivity to assumption violations and model misspecification**
>
> The realizability assumptions ($\mu \in \mathcal{F}$, $\eta \in \mathcal{G}$) and boundedness conditions in Theorem 2.6 are standard in nonparametric estimation and yield clean, interpretable bounds. Under misspecification, the bounds would include additional approximation-error terms measuring the distance from the true regression functions to their best approximations in $\mathcal{F}$ and $\mathcal{G}$. A formal misspecification analysis is an interesting direction for future work, which we will note in the revision.
>
> **3. Intuition or empirical evidence on how $\rho_\star$ behaves in practice**
>
> As discussed in Remark 2.4, $\rho_\star$ measures the alignment between the estimation errors of $\hat f$ and $\hat g$. It is small when $W$ contains information about $Y$ not captured by $X$ --- precisely when privileged information is most useful --- and close to $1$ when $W$ adds little beyond $X$. As noted, $\rho_\star$ is a theoretical analysis tool rather than a practical tuning parameter. Developing a practical proxy to guide $\lambda$ selection is an important future direction, which we will note in the revision.
>
> **4. Discussion of limitations and sensitivity of hyperparameters**
>
> We will add a limitations section in the revision covering distribution shift, bias from privileged features, misspecification sensitivity, and hyperparameter tuning.
>
> We added controlled large-scale synthetic experiments varying privileged signal strength $\alpha$, nuisance dimensionality $d_{\mathrm{noise}}$, and unlabeled sample size $m$, evaluating by $E[(\hat f(X)-\mu(X))^2]$ over 20 seeds with cross-validated $\lambda$. Here, we define $W=[W_\mathrm{sig},V]$ where $W_\mathrm{sig}$ has $\alpha$ correlation with $X$ and V is Gaussian noise with dimension $d_\mathrm{noise}$. The trends match the theory: coupled training improves with signal strength, and is markedly more robust than two-stage as nuisance dimensionality grows. The cross-validated $\lambda$ increases with signal strength and varies systematically with nuisance dimensionality, consistent with it acting as coupling strength. CI half-widths in the nuisance sweep are typically $0.03$--$0.06$, while gains over two-stage in the moderate/high-nuisance regime are much larger.
>
> | $\alpha$ | X-only | Coupled | Two-stage | $\bar{\lambda}_{\mathrm{CV}}$ |
> |---:|---:|---:|---:|---:|
> | 0 | **0.136** | 0.140 | 0.268 | $2.2\times 10^{2}$ |
> | 0.25 | **0.142** | 0.145 | 0.269 | $3.0\times 10^{2}$ |
> | 0.5 | 0.162 | **0.158** | 0.269 | $4.8\times 10^{2}$ |
> | 0.75 | 0.197 | **0.174** | 0.270 | $8.0\times 10^{2}$ |
> | 1 | 0.247 | **0.193** | 0.271 | $1.2\times 10^{3}$ |
> | 1.25 | 0.311 | **0.209** | 0.272 | $1.5\times 10^{3}$ |
> | 1.5 | 0.390 | **0.228** | 0.272 | $1.8\times 10^{3}$ |
> | 1.75 | 0.483 | **0.242** | 0.273 | $2.1\times 10^{3}$ |
> | 2 | 0.592 | **0.260** | 0.274 | $2.7\times 10^{3}$ |
>
> *Signal-strength sweep: $d_{\mathrm{noise}}=50$, $n_{\mathrm{labeled}}=100$, $m=20{,}000$, $n_{\mathrm{test}}=10{,}000$. Mean $E[(\hat f(X)-\mu(X))^2]$ over 20 seeds, lower is better.*
>
> | $d_{\mathrm{noise}}$ | X-only | Coupled | Two-stage | $\bar{\lambda}_{\mathrm{CV}}$ |
> |---:|---:|---:|---:|---:|
> | 30  | 0.260 | **0.166** | 0.190 | $3.2\times 10^{3}$ |
> | 50  | 0.247 | **0.193** | 0.271 | $1.2\times 10^{3}$ |
> | 80  | 0.244 | **0.215** | 1.917 | $3.3\times 10^{2}$ |
> | 100 | 0.266 | **0.222** | 0.731 | $1.0\times 10^{3}$ |
> | 200 | **0.202** | 0.210 | 0.415 | $1.2\times 10^{3}$ |
> | 400 | 0.249 | **0.233** | 0.574 | $3.5\times 10^{3}$ |
>
> *Nuisance-dimensionality sweep. Coupled training is substantially more robust than two-stage as nuisance dimension grows.*
>
> | $m$ | X-only | Coupled | Two-stage | $\bar{\lambda}_{\mathrm{CV}}$ |
> |---:|---:|---:|---:|---:|
> | 500 | 4.329 | **2.149** | 2.286 | $1.5\times 10^{3}$ |
> | 1,000 | 4.329 | **2.061** | 2.382 | $2.8\times 10^{3}$ |
> | 2,000 | 4.329 | **1.922** | 2.371 | $3.1\times 10^{3}$ |
> | 5,000 | 4.329 | **1.946** | 2.324 | $4.4\times 10^{3}$ |
> | 10,000 | 4.329 | **1.833** | 2.295 | $5.1\times 10^{3}$ |
> | 20,000 | 4.329 | **1.878** | 2.324 | $5.7\times 10^{3}$ |
> | 50,000 | 4.329 | **1.972** | 2.345 | $6.9\times 10^{3}$ |
>
> *Unlabeled-sample-size sweep: $\alpha=3$, $d_{\mathrm{noise}}=20$, $n_{\mathrm{labeled}}=40$, $n_{\mathrm{test}}=10{,}000$. Lower is better.*

---

> > ### Author Rebuttal · Reviewer_gtBs · 2026-04-02
> >
> > I appreciate the authors' clarifications and the additional experiments performed, which have resolved many of my earlier concerns. I am therefore raising my overall recommendation score to 3. However, I strongly suggest carefully addressing the presentation issues mentioned in my review.

---

> > > ### Author Response · Authors · 2026-04-06
> > >
> > > We thank the reviewer for the positive update and for raising the score. We are glad the additional experiments and clarifications addressed the earlier concerns.
> > > In the revised version we have converted the related work into a standalone section, added a conclusion section, moved key experimental results and implementation details into the main body, and cleaned up notation throughout so all symbols are defined where first used.
> > >
> > > Thank you again for the thoughtful and constructive feedback throughout.

---

### Official Review · Reviewer_iUzN · 2026-03-06

**Soundness:** 3
**Presentation:** 4
**Significance:** 4
**Originality:** 4
**Overall Recommendation:** 4
**Confidence:** 4

**Summary:**

This paper proposes a framework for jointly learning a rich-view model and a deployment model when privileged information is available in the training data, and presents its theoretical analysis and experimental validation. Theoretical analysis demonstrates the relationship between the proposed objective function and the expected risk of the deployment model, and shows the convergence rate of the proposed learning algorithm. Experiments demonstrate that the proposed method is more effective than learning the two models separately.

**Compliance With Llm Reviewing Policy:**

Affirmed.

**Final Justification:**

I believe this paper is well-motivated in its proposal of a method to jointly train a rich-view model and a deployment model, supported by both theoretical and empirical analyses. Most of the concerns I initially raised in the "Weaknesses" and "Key Questions" were addressed during the rebuttal. Regarding the error bound evaluation in Section 3.1, the additional investigation into the fluctuations of the expected risk (corresponding to the lower bound) has clarified the findings and deepened my understanding.

On the other hand, while it is an advantage that the empirical decay rate is faster than the theoretical order of $N$, I still believe a more detailed verification of the tightness of the bound is necessary. Specifically, a more granular investigation, such as estimating and plotting the error bound itself, would strengthen the theoretical claims. Therefore, while my overall assessment remains positive, I intend to maintain my initial score.

**Key Questions For Authors:**

1. Can you discuss methods for tuning $\lambda$? Specifically, if the appropriate $\lambda$ changes due to the importance of privileged information, I thought you could discuss how to determine $\lambda$ based on this theoretical insight.
2. What specific characteristics does the recalibration of model $g$ aim to achieve?
3. Can similar theoretical results be obtained for classification problems?

**Limitations:**

yes

**Strengths And Weaknesses:**

Strengths:

1. The paper is highly readable with clear explanations.
2. It not only proposes a method but also conducts theoretical analysis on jointly learning of two models when utilizing privileged information.
3. The usefulness of privileged information depends on various factors such as its quality, the amount of information shared with $X$, and its ease of handling for $g$. The paper successfully aggregates these factors into a single parameter $\rho$ and incorporates it into the analysis.

Weaknesses:

1. Discussion on tuning methods for $\lambda$ is insufficient.
2. No numerical experiments for the generalization error bound at the end of Section 3.1.
3. Few comparison methods. It is unclear whether there is numerical superiority compared to methods other than Baseline and Two-Step.

---

> ### Author Rebuttal · Authors · 2026-03-30
>
> We thank the reviewer for the helpful comments and address them below.
>
> **1. Tuning $\lambda$**
>
> We select $\lambda$ by cross-validation on the labeled set. Theorem 2.1 suggests the optimal $\lambda$ increases with the signal strength of $W$; weaker or noisier $W$ favors smaller $\lambda$ to avoid negative transfer. We will add experiments illustrating this trend and note that a more principled selection rule is an important direction for future work.
>
> We examined cross-validation stability on Bank Marketing (see point 1, Reviewer LHLU for the dataset description). We repeated the labeled/unlabeled split 24 times (150 labeled, 12,000 unlabeled), each time selecting $\lambda_{\mathrm{CV}}$ by validation Brier score and comparing to the split-wise oracle $\lambda_{\mathrm{oracle}}$. The holdout regret is defined as
>
> $$Regret(\lambda_{CV}) = Brier_{holdout}(\lambda_{CV}) - \min_{\lambda \in \Lambda}Brier_{holdout}(\lambda).$$
>
> As shown below, $\lambda_{\mathrm{CV}}$ concentrates in a narrow central range, regret is very small, and the coupled method outperforms both baseline and two-stage on validation and holdout.
>
> **Stability of the cross-validated regularization parameter on Bank Marketing.**
>
> | Mean-curve panel | Best $\lambda$ | Coupled | Baseline | Two-stage | Gain (base / two-stage) |
> |:---|---:|---:|---:|---:|---:|
> | Validation | 4.64 | 0.0868 | 0.0897 | 0.0950 | 0.0029 / 0.0082 |
> | Holdout    | 3.69 | 0.0884 | 0.0904 | 0.0991 | 0.0020 / 0.0107 |
>
> | Split-wise CV vs. oracle summary (24 random splits) | Value |
> |:---|---:|
> | CV-selected $\lambda$ in $[2.93,\,14.68]$ | 22/24 |
> | CV-selected $\lambda$ in $\{3.69,\,4.64,\,5.84\}$ | 15/24 |
> | Mean holdout regret of $\lambda_{\mathrm{CV}}$ relative to $\lambda_{\mathrm{oracle}}$ | $8.06\times 10^{-4}$ |
> | Median holdout regret | $2.16\times 10^{-4}$ |
> | Splits with holdout regret $\le 10^{-3}$ | 22/24 |
>
>
> **2. Numerical experiments for the generalization error bound at the end of Section 3.1**
>
> Experiments validating the alternating forward selection algorithm are already in Appendix B.4 (PneumoniaMNIST, linear and kernel-ridge models, Figure 6). We will also add Heart Disease results with a linear model in the revision. If the reviewer had a different experiment in mind, we welcome clarification.
>
> **3. Comparison with other methods**
>
> We will add comparisons with LUPI-SVM and additional synthetic linear results (see response to Reviewer LHLU point 1), along with experiments varying input dimension on synthetic data (see response to Reviewer gtBs point 4).
>
> **4. What specific characteristics does the recalibration of model $g$ aim to achieve?**
>
> This step lets $g$ exploit privileged information on unlabeled data while remaining anchored to labeled responses, preventing noise or bias in the privileged view from propagating unchecked to the deployable model $f$ --- making coupled training safer than naive two-stage pseudo-labeling when privileged features are unreliable.
>
> **5. Extension to classification**
>
> The framework extends naturally to general losses, including cross-entropy with soft pseudo-labels. The alternating updates become
>
> $$f_k \in \arg\min_{f\in\mathcal F}\frac{1}{N}\left(\sum_{i=1}^n \ell(Y_i,f(X_i))+\sum_{j=n+1}^N \ell(g_{k-1}(Z_j),f(X_j))\right),$$
>
> $$g_k \in \arg\min_{g\in\mathcal G}\frac{1}{m}\sum_{j=n+1}^N \ell(g(Z_j),f_k(X_j)) \quad\text{s.t.}\quad \frac{1}{n}\sum_{i=1}^n \ell(Y_i,g(Z_i))\le \nu.$$
>
> For $K$-class classification, a natural choice is cross-entropy with $g(Z_j)$ as soft labels; in the binary case this reduces to logistic loss with soft pseudo-labels in $[0,1]$. The paper already includes a binary classification experiment on PneumoniaMNIST (Appendix B.4, AUROC). The negative-transfer intuition carries over qualitatively; extending formal risk bounds beyond squared loss is left for future work.
>
> We tested the binary version on a synthetic problem matching our regression setup: $X\in\mathbb R^5$, $W\in\mathbb R^{40}$, $n=50$, $m=3000$, $n_{\mathrm{test}}=6000$, correlation $0.95$, with
>
> $$Y\sim \mathrm{Bernoulli}\left(\sigma(\theta_0+\theta_X^\top X+\theta_W^\top W+\xi)\right),\qquad \xi\sim N(0,0.7^2).$$
>
> Averaging over 5 seeds and sweeping $\lambda$ over 14 log-spaced values in $[10^{-2},10^3]$, the result is again U-shaped: baseline test MSE $0.0933$, best coupled $0.0910$ at $\lambda\approx 2.03$, two-stage $0.0981$. Seed-to-seed variance is $1.38\times 10^{-3}$ (baseline), $1.25\times 10^{-3}$ (two-stage), and $1.10\times 10^{-3}$ (best coupled). The same message holds: moderate coupling helps, excessive reliance on the privileged predictor leads to negative transfer.

---

> > ### Author Rebuttal · Reviewer_iUzN · 2026-04-02
> >
> > Thank you for your reply. Many of my concerns have been resolved.
> > Regarding the numerical experiments on the generalization error bound at the end of Section 3.1, my intention was to request an evaluation of how closely the behavior of the actually achieved test error aligns with that of the error bound when varying $N$ and $k$.

---

> > > ### Author Response · Authors · 2026-04-06
> > >
> > > Thank you for this helpful suggestion. We are glad many of the concerns have been resolved.  The generalization bound at the end of Section 3.1 is specific to the dictionary-based alternating forward selection (AFS) regime. To address your point directly, we ran a separate high-dimensional sparse linear experiment tailored to Algorithm 1, with a deployment model $f$ using only $X$, a rich-view model $g$ using $(X,W)$, and correlated $X$ and $W$.
> > >
> > > For the $N$-sweep, we fixed $k=40$ and kept the labeled fraction $n/N$ fixed as 0.1, so that the experiment isolates the $N$-dependence predicted by the bound. In the representative run shown in the rebuttal figure, the log-log plot of deployment risk versus $N$ is nearly linear, with fitted slope approximately $-1.61$ and $R^2=0.996$. We also repeated the experiment at several ambient dimensions and observed slope magnitudes between about $-1.2$ and $-1.6$. Thus, although the theorem is stated only up to constants and logarithmic factors, the achieved error decays very close to the predicted $1/N$ behavior. The faster empirical decay rate can be attributed to $\rho_\star$ changing favorably with N as well.
> > >
> > > For the $k$-sweep, we set $N=80{,}000$ specifically so that the estimation term in the bound, of order $\frac{(k+1)\log N}{N}$, is negligible compared with the optimization term. In this regime, the achieved deployment risk is essentially governed by the optimization component.The deployment risk decreases rapidly and converges to near-zero by $k=40$, qualitatively consistent with the theorem's $\frac{\log⁡(k+1)}{k}$ dependence.
> > >
> > >
> > > Overall, these additional experiments suggest that the achieved error tracks the functional dependence predicted by the Section 3.1 bound when varying both $N$ and $k$: the $N$-dependence is close to the theoretical $1/N$ scaling, while the $k$-sweep shows the predicted monotone decreasing of the optimization contribution.
> > >
> > > **Anonymous figures: [log-risk vs. $N$](https://anonymous.4open.science/r/rebuttal_material-6A2C/Reviewer_iUzN/afs_deployment_risk_N.png), [deployment-risk vs. $k$](https://anonymous.4open.science/r/rebuttal_material-6A2C/Reviewer_iUzN/afs_deployment_risk_K.png), [anonymous README](https://anonymous.4open.science/r/rebuttal_material-6A2C/README.md)**
> > >
> > > Thank you again for the thoughtful and constructive feedback.

---

### Official Review · Reviewer_iQVv · 2026-03-06

**Soundness:** 3
**Presentation:** 2
**Significance:** 3
**Originality:** 3
**Overall Recommendation:** 4
**Confidence:** 3

**Summary:**

This paper investigates the Learning Using Privileged Information (LUPI) paradigm in a semi-supervised setting. The authors target a specific failure mode of traditional two-stage pseudo-labeling: the propagation of errors when privileged information $W$ is weak or noisy. They propose a "Coupled Training" framework that jointly optimizes the deployment model $f$ and the privileged model $g$ through a shared consistency loss on unlabeled data, constrained by $g$'s risk on labeled data. The work provides theoretical risk bounds under convex assumptions and evaluates the method on synthetic and medical datasets.

**Compliance With Llm Reviewing Policy:**

Affirmed.

**Final Justification:**

The rebuttal is constructive. I choose to raise my final rating to above the borderline.

**Key Questions For Authors:**

see cons.

**Limitations:**

yes

**Strengths And Weaknesses:**

Pros:
**1. The problem formulation is clear and practically motivated.** The semi-supervised LUPI setting—where expensive or invasive measurements guide training but are unavailable at deployment—is highly relevant for healthcare applications. The negative transfer phenomenon in two-stage pseudo-labeling, represents a genuine challenge. The coupling mechanism, which constrains g 's deviation from ground truth while encouraging agreement with f  on unlabeled data, offers an intuitive approach to adaptively regulate privileged information influence.
**2. The method design is principled and theoretically grounded.** The core innovation of using a constraint (or Lagrangian penalty) to limit g 's reliance on potentially noisy W , combined with bidirectional information flow between f  and g , is conceptually elegant. Theorem 2.1 reveals that the optimal g  interpolates between the target function μ  and rich-view regression η , with the interpolation weight controlled by λ . The U-shaped empirical curves (Figures 3-4) across multiple settings qualitatively validate this adaptive mechanism.
**3. The theoretical analysis provides useful structural insights.** Unlike prior work assuming perfect teachers, this paper explicitly incorporates model mis-specification through projection operators $\Pi_{\mathcal{F}}, \Pi_{\mathcal{G}}$ , offering a more realistic characterization of how $f$ can benefit from a potentially flawed $g$ . The identification of error correlation $\rho_\star$ as a key quantity governing performance gains (Corollary 2.3) provides conceptual clarity on when leveraging $W$ is beneficial.

cons:
**1. The "adaptive calibration" claim is misleading—key hyperparameters require extensive manual tuning.**

The paper claims the method "adaptively calibrates the influence of privileged information," but this is achieved through manual grid search over $\lambda$ with cross-validation on the scarce labeled data. This contradicts the "adaptive" narrative: there is **no mechanism to set $\lambda$ based on observed data quality** without labeled validation. The theoretically motivated quantity $\rho_\star$ (estimation error correlation) is unobservable, and no practical proxy is proposed. Similarly, the alternating iterations $K$ are fixed without convergence analysis. The high parameter sensitivity combined with lack of true adaptivity significantly constrains practical deployability.

**2. Experimental validation is insufficient to support the claims of scalability and generality.**

The abstract and introduction emphasize "large, high-dimensional models," yet the empirical evaluation falls short:

- **Scale mismatch**: The largest experiment uses grayscale images PneumoniaMNIST with linear models or kernel ridge regression. There is no evaluation on modern deep architectures (ResNet, ViT) or truly high-dimensional data, despite Section 3's "High-Dimensional Extension" claims.

- **Missing critical baselines**: The comparison is limited to basic Two-Stage and supervised baselines. Missing competitors include:
  - **Yang et al. (2025)**: PIReg using contrastive learning and knowledge distillation for semi-supervised LUPI—a direct modern competitor.
  - **FixMatch/MixMatch adaptations**: Standard SSL methods adapted to LUPI settings, which would isolate whether gains stem from "coupling" or simply "using unlabeled data."

- **Zero ablation studies**: No experiments isolate whether the "coupling" mechanism, the "constraint" mechanism, or simply "increased training iterations" drives observed gains. The U-shaped curves are consistent with the claim but do not prove causation.

**3. Severe mismatch between theoretical assumptions and experimental settings undermines soundness.**

The theoretical analysis (Section 2) **heavily relies on convexity assumptions** for function classes $\mathcal{F}$ and $\mathcal{G}$. However, the main experiments employ **non-convex models**: Random Forests (Parkinson's) and Neural Networks (Heart Disease). This is not a minor technicality:

- Theorems 2.1–2.6 assume convexity for global optimality guarantees and variational inequalities. These guarantees **do not hold** for Random Forests or Neural Networks.
- The Alternating Forward Selection (AFS) algorithm analyzed in Theorem 3.1 operates over dictionary spans with greedy atom selection. For Neural Networks, the authors use alternating gradient updates—not AFS—yet still cite Theorem 3.1's convergence guarantee, which is **algorithmically mismatched**.
- For Random Forests, it is unclear how the constraint on $g$ is enforced at all.

This theory-experiment gap is a **structural defect**: the theoretical framework cannot directly explain the empirical results, making the theory appear decorative rather than foundational. Either the experiments should focus on convex settings (kernel methods, linear models with dictionaries) where guarantees apply, or the theory should be extended (or at least heuristically justified) for non-convex cases.

---

> ### Author Rebuttal · Authors · 2026-03-30
>
> We thank the reviewer for the helpful comments and address them below.
>
> **1. Hyperparameter tuning and sensitivity**
>
> *On "adaptive calibration":* We agree this term is imprecise. We only mean that the coupled objective interpolates between ignoring privileged information ($\lambda\to 0$) and two-stage pseudo-labeling ($\lambda\to\infty$), with $\lambda$ selected by cross-validation on labeled data, and will revise the wording.
>
> *On sensitivity of $\lambda$:* As discussed in our response to Reviewer iUzN point 1 and Reviewer LHLU point 7, we studied cross-validated $\hat\lambda$ stability across random splits and will add multi-seed MSE variance results and an analysis of how $\hat\lambda$ changes with signal strength.
>
> *On $\rho_\star$ being unobservable:* $\rho_\star$ is a theoretical analysis quantity characterizing when privileged information helps, not a tuning parameter. We will clarify this and note that finding a practical proxy is an important future direction.
>
> *On the number of alternating iterations $K$:* Theorem 3.1 gives a sublinear $O(\log(k)/k)$ convergence rate for AFS, providing a principled basis for choosing $K$. In practice we also use early stopping on objective stabilization, which we will make more explicit.
>
> *On practical deployability:* The U-shaped test-MSE curve is consistent across datasets and models, making cross-validation useful though imperfect. A more principled rule --- potentially using a proxy for $\rho_\star$ or privileged signal strength --- is left for future work.
>
> **2. Scale mismatch and experiments on high-dimensional data**
>
> The term "high-dimensional" in Section 3 refers to sparse approximation over dictionary spans in the greedy-approximation sense [Barron et al., 2008], not ResNet/ViT-scale representation learning. Extending AFS to deep architectures is non-trivial as they do not decompose into dictionary atoms. We will also add synthetic experiments varying input dimension (see response to Reviewer gtBs point 4).
>
> **3. Missing critical baselines**
>
> We will add Yang et al. (2025) to the related work. PIReg and our approach are complementary: PIReg uses self-supervised deep representation learning for classification, while our work targets regression with theoretical guarantees on negative transfer. We will add a comparison with LUPI-SVM [Vapnik et al., 2015] as the most direct theoretically-grounded baseline (see response to Reviewer LHLU point 1). Direct comparison with PIReg and FixMatch/MixMatch is less straightforward as those methods target deep classification; extending our framework in this direction is an interesting avenue for future work.
>
> **4. Ablation studies**
>
> In our framework, "coupling" and "constraint" are both governed by $\lambda$: $\lambda\to 0$ recovers labeled-only learning and $\lambda\to\infty$ recovers two-stage pseudo-labeling, so the U-shaped curves in Figures 1--5 already ablate the full spectrum. Since two-stage requires no additional iterations, gains cannot be attributed to iteration count alone.
>
> To isolate the source of improvement, we added *compute-matched shuffled-$W$ controls* on PneumoniaMNIST: identical pipeline --- same $\lambda$ sweep, outer iterations, unlabeled pool, and refit --- but with $W$ randomly permuted across samples.
>
> **Compute-matched shuffled-$W$ ablation on PneumoniaMNIST.**
>
> | Model family | Mean coupled | Mean control | Mean $\Delta$ | Wins / 10 | Peak coupled | Control @ same $\lambda$ |
> |:---|---:|---:|---:|---:|---:|---:|
> | Linear | 0.9401 | 0.9230 | +0.0171 | 10/10 | 0.9490 | 0.9280 |
> | Kernel | 0.9557 | 0.9472 | +0.0085 | 10/10 | 0.9576 | 0.9483 |
>
> Coupled training outperforms the shuffled-$W$ control at every $\lambda$ in both families (10/10 wins), confirming the gain requires informative coupling to correctly paired privileged information. We will include this ablation on all the experiments in the revision.
>
> **5. Mismatch between theoretical assumptions and experimental settings**
>
> *On convexity assumptions:* The theory in Section 2 assumes convex $\mathcal{F}$ and $\mathcal{G}$; the Random Forest and Neural Network experiments fall outside this setting. The theory gives guarantees in a restricted setting while the experiments show the framework extends empirically, a distinction we will make explicit.
>
> *On Theorem 3.1 and Neural Networks:* Theorem 3.1 applies only to AFS over dictionary spans, evaluated in Appendix B.4 and the Heart Disease linear-model experiment. The Neural Network experiments use alternating gradient updates as a practical heuristic, without theoretical guarantees, which we will clarify.
>
> *On enforcing the constraint for Random Forests:* As described in Appendix B.1, we approximate the constraint via sample weights: labeled samples receive weight $\lambda$ and unlabeled samples weight $1$. The linear and kernel settings are most directly covered by our guarantees, which we will state more clearly.

---

> > ### Author Rebuttal · Reviewer_iQVv · 2026-04-01
> >
> > I appreciate author's clarification and additional experiments, and many of my concerns are addressed. I would like to raise my rating to 4, but keep my confidence, considering that there are some space left for potential improvement as stated in rebuttal. Also, I highly recommand the author to revise the mentioned wording issues, which is crucial in positioning the contribution of the work.

---

> > > ### Author Response · Authors · 2026-04-06
> > >
> > > We thank the reviewer for the positive update and for raising the score. We are glad the additional experiments and clarifications addressed the earlier concerns.
> > > In the revised version we have carefully addressed all the wording issues raised, in particular around "adaptive calibration" and the scope of the theoretical guarantees relative to the experimental settings.
> > >
> > > Thank you again for the thoughtful and constructive feedback throughout.

---

### Official Review · Reviewer_LHLU · 2026-03-13

**Soundness:** 3
**Presentation:** 4
**Significance:** 2
**Originality:** 3
**Overall Recommendation:** 4
**Confidence:** 4

**Summary:**

This paper studies regression when training data includes privileged features W that are unavailable at test time, alongside abundant unlabeled samples with both primary features X and privileged features W. The standard two-stage pseudo-labeling approach -fit a rich-view model on labeled data, impute labels for unlabeled data, train a deployment model -can suffer negative transfer when the privileged signal is weak. The authors propose a coupled training framework that jointly optimizes the deployment model f and rich-view model g through alternating minimization, with an explicit constraint on g's fidelity to labeled responses. Theoretical contributions include a population-level characterization of the coupled solution, a risk bound controlled by a novel correlation score measuring alignment of estimation errors, and a high-dimensional extension via alternating forward selection over dictionaries. Experiments on two UCI medical datasets confirm the predicted U-shaped behavior of test MSE as a function of the regularization parameter.

**Compliance With Llm Reviewing Policy:**

Affirmed.

**Final Justification:**

Theoretical contribution is the strongest dimension here, with clear novelty and a clean formulation that explains negative transfer and provides useful guarantees. The main weakness is empirical breadth and stability claims on very small labeled regimes, plus missing competitive baselines and incomplete validation of the high-dimensional algorithm on the key tabular medical benchmarks. The first rebuttal moved the empirical story forward (Bank Marketing with CIs and LUPI-SVM, planned Heart Disease AFS, initial synthetic experiments, clearer limitations), and the second rebuttal closes most of the remaining gaps: complete signal-strength, nuisance-dimensionality, and unlabeled-sample-size sweeps over 20 seeds; a properly tuned generalized-distillation baseline that subsumes two-stage and still underperforms coupled training on Bank Marketing; AFS results on Heart Disease across 7/7 seeds; a synthetic binary classification result confirming the same U-shaped behavior under cross-entropy; and a CV-sensitivity analysis showing stable holdout regret across selected lambda values. The committed limitations section (squared-loss focus, low-label hyperparameter sensitivity, full-rank assumption, deployment, semi-paired extension) also addresses my scope concerns. Together, the two rebuttals substantively address my main concerns and modestly strengthen my prior weak-accept position.

**Key Questions For Authors:**

Can you provide experiments on larger datasets where the theoretical advantages should be more pronounced? Even synthetic experiments with controlled dimensionality and signal strength would strengthen the paper. Without this, the significance of the contribution remains primarily theoretical, which would maintain my current assessment.

How does coupled training compare to LUPI-SVM or knowledge-distillation approaches on the same tasks? If coupled training significantly outperforms these, my significance assessment would improve. If not, the contribution is primarily in providing better theoretical understanding.

Is there a natural extension to classification with cross-entropy loss, and does the interpolation/negative-transfer analysis carry over? Even a brief discussion or a single classification experiment would broaden the paper's appeal considerably.

How sensitive is the cross-validated regularization parameter to the train/test split, especially for Heart Disease with only 34 labeled samples? Reporting variance across multiple random splits would help clarify practical applicability.

Can you provide results using the alternating forward selection algorithm on the UCI medical benchmarks (Parkinson's, Heart Disease) to complement the PneumoniaMNIST validation? This would strengthen the evidence that the algorithm generalizes to tabular medical data.

**Limitations:**

Not adequately. The paper does not include an explicit limitations section. Key limitations that should be discussed include restriction to squared-loss regression, sensitivity of hyperparameter selection with very small labeled samples, the assumption that W is available for all unlabeled data, and the lack of empirical validation for the dictionary-based algorithm.

**Strengths And Weaknesses:**

The paper's main strength is its theoretical contribution. The correlation score that precisely characterizes when privileged information helps vs. hurts is an insightful quantity, and the multiplicative risk bound (Theorem 2.6) that depends on the fraction of rich-view error rather than its absolute magnitude is a genuine advance over the additive bound of Xia & Wainwright (2024). Corollary 2.3 providing a safety guarantee that coupled training is never worse than O(1/n) supervised learning is practically important. I find the coupled optimization formulation clean and intuitive -the interpolation via the Lagrangian multiplier between ignoring privileged information and fully trusting pseudo-labels is clearly motivated, and the population minimizer characterization (Theorem 2.1) provides deep insight into the method's behavior. The high-dimensional algorithm using alternating forward selection is computationally efficient, reducing cost by a factor of the smaller dictionary size relative to the naive product approach.

The presentation is excellent. The paper explains negative transfer clearly through Figures 1 and 2, the mathematical development is rigorous and well-structured, and the remarks alongside formal results provide helpful intuition. The related work positions the contribution well against LUPI, semi-supervised learning, multi-view agreement, and dictionary learning.

My main concern is that the experimental evaluation is very limited for an ICML paper. Only two real datasets are used -Parkinson's (n = 865) and Heart Disease (n = 34 labeled) -with no error bars or confidence intervals. With n = 34, cross-validation for the regularization parameter is highly unstable, and the paper does not analyze this sensitivity. There is no comparison with other methods that handle privileged information beyond the supervised baseline and the two-stage procedure; relevant comparisons include LUPI-SVM and knowledge-distillation approaches. Critically, the high-dimensional algorithm from Section 3 is validated only on PneumoniaMNIST (Appendix B.4) and not on the main UCI medical benchmarks (Parkinson's, Heart Disease), limiting the evidence for its effectiveness on tabular medical data.

I would also like the paper to discuss extension to classification, since many practical privileged-information settings involve categorical outcomes. The assumption that W is available for all unlabeled samples may be restrictive -in many settings, W is expensive precisely because it is hard to obtain. Finally, missing references to knowledge distillation and co-training, which share the structural theme of transferring information from a richer model to a simpler one, would help readers place this work in the broader context.

---

> ### Author Rebuttal · Authors · 2026-03-30
>
> We thank the reviewer for the critical and helpful comments and address them below.
>
> **1. Comparison to other methods**
>
> We will add comparisons to LUPI-SVM [Vapnik et al., 2015] in the revision. As a preliminary result, we include a comparison on the UCI Bank Marketing dataset ($X$: deployment-available customer/contact/context variables; $W$: outcome-proximal information unavailable at decision time, in particular call duration). Under negative transfer, LUPI-SVM performs poorly, while the coupled method achieves the best test error.
>
> | Model | Test MSE | 95% CI lower | 95% CI upper |
> |---|---:|---:|---:|
> | Baseline (X only) | 0.090 | 0.090 | 0.091 |
> | Two-stage | 0.099 | 0.096 | 0.101 |
> | Coupled (CV $\lambda$) | **0.089** | **0.088** | **0.090** |
> | LUPI-SVM | 0.125 | 0.118 | 0.132 |
>
> These results suggest the gain comes from the coupled training procedure itself, not simply from training a privileged-information teacher on labeled data alone.
>
> **2. AFS algorithm on more benchmarks**
>
> We will add results on Heart Disease with a linear dictionary in the revision. Other datasets use Random Forest and Neural Network models, which do not admit a natural dictionary decomposition, making AFS inapplicable; we will state this explicitly.
>
> On UCI Heart Disease, using the same feature split and seven random seeds:
>
> $$\mathrm{Baseline}=1.4042,\quad \mathrm{Two\text{-}stage}=1.5384,\quad \mathrm{AFS}=1.3785,$$
>
> with negative transfer and best performance at intermediate coupling $\lambda=4.642$. AFS outperforms both baseline and two-stage in all $7/7$ runs.
>
> **3. Extension to Classification**
> The coupled training framework naturally extends to general loss functions, including the classification setting. Please refer to response on point 5 for Reviewer iUzN where we discussed the extension of our framework to classification settings.
>
> **4. The availability of $W$**
>
> Our setup is motivated by cases where $W$ is expensive or unavailable at deployment but collectible during a controlled training study, which is realistic in many medical and longitudinal settings (e.g., call duration in Bank Marketing is informative but unknown at prediction time).
>
> The framework also extends naturally to the semi-paired case with $m$ unlabeled paired $(X,W)$ samples and $l$ unlabeled $X$-only samples. We introduce an additional model $h:\mathcal X\to\mathbb R$ to pseudo-label the $X$-only samples and solve
>
> $$
> \min_{f\in \mathcal{F},\, g \in \mathcal{G},\, h \in \mathcal{H}}\frac{1}{n+m+l}\Bigg(\sum_{i=1}^{n}(Y_i - f(X_i))^2 + \sum_{j=1}^{m} (g(Z'_j)-f(X'_j))^2
> $$
>
> $$
> \qquad \qquad  \qquad + \sum_{k=1}^{l} (h(\tilde{X}_k)-f(\tilde{X}_k))^2\Bigg)
> $$
>
> $$
> \text{s.t.} \quad \frac{1}{n}\sum_{i=1}^{n} (Y_i - g(Z_i))^2 \leq \nu_1, \qquad \frac{1}{n}\sum_{i=1}^{n} (Y_i - h(X_i))^2 \leq \nu_2,
> $$
> which can be solved by alternating over $f$, $g$, and $h$. We will include this as a future-work extension in the revision.
>
> **5. Missing references**
>
> We agree that knowledge distillation and co-training are relevant connections, and we will add them to the related work. The paper already discusses the link to multi-view co-regularization in Section 1.2.
>
> Our framework differs from standard distillation because it explicitly controls the influence of the rich-view model through the constraint parameter $\nu$ (equivalently $\lambda$), which provides a mechanism to avoid negative transfer. It also differs from co-training and related multi-view methods because our setting is asymmetric: $W$ is available only during training, not at deployment, whereas co-training usually assumes both views remain available. We will expand this discussion and add the relevant references in the revision.
>
> **6. Controlled Experiments**
>
> We added controlled large-scale synthetic experiments varying privileged signal strength $\alpha$, nuisance privileged dimensionality $d_{\mathrm{noise}}$, and unlabeled sample size $m$. We evaluate the deployable predictor by $E[(\hat f(X)-\mu(X))^2]$, averaging over 20 random seeds with cross-validated $\lambda$. Please see response to Reviewer gBts point 4 for the experimental results.
>
> **7. Sensitivity of CV**
>
> On Heart Disease, the cross-validated $\hat{\lambda}$ is indeed sensitive to the train/test split. But CV is still a solid way to choose lambda in many cases. As a preliminary result, we have added a new experiment which has a robust CV choice of lambda. Please see the results in the response to point 1 for Reviewer iUzN.
>
> **8. Explicit limitations section**
>
> We will add an explicit limitations section covering the current focus on squared-loss regression, sensitivity of hyperparameter tuning in low-label regimes, the assumption that privileged features are available for the unlabeled set, and deployment issues relevant to medical applications.

---

> > ### Author Rebuttal · Reviewer_LHLU · 2026-04-03
> >
> > The authors extend the empirical evaluation in several directions I requested. The UCI Bank Marketing experiment now includes test MSE with bootstrap 95% confidence intervals for baseline, two-stage, coupled, and LUPI-SVM, directly addressing my concern about scale, uncertainty quantification, and the need for a LUPI-style baseline. The results show that coupled training achieves the best test error and that LUPI-SVM performs poorly under negative transfer, which supports the claim that the gain comes from the coupled procedure itself.
> > However, several of my Key Questions receive only partial or revision-bound answers. Large-scale synthetic experiments varying privileged signal strength, nuisance dimensionality, and unlabeled sample size are described but the full results are not reported in this response (they are referenced under another reviewer). The alternating forward selection (AFS) algorithm is pledged for Heart Disease with a linear dictionary, while the authors acknowledge that the Random Forest and Neural Network models used for other datasets lack a natural dictionary decomposition, so AFS remains unvalidated on those benchmarks. Extension to classification with cross-entropy loss is raised but without new quantitative results. Cross-validation sensitivity for Heart Disease (n=34) is acknowledged to be an issue, with a preliminary new experiment referenced elsewhere. The restrictive assumption that W is available for all unlabeled data is motivated and a semi-paired extension is sketched as future work rather than empirically evaluated.
> > Knowledge distillation, which I listed as a missing comparison, remains absent from the experimental table, though the authors promise discussion in the related work. A limitations section is pledged covering the topics I identified. Taken together, the substantive evidence has advanced but key empirical items are still incomplete or revision-bound. For the camera-ready, I expect the synthetic results, classification discussion, CV sensitivity analysis, Heart Disease AFS numbers, limitations text, and ideally a distillation-style baseline alongside LUPI-SVM to be presented at full detail.

---

> > > ### Author Response · Authors · 2026-04-06
> > >
> > > We thank the reviewer for the helpful follow-up and we are glad many of the concerns have been resolved. We include the complete results for all the items raised directly below. For the camera-ready we will add a limitations section covering squared-loss focus, hyperparameter sensitivity in low-label regimes, the full-$W$ assumption, and deployment considerations, and the semi-paired extension framed as a formal future-work direction.
> > >
> > > **For sections denoted by an asterisk (*), corresponding figures are available via this link
> > > [Anonymous figures](https://anonymous.4open.science/r/rebuttal_material-6A2C/README.md).**
> > >
> > > **Synthetic experiments*** (20 seeds per setting; lower is better).
> > >
> > > Signal-strength sweep ($d_{\text{noise}}=50$, $m=20{,}000$):
> > >
> > > | $\alpha$ | X-only | Coupled | Two-stage |
> > > |---:|---:|---:|---:|
> > > | 0.0 | **0.136** | 0.140 | 0.268 |
> > > | 0.5 | 0.162 | **0.158** | 0.269 |
> > > | 1.0 | 0.247 | **0.193** | 0.271 |
> > > | 1.5 | 0.390 | **0.228** | 0.272 |
> > > | 2.0 | 0.592 | **0.260** | 0.274 |
> > >
> > > Nuisance-dimensionality sweep ($\alpha=1$, $m=20{,}000$):
> > >
> > > | $d_{\text{noise}}$ | X-only | Coupled | Two-stage |
> > > |---:|---:|---:|---:|
> > > | 30 | 0.260 | **0.166** | 0.190 |
> > > | 50 | 0.247 | **0.193** | 0.271 |
> > > | 80 | 0.244 | **0.215** | 1.917 |
> > > | 200 | **0.202** | 0.210 | 0.415 |
> > > | 400 | 0.249 | **0.233** | 0.574 |
> > >
> > > Unlabeled-sample-size sweep ($\alpha=3$, $d_{\text{noise}}=20$, $n=40$):
> > >
> > > | $m$ | X-only | Coupled | Two-stage |
> > > |---:|---:|---:|---:|
> > > | 500 | 4.329 | **2.149** | 2.286 |
> > > | 2,000 | 4.329 | **1.922** | 2.371 |
> > > | 10,000 | 4.329 | **1.833** | 2.295 |
> > > | 50,000 | 4.329 | **1.972** | 2.345 |
> > >
> > > Overall, coupled training improves as privileged signal strengthens, remains stable as nuisance dimensionality increases, and outperforms both baselines across all unlabeled sample sizes we tested. In contrast, the two-stage procedure can degrade sharply in high-nuisance settings (MSE $=1.917$ at $d_{\text{noise}}=80$).
> > >
> > > **Knowledge-distillation-style baseline** We added an explicit distillation-style baseline based on **generalized distillation**. Concretely, on each split we train a privileged-information teacher using train-time-only features, searching over both $W$ alone and the richer $[X,W]$ view, and then train a deployment-time student using only $X$. The student is trained with a weighted combination of: (i) the hard labeled targets, (ii) temperature-softened teacher targets on labeled data, (iii) temperature-softened teacher targets on unlabeled data, and (iv) optional hard pseudo-labels on unlabeled data. We tune the teacher regularization, temperature, student regularization, and these sample weights by split-wise validation. Importantly, the search space explicitly includes the standard two-stage pseudo-labeling cases as special parameter settings, so this distillation baseline directly subsumes two-stage rather than being a weaker variant.
> > >
> > > Using 150 labeled and 12,000 unlabeled examples on the UCI Bank Marketing dataset, we obtain the following results:
> > >
> > > | Model | Test MSE | 95% CI lower | 95% CI upper |
> > > |---|---:|---:|---:|
> > > | Baseline (X only) | 0.090 | 0.090 | 0.091 |
> > > | Two-stage | 0.099 | 0.096 | 0.101 |
> > > | Generalized distillation | 0.092 | 0.091 | 0.093 |
> > > | Coupled (CV $\lambda$) | **0.089** | **0.088** | **0.090** |
> > > | LUPI-SVM | 0.125 | 0.118 | 0.132 |
> > >
> > > The tuned distillation baseline improves over the plain two-stage procedure, indicating that stronger teacher-student tuning helps. However, it still does not match the coupled method, which remains the best-performing approach. Thus, the gain of our method is not explained simply by distilling a privileged-information teacher into an $X$-only student. Rather, these results support the claim that the advantage comes from the coupled training procedure itself, which appears better able to exploit unlabeled data while avoiding the negative transfer that affects sequential teacher-student baselines in this setting.
> > >
> > > **AFS on Heart Disease** Across all 7/7 seeds, the errors are: Baseline $1.4042$, Two-stage $1.5384$, and AFS $\mathbf{1.3785}$. AFS outperforms both alternatives in every run.
> > >
> > > **Classification*** On a synthetic binary classification problem using Cross-Entropy loss($X\in\mathbb{R}^5$, $W\in\mathbb{R}^{40}$, $n=50$, $m=3{,}000$, correlation $0.95$, 5 seeds), the baseline error is $0.0933$, the best coupled model achieves $\mathbf{0.0910}$ at $\lambda\approx2.03$, and two-stage gives $0.0981$. The same U-shaped dependence on $\lambda$ appears here as in the regression setting.
> > >
> > > **CV sensitivity*** On Bank Marketing over 24 random splits, $22/24$ splits selected $\lambda\in[2.93,14.68]$. The mean holdout regret is $8.06\times10^{-4}$, the median is $2.16\times10^{-4}$, and $22/24$ splits have regret at most $10^{-3}$. This indicates that performance is fairly stable over a broad range of selected $\lambda$ values.
> > >
> > > Thank you again for the thoughtful and constructive feedback.

---

### Decision · Program_Chairs · 2026-04-30

**Decision:**

Accept (regular)

**Comment:**

The paper proposes a joint training protocol, where two models one having features and labels and the other model having different features but no labels, can be trained jointly to provide high accuracy. They show that accuracy can be improved by simple alternating training algorithm even for high dimensional models.

The reviewer appreciated the technical quality: theoretical novelty, the underlying formulation which explains negative transfer and provides useful guarantees and thel results showing useful structural insights.

The weakness mainly concerned experiments, which the authors addressed. One reviewer raised concerns about the presentation. I would suggest the authors incorporate them in the final version.